# Expressive Power of Graph Neural Networks for (Mixed-Integer) Quadratic Programs

## Abstract

Quadratic programming (QP) is the most widely applied category of problems in nonlinear programming. Many applications require real-time/fast solutions, though not necessarily with high precision. Existing methods either involve matrix decomposition or use the preconditioned conjugate gradient method. For relatively large instances, these methods cannot achieve the real-time requirement unless there is an effective preconditioner. Recently, graph neural networks (GNNs) opened new possibilities for QP. Some promising empirical studies of applying GNNs for QP tasks show that GNNs can capture key characteristics of an optimization instance and provide adaptive guidance accordingly to crucial configurations during the solving process, or directly provide an approximate solution. Despite notable empirical observations, theoretical foundations are still lacking.

In this work, we investigate the expressive or representative power of GNNs, a crucial aspect of neural network theory, specifically in the context of QP tasks, with both continuous and mixed-integer settings. We prove the existence of message-passing GNNs that can reliably represent key properties of quadratic programs, including feasibility, optimal objective value, and optimal solution. Our theory is validated by numerical results.

## 1 Introduction

**Quadratic programming (QP)** is an important type of optimization problem, with extensive applications across domains such as graph matching, portfolio optimization, and dynamic control (Vogelstein et al., 2015; Markowitz, 1952; Rockafellar, 1987). The goal of QP is to minimize a quadratic objective function while satisfying specified constraints. These constraints can vary, leading to different subcategories of QP. When all the constraints are linear, we call a QP problem a linearly constrained quadratic program (LCQP). When they also involve quadratic inequalities, we call the problem a quadratically constrained quadratic program (QCQP). Furthermore, if the problem requires some variables to be integers, we call it mixed-integer QP. In this study, we focus on LCQP and its mixed-integer variant MI-LCQP.

*In many real-world applications, finding solutions quickly is crucial, even if they are not perfectly precise.* For example, in transportation systems, such as ride-hailing platforms like Uber or Lyft, matching drivers with passengers requires quick decision-making to minimize waiting times, even if the optimal solution is not attained. Similarly, in financial trading, algorithms must swiftly adjust investment portfolios in response to market changes, even if it is not the most optimal move.

Unfortunately, existing methods for solving QP often rely on some computationally expensive techniques such as matrix decomposition and the preconditioned conjugate gradient method. For instance, matrix decomposition techniques like LU decomposition typically require $\mathcal{O}(n^3)$ operations for a matrix with size $n \times n$ (Golub & Van Loan, 2013), although more advanced algorithms can achieve lower complexities. Similarly, the preconditioned conjugate gradient method involves $\mathcal{O}(n^2)$ operations per iteration, and a high condition number of the matrix can lead to slow convergence or numerical instability (Shewchuk et al., 1994). These considerations underscore the clear need for novel techniques to address the demands of real-time applications.

**Machine learning (ML)** brings new chances to QP. Recent research indicates that deep neural networks (DNNs) can significantly improve the efficiency of the QP solving process. Based on the role of DNNs in the solving process, these studies can be broadly categorized into two classes:

- **Type I:** DNNs are used to accelerate an existing QP solver by generating adaptive configurations tailored to the specific instance and context, speeding up the solving process (Bonami et al., 2018; 2022; Ichnowski et al., 2021; Getzelman & Balaprakash, 2021; Jung et al., 2022; King et al., 2024). The success of such an approach relies on DNNs' capacity to capture in-depth features of QP instances and provide customized guidance to the solver.
- **Type II:** DNNs replace or warm-start a QP solver. Here, DNNs take in a QP instance and directly output an approximate solution. This approximate solution can be used directly or as an initial solution for further refinement by a QP solver (Nowak et al., 2017; Chen et al., 2018; Karg & Lucia, 2020; Wang et al., 2020a;b; 2021; Qu et al., 2021; Gao et al., 2021; Bertsimas & Stellato, 2022; Liu et al., 2022a; Sambharya et al., 2023; Pei et al., 2023; Tan et al., 2024).

Among the various types of DNNs, this paper focuses on **graph neural networks (GNNs)** (Scarselli et al., 2008), an architecture designed for graphs and widely applied across various domains. By conceptualizing QPs as graphs (Figure 1), GNNs can efficiently handle these QP tasks (Nowak et al., 2017; Wang et al., 2020b; 2021; Qu et al., 2021; Gao et al., 2021; Tan et al., 2024; Jung et al., 2022). For instance, (Wang et al., 2021) demonstrates using GNNs to solve Lawler's QAP (Lawler, 1963) with up to $150^2$ variables, while (Wang et al., 2019; Yu et al., 2020) apply GNNs to Koopman-Beckmann's QAP (Loiola et al., 2007) with $256^2$ variables, all employing 3-layer GNNs with hidden dimensions of 512, 1024, or 2048. They exploits key strengths of GNNs: *adaptability to varying graph sizes*, allowing the same model applied to various QPs, and *permutation invariance*, ensuring consistent outputs regardless of node order.

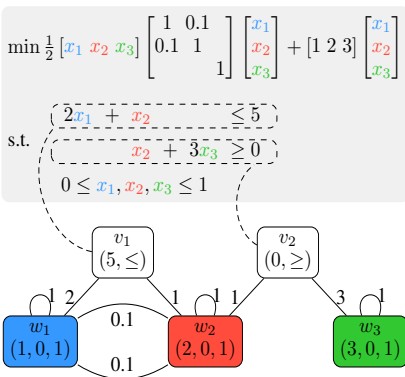

Figure 1: An illustrative example of LCQP and its graph representation.

However, despite notable empirical results, a systematic understanding of GNN for QP is still lacking. To thoroughly understand its pros and cons, some critical questions must be addressed:

- (Existence). Are there GNNs that can either capture the essential characteristics of QPs or provide approximate solutions? This question is named **the expressive power of GNNs**.
- (Trainability). If such GNNs exist, can we find them? The process of finding such GNNs is named training, which involves gathering data, creating a method to measure success or failure (a loss function), and then refining the GNN to reduce the loss function.
- (Generalization). Can a trained GNN perform effectively on QP instances it has not previously encountered? This concerns the generalization ability of GNNs.

*This paper primarily addresses the first question about expressive power.* For Type I applications, we investigate whether GNNs can accurately map a QP to its crucial features, focusing on *feasibility* and the *optimal objective value*. For Type II, we examine whether GNNs can map a QP to one of its optimal solutions. Formally, the question motivating this paper is:

$$\text{Are there GNNs that can accurately predict the feasibility,}$$
$$\text{optimal objective value, and an optimal solution of a QP?} \tag{1.1}$$

The literature has explored the expressive capabilities of GNNs on general graph tasks (Xu et al., 2019; Azizian & Lelarge, 2021; Geerts & Reutter, 2022; Zhang et al., 2023; Li & Leskovec, 2022; Sato, 2020) and their ability to approximate continuous functions on graphs (Azizian & Lelarge, 2021; Geerts & Reutter, 2022). However, significant gaps remain in understanding how these results relate to QP, as the connections between QP features (such as feasibility and optimal objective value) and graph properties have not been established. The most relevant works Chen et al. (2023a;b) investigate the representation power of GNNs for (mixed-integer) linear programs, but their analysis highly depends on the linear structure and does not cover nonlinear programs like QP.

**Contributions.** Overall, as several studies have empirically shown that incorporating a GNN can greatly improve the performance of a QP solver on specific datasets — either via GNN-generated real-time warm starts or GNN-suggested adaptive configurations — our primary aim is to theoretically investigate the expressive power of GNNs in these tasks and to determine if there is room for improvement or any considerations to be aware of. Specifically, contributions of this paper include:

- (GNN for LCQP). We provide an affirmative answer to question (1.1), establishing a theoretical foundation for using GNNs for LCQP, across both Type I and II applications.

- (GNN for MI-LCQP). In the case of MI-LCQP, our findings generally suggest a negative answer to question (1.1). However, we identify specific, precisely defined subclasses of MI-LCQP where GNNs can accurately predict feasibility, boundedness, and an optimal solution.
- (Experimental Validation). We conduct experiments that directly validate the above results.

## 2 PRELIMINARIES

This section introduces foundational concepts and preliminary definitions. We focus on linearly constrained quadratic programming (LCQP), which is formulated as follows:

$$\min_{x \in \mathbb{R}^n} \frac{1}{2} x^\top Q x + c^\top x, \quad \text{s.t. } Ax \circ b, \ l \le x \le u, \tag{2.1}$$

where $Q \in \mathbb{R}^{n \times n}$, $c \in \mathbb{R}^n$, $A \in \mathbb{R}^{m \times n}$, $b \in \mathbb{R}^m$, $l \in (\mathbb{R} \cup \{-\infty\})^n$, $u \in (\mathbb{R} \cup \{+\infty\})^n$, and $\circ \in \{\le, =, \ge\}^m$. In this paper, we always assume that $Q$ is symmetric.

**Basic concepts of LCQPs.** An $x$ satisfying all constraints of (2.1) is named a *feasible solution*. The set of all feasible solutions, defined as $X =: \{x \in \mathbb{R}^n : Ax \circ b, \ l \le x \le u\}$, is referred to as the *feasible set*. The LCQP is considered *feasible* if this set is non-empty; otherwise, it is infeasible. The value of $\frac{1}{2} x^\top Q x + c^\top x$ is named the *objective (function) value*. Its infimum across the feasible set is termed the *optimal objective value*. If this infimum is $-\infty$, suggesting the objective value could indefinitely decrease, the LCQP is deemed *unbounded*. Conversely, when the optimal objective value is finite, the corresponding $x$ is identified as an *optimal solution*.

**Graph representation of LCQPs.** We present a graph structure, termed the *LCQP-graph* $G_{\text{LCQP}} = (V, W, A, Q, H_V, H_W)$, that encodes all the elements of a LCQP (2.1). Particularly,

- The graph contains two distinct types of nodes. Nodes in $V = \{1, 2, \ldots, m\}$, labeled as $i$, represent the $i$-th constraint and are called *constraint nodes*. Nodes in $W = \{1, 2, \ldots, n\}$, labeled as $j$, represent the $j$-th variable and are known as *variable nodes*. The union set $V \cup W$ includes all the vertices of the entire LCQP-graph $G_{\text{LCQP}}$.
- The graph comprises two distinct edge types. An edge connects $i \in V$ to $j \in W$ if $A_{ij}$ is nonzero, with $A_{ij}$ serving as the edge weight. Similarly, the edge between nodes $j, j' \in W$ exists if $Q_{jj'} \ne 0$, with $Q_{jj'}$ as the edge weight. Self loops ($j = j'$) are permitted.
- Attributes/features $v_i = (b_i, \circ_i)$ are attached to the $i$-th constraint node for $i \in V$. The collection of all such attributes is denoted as $H_V = (v_1, v_2, \ldots, v_m)$.
- Attributes/features $w_j = (c_j, \ell_j, u_j)$ are attached to the $j$-th variable node for $j \in W$. The collection of all such attributes is denoted as $H_W = (w_1, w_2, \ldots, w_n)$.

Such a representation is illustrated by an example shown in Figure 1 and it can be regarded as fundamental since it is minimal in the sense that every entry in $(A, b, c, Q, l, u, \circ)$ is used exactly once. To the best of our knowledge, this particular representation is only detailed in Jung et al. (2022), yet it forms the foundation or core module for numerous related studies. For instance, removing nodes in $V$ and their associated edges reduces the graph into the assignment graph used in graph matching problems (Nowak et al., 2017; Wang et al., 2020b; 2021; Qu et al., 2021; Gao et al., 2021; Tan et al., 2024). In these cases, the linear constraints $Ax \circ b$ are typically bypassed by applying the Sinkhorn algorithm to ensure that $x$ meets these constraints. Another scenario involves LP and MILP: removing edges associated with $Q$ simplifies the graph to a bipartite structure, which reduces the LCQP to an LP (Chen et al., 2023a; Fan et al., 2023; Liu et al., 2024; Qian et al., 2024). Further, by incorporating an additional node feature, an approach detailed in Section 4, this bipartite graph is also capable of representing MILP (Gasse et al., 2019; Chen et al., 2023b; Nair et al., 2020; Gupta et al., 2020; Shen et al., 2021; Gupta et al., 2022; Khalil et al., 2022; Paulus et al., 2022; Scavuzzo et al., 2022; Liu et al., 2022b; Huang et al., 2023).

**GNNs for solving LCQPs.** Building on the established concepts, we present *message-passing graph neural networks (hereafter referred to simply as GNNs)* tailored for LCQPs using LCQP-graphs. These GNNs take in an LCQP-graph $G_{\text{LCQP}}$ (including all the node attributes and edge weights) as input and update node attributes sequentially across layers via a message-passing mechanism. Initially, node attributes $s_i^0, t_j^0$ are computed using embedding mappings $f_0^V, f_0^W$:

- $s_i^0 = f_0^V(v_i)$ for $i \in V$, and $t_j^0 = f_0^W(w_j)$ for $j \in W$.

The architecture includes $L$ standard **message-passing** layers where each layer (where $1 \le l \le L$) updates node attributes by locally aggregating neighbor information:

- $s_i^l = f_l^V\left(s_i^{l-1}, \sum_{j \in W} A_{ij} g_l^W(t_j^{l-1})\right)$ for $i \in V$, and
- $t_j^l = f_l^W\left(t_j^{l-1}, \sum_{i \in V} A_{ij} g_l^V(s_i^{l-1}), \sum_{j' \in W} Q_{jj'} g_l^Q(t_{j'}^{l-1})\right)$ for $j \in W$.

Finally, there are two types of output layers. For applications where the GNN maps LCQP-graphs to a singular real value, such as evaluating properties like feasibility of the LCQP, a **graph-level output** layer is employed that computes a single real number encompassing the entire graph:

- $y = r_1\left(\sum_{i \in V} s_i^L, \sum_{j \in W} t_j^L\right) \in \mathbb{R}$.

Alternatively, if the GNN is required to map the LCQP-graph to a vector $y \in \mathbb{R}^n$, assigning a real number to each variable node as its output (as is typical in applications where GNNs are used to predict solutions), then a **node-level output** should be utilized. This output layer computes the value for the $j$-th output as follows:

- $y_j = r_2\left(\sum_{i \in V} s_i^L, \sum_{j \in W} t_j^L, t_j^L\right)$.

In our theoretical analysis, we assume all the mappings $f_l^V, f_l^W$ ($0 \le l \le L$), $g_l^V, f_l^W, g_l^Q$ ($1 \le l \le L$), and $r_1, r_2$ to be continuous. In practice, these continuous mappings are learned from data. We aim to find mappings that enable all the LCQP-graphs $G_{\text{LCQP}}$ from a dataset to be mapped accurately to their desired outputs $y$. To achieve this, we parameterize these mappings using multilayer perceptrons (MLPs) and optimize them within the parametric space.

**Definition 2.1** (Space of LCQP-graphs and space of GNNs). *The set of all LCQP-graphs, denoted as $\mathcal{G}_{\text{LCQP}}^{m,n}$[1], comprises graphs with $m$ constraints and $n$ variables, where the matrix $Q$ is symmetric.*

**Definition 2.2** (Spaces of GNNs). *The collection of all message-passing GNNs, denoted as $\mathcal{F}_{\text{LCQP}}$ for graph-level outputs (or $\mathcal{F}_{\text{LCQP}}^W$ for node-level outputs), consists of all GNNs constructed using continuous mappings $f_l^V, f_l^W$ ($0 \le l \le L$), $g_l^V, f_l^W, g_l^Q$ ($1 \le l \le L$), and $r_1$ (or $r_2$).*

Note that the input graph size for GNNs within $\mathcal{F}_{\text{LCQP}}$ and $\mathcal{F}_{\text{LCQP}}^W$ is unspecified, as the functions $f_l^V, f_l^W$ ($0 \le l \le L$), $g_l^V, f_l^W, g_l^Q$ ($1 \le l \le L$), and $r_1$ (or $r_2$) are independent of $m, n$. This independence highlights a key advantage of GNNs discussed in Section 1: their adaptability to various graph sizes, allowing the same model to be consistently applied across different QPs.

**Definition 2.3** (Target mappings). *We define three mappings for LCQPs.*

- *Feasibility mapping: $\Phi_{\text{feas}}(G_{\text{LCQP}}) = 1$ if the LCQP problem associated to $G_{\text{LCQP}}$ is feasible and $\Phi_{\text{feas}}(G_{\text{LCQP}}) = 0$ if it is infeasible.*

- *Optimal objective value mapping: $\Phi_{\text{obj}}(G_{\text{LCQP}}) \in \mathbb{R} \cup \{\pm\infty\}$ computes the optimal objective value of the LCQP problem associated to $G_{\text{LCQP}}$. $\Phi_{\text{obj}}(G_{\text{LCQP}}) = +\infty$ means the problem is infeasible and $\Phi_{\text{obj}}(G_{\text{LCQP}}) = -\infty$ means the problem is unbounded.*

- *Optimal solution mapping: For a feasible and bounded LCQP problem (i.e., $\Phi_{\text{obj}}(G_{\text{LCQP}}) \in \mathbb{R}$), an optimal solution exists (Eaves, 1971) though it might not be unique. However, the optimal solution with the smallest $\ell_2$-norm must be unique if $Q \succeq 0$ and we define it as $\Phi_{\text{sol}}(G_{\text{LCQP}})$.*

Given the definitions above, we can formally pose the question in (1.1) as follows: Is there any $F \in \mathcal{F}_{\text{LCQP}}$ that well approximates $\Phi_{\text{feas}}$ or $\Phi_{\text{obj}}$? Similarly, is there any function $F_W \in \mathcal{F}_{\text{LCQP}}^W$ that well approximates $\Phi_{\text{sol}}(G_{\text{LCQP}})$?

## 3 UNIVERSAL APPROXIMATION OF GNNS FOR LCQPS

This section presents our main theoretical results for the expressive power of GNNs for representing properties of LCQPs. In particular, we show that for any LCQP data distribution, there always be a GNN that can predict LCQP properties, in the sense of universally approximating target mappings in Definition 2.3, within given error tolerance. Although it is known in the previous literature that there exists some continuous function that cannot be approximated by GNNs with arbitrarily small error, see e.g., Xu et al. (2019); Azizian & Lelarge (2021); Geerts & Reutter (2022), our results in this section indicate that approximating the target mappings of LCQPs (defined in Definition 2.3) do not suffer from this fundamental limitation. Such results answer the question (1.1) positively.

**Assumption 3.1.** $\mathbb{P}$ *is a Borel regular probability measure on $\mathcal{G}_{\text{LCQP}}^{m,n}$.*

The assumption of Borel regularity is generally satisfied for most data distributions in practice, including discrete distributions, gaussian distributions, etc. With this assumption, we have:

**Theorem 3.2.** *For any probability measure $\mathbb{P}$ satisfying Assumption 3.1 and any $\epsilon > 0$, there exists $F \in \mathcal{F}_{\text{LCQP}}$ such that $\mathbb{I}_{F(G_{\text{LCQP}}) > \frac{1}{2}}$ acts as a classifier for LCQP-feasibility, with an error of up to $\epsilon$:*

---

[1]The space $\mathcal{G}_{\text{LCQP}}^{m,n}$ is equipped with the subspace topology induced from the product space $\left\{(A, b, c, Q, l, u, \circ) : A \in \mathbb{R}^{m \times n}, b \in \mathbb{R}^m, c \in \mathbb{R}^n, Q \in \mathbb{R}^{n \times n}, l \in (\mathbb{R} \cup \{-\infty\})^n, u \in (\mathbb{R} \cup \{+\infty\})^n, \circ \in \{\le, =, \ge\}^m\right\}$, where all Euclidean spaces have standard Eudlidean topologies, discrete spaces $\{-\infty\}, \{+\infty\}$, and $\{\le, =, \ge\}$ have the discrete topologies, and all unions are disjoint unions.

$$\mathbb{P}\left[\mathbb{I}_{F(G_{\text{LCQP}})>\frac{1}{2}} \neq \Phi_{\text{feas}}(G_{\text{LCQP}})\right] < \epsilon,$$

where $\mathbb{I}.$ is the indicator function: $\mathbb{I}_{F(G_{\text{LCQP}})>\frac{1}{2}} = 1$ if $F(G_{\text{LCQP}}) > \frac{1}{2}$; $\mathbb{I}_{F(G_{\text{LCQP}})>\frac{1}{2}} = 0$ otherwise.

This result suggests that a GNN is a universal classifier for LCQP feasibility: for any data distribution of LCQPs satisfying Assumption 3.1, there exists a GNN that can classify LCQP feasibility with arbitrarily high accuracy. This is a natural extension of the feasibility classification for linear programs (Chen et al., 2023a), as feasibility is solely determined by the constraints, independent of the objective function, and all LCQP constraints are linear.

However, using GNNs to predict the optimal objective value or an optimal solution is highly non-trivial due to the nonlinear term $x^\top Q x$. Fortunately, when restricting LCQPs to convex cases, GNNs can universally represent the optimal objective value and an optimal solution for these LCQPs.

**Theorem 3.3.** *Let $\mathbb{P}$ be a probability measure on $\mathcal{G}_{\text{LCQP}}^{m,n}$ satisfying Assumption 3.1 with $\mathbb{P}[Q \succeq 0] = 1$, i.e., $Q$ is positive semidefinite almost surely. For any $\epsilon > 0$, there exists $F_1 \in \mathcal{F}_{\text{LCQP}}$ such that*

$$\mathbb{P}\left[\mathbb{I}_{F_1(G_{\text{LCQP}})>\frac{1}{2}} \neq \mathbb{I}_{\Phi_{\text{obj}}(G_{\text{LCQP}})\in\mathbb{R}}\right] < \epsilon. \tag{3.1}$$

*Addtitionally, if $\mathbb{P}[\Phi_{\text{obj}}(G_{\text{LCQP}}) \in \mathbb{R}] = 1$, then for any $\epsilon, \delta > 0$, there exists $F_2 \in \mathcal{F}_{\text{LCQP}}$ such that*

$$\mathbb{P}\left[|F_2(G_{\text{LCQP}}) - \Phi_{\text{obj}}(G_{\text{LCQP}})| > \delta\right] < \epsilon. \tag{3.2}$$

This theorem indicates that GNNs can approximate the optimal objective value mapping $\Phi_{\text{obj}}$ very well in two senses: (1) GNN can predict whether the optimal objective value is a real number or $\pm\infty$, i.e., whether the LCQP problem is feasible and bounded or not. (2) For a data distribution over feasible and bounded LCQP problems, GNN can approximate the real-valued mapping $\Phi_{\text{obj}}$.

Our last theorem for LCQP is that GNN can approximate the optimal solution map $\Phi_{\text{sol}}$ that returns the optimal solution with the smallest $\ell_2$-norm of feasible and bounded LCQP problems.

**Theorem 3.4.** *Let $\mathbb{P}$ be a probability measure on $\mathcal{G}_{\text{LCQP}}^{m,n}$ satisfying Assumption 3.1 and $\mathbb{P}[Q \succeq 0] = \mathbb{P}[\Phi_{\text{obj}}(G_{\text{LCQP}}) \in \mathbb{R}] = 1$. For any $\epsilon, \delta > 0$, there exists $F_W \in \mathcal{F}_{\text{LCQP}}^W$ such that*

$$\mathbb{P}\left[\|F_W(G_{\text{LCQP}}) - \Phi_{\text{sol}}(G_{\text{LCQP}})\| > \delta\right] < \epsilon.$$

The proofs of Theorems 3.3 and 3.4 will be presented in Appendix A. We briefly describe the main idea here. The Stone-Weierstrass theorem and its variants are a powerful tool for proving universal-approximation-type results. Recall that the classic version of the Stone-Weierstrass theorem states that under some assumptions, a function class $\mathcal{F}$ can uniformly approximate every continuous function if and only if it **separates points**, i.e., for any $x \neq x'$, one has $F(x) \neq F(x')$ for some $F \in \mathcal{F}$. Otherwise, we say $x$ and $x'$ are **indistinguishable** by any $F \in \mathcal{F}$. Therefore, the key component in the proof is to establish some separation results in the sense that two LCQP-graphs with different optimal objective values (or different optimal solutions with the smallest $\ell_2$-norm) must be distinguished by some GNN in the class $\mathcal{F}_{\text{LCQP}}$ (or $\mathcal{F}_{\text{LCQP}}^W$). It is first established in Xu et al. (2019) that the **separation power**[2] of GNNs is equivalent to the Weisfeiler-Lehman (WL) test (Weisfeiler & Leman, 1968), a classical algorithm for the graph isomorphism problem, which is further developed in many recently works, see e.g. Azizian & Lelarge (2021); Geerts & Reutter (2022). We show that, any two LCQP-graphs that are indistinguishable by the WL test, or equivalently by all GNNs, even if they are not isomorphic, some of their structures must be identical, which guarantees that they must have identical optimal objective value and identical optimal solution with the smallest $\ell_2$-norm (see Definition A.1, Theorem A.2, and Theorem A.3).

The universal approximation results of GNNs for LCQPs can be extended to quadratically constrained quadratic programs (QCQPs) that have additional quadratic terms in the constraints compared to LCQPs. Specifically, we modify the graph representation with additional hyperedges to represent the quadratic terms in the constraints, and modify the GNN architecture that updates both vertex features and edge features layer by layer. The details are deferred to Appendix E.

## 4 THE CAPACITY OF GNNS FOR MI-LCQPS

In this section, we discuss the expressive power of GNNs for mixed-integer linearly constrained quadratic programs (MI-LCQPs), for which the general form is almost the same as (2.1) except

---

[2]Given two sets of functions, $\mathcal{F}$ and $\mathcal{F}'$, both defined over the same domain $X$, if $\mathcal{F}$ separating points $x$ and $x'$ implies that $\mathcal{F}'$ also separates $x$ and $x'$ for any $x, x' \in X$, then the separation power of $\mathcal{F}'$ is considered to be stronger than or at least equal to that of $\mathcal{F}$.

that some entries of $x$ are constrained to be integers: $x_j \in \mathbb{Z}, \ \forall \ j \in I$, where $I \subset \{1, 2, \dots, n\}$ collects the indices of all integer variables. Before proceeding, we extend LCQP-graphs and the corresponding GNNs and target mappings to the MI-LCQP setting.

**MI-LCQP-graph** is modified from the LCQP-graph (Section 2 and Figure 1) by adding a new entry to the feature of each variable node $j \in W$. The new feature is $w_j = (c_j, l_j, u_j, \delta_I(j))$ where $\delta_I(j) = 1$ if $j \in I$ and $\delta_I(j) = 0$ otherwise. We use $\mathcal{G}_{\text{MI-LCQP}}^{m,n}$ to denote the collection of all MI-LCQP-graphs with $m$ constraints, $n$ variables, and symmetric and positive semi-definite $Q$.

**GNNs for MI-LCQP-graphs** are constructed following the same mechanism as for LCQP-graphs, with the difference that the message-passing layer is modified as

- $s_i^l = f_l^V \left( s_i^{l-1}, \sum_{j \in \mathcal{N}_i^W} g_l^W(t_j^{l-1}, A_{ij}) \right)$ for $i \in V$, and
- $t_j^l = f_l^W \left( t_j^{l-1}, \sum_{i \in \mathcal{N}_j^V} g_l^V(s_i^{l-1}, A_{ij}), \sum_{j' \in \mathcal{N}_j^W} g_l^Q(t_{j'}^{l-1}, Q_{jj'}) \right)$ for $j \in W$,

where $\mathcal{N}_i^W = \{j \in W : A_{ij} \neq 0\}, \mathcal{N}_j^V = \{j \in V : A_{ij} \neq 0\}$, and $\mathcal{N}_j^W = \{j' \in W : Q_{jj'} \neq 0\}$ are the sets of neighbors. We use $\mathcal{F}_{\text{MI-LCQP}}$ and $\mathcal{F}_{\text{MI-LCQP}}^W$ to denote the GNN classes for MI-LCQP-graphs with graph-level and node-level output, respectively.

**Target mappings** for MI-LCQPs considered in this section are also similar to those in Definition 2.3. In particular, the feasibility mapping $\Phi_{\text{feas}}$ and the optimal objective value mapping $\Phi_{\text{obj}}$ are defined in the same way as in Definition 2.3, while the optimal solution mapping $\Phi_{\text{sol}}$ can only be defined on a subset of the class of feasible and bounded MI-LCQPs, which will be discussed in Appendix C.

## 4.1 GNNs cannot universally represent MI-LCQPs

In this subsection, we answer the question (1.1) for MI-LCQP. When integer variables are introduced, the situation changes. Particularly, we present some counter-examples illustrating the fundamental limitation of GNNs for representing properties of MI-LCQPs.

**Proposition 4.1.** *There exist two MI-LCQP problems, with one being feasible and the other being infeasible, such that their graphs are indistinguishable by any GNN in $\mathcal{F}_{\text{MI-CLQP}}$.*

**Proposition 4.2.** *There exist two feasible MI-LCQP problems, with different optimal objective values, such that their graphs are indistinguishable by any GNN in $\mathcal{F}_{\text{MI-CLQP}}$.*

**Proposition 4.3.** *There exist two feasible MI-LCQP problems with the same optimal objectives but disjoint optimal solution sets, such that their graphs are indistinguishable by any GNN in $\mathcal{F}_{\text{MI-CLQP}}^W$.*

Propositions 4.1, 4.2, and 4.3 indicate that for some MI-LCQP data distribution, it is *impossible* to train a GNN to predict MI-LCQP properties, *regardless of the size or the complexity of the GNN*. Particularly, one can choose the uniform distribution over pairs of instances satisfying Propositions 4.1, 4.2, and 4.3: any GNN making good approximation on one instance must fail on the other.

The detailed proofs of all three propositions are provided in Appendix B. Here we present a pair of MI-LCQP instances that prove Proposition 4.3. This pair is the most interesting among those related to Propositions 4.1, 4.2, and 4.3. Consider the following two MI-LCQPs:

$$\min_{x \in \mathbb{R}^7} \frac{1}{2} x^\top \mathbf{1} \mathbf{1}^\top x + \mathbf{1}^\top x,$$
$$\text{s.t. } x_1 - x_2 = 0, \ x_2 - x_1 = 0,$$
$$x_3 - x_4 = 0, \ x_4 - x_5 = 0,$$
$$x_5 - x_6 = 0, \ x_6 - x_7 = 0, \ x_7 - x_3 = 0,$$
$$x_1 + x_2 + x_3 + x_4 + x_5 + x_6 + x_7 = 6$$
$$0 \le x_j \le 3, \ x_j \in \mathbb{Z}, \ \forall \ j \in \{1, 2, \dots, 7\}.$$



$$\min_{x \in \mathbb{R}^7} \frac{1}{2} x^\top \mathbf{1} \mathbf{1}^\top x + \mathbf{1}^\top x,$$
$$\text{s.t. } x_1 - x_2 = 0, \ x_2 - x_3 = 0, \ x_3 - x_1 = 0,$$
$$x_4 - x_5 = 0, \ x_5 - x_6 = 0,$$
$$x_6 - x_7 = 0, \ x_7 - x_4 = 0,$$
$$x_1 + x_2 + x_3 + x_4 + x_5 + x_6 + x_7 = 6$$
$$0 \le x_j \le 3, \ x_j \in \mathbb{Z}, \ \forall \ j \in \{1, 2, \dots, 7\}.$$

Firstly, both MI-LCQPs are feasible and share the same optimal objective value, but their optimal solutions differ. In the first instance, the unique feasible (and thus optimal) solution is $(3, 3, 0, 0, 0, 0, 0)$, while in the second instance, it is $(2, 2, 2, 0, 0, 0, 0)$. In both instances, the optimal objective values are identical, as $\mathbf{1}^\top x = 6$ leads to $\frac{1}{2}x^\top \mathbf{1}\mathbf{1}^\top x + \mathbf{1}^\top x = 24$.

Secondly, the two instances cannot be distinguished by any GNN in $\mathcal{F}^W_{\text{MI-CLQP}}$. Initially, each variable node $w_j$ is assigned the same attribute, $w_j = (1, 0, 3, 1)$, which represents an objective coefficient of $c_j = 1$, lower bound $l_j = 0$, upper bound $u_j = 3$, and an integral indicator $\delta_I(j) = 1$. These concepts are detailed in Section 2 and the beginning of Section 4. We refer to these nodes as "red nodes". Similarly, the first seven constraint nodes $v_i$ (for $1 \leq i \leq 7$) are assigned the same attribute, $v_i = (0, =)$, which we label as "blue nodes". The eighth constraint node $v_8$ is unique, with the attribute $v_8 = (6, =)$, and is called the "brown node". Based solely on node information, the two graphs are indistinguishable since both have seven red nodes, seven blue nodes, and one brown node.

Even after multiple rounds of message passing (as described in Section 2), the two graphs remain indistinguishable. To explain, consider any red node $w_j$, which is connected to a blue node with weight $A_{ij} = 1$ (solid lines), another blue node with weight $A_{ij} = -1$ (dashed lines), the brown node with weight $A_{ij} = 1$ (green lines), and all seven red nodes with weights $Q_{jj'} = 1$ (brown curves). Thus, the red node's attribute is updated as follows (an informal but illustrative equation):

$$t_j^l = f_l^w \left( \text{red node}, g_l^V(\text{blue node}) - g_l^V(\text{blue node}) + g_l^V(\text{brown node}), 7g_l^Q(\text{red node}) \right).$$

After the update, all red nodes $t_j^l (1 \leq j \leq 7)$ in both graphs retain identical attributes and are still indistinguishable. The same applies to the blue and brown nodes, leading to the conclusion that, regardless of how many message-passing rounds occur, both graphs will still have seven red nodes, seven blue nodes, and one brown node. This conclusion holds for any parameterized mappings used in GNNs ($f_l^V$, $f_l^W$, $g_l^V$, $g_l^W$, and $g_l^Q$), meaning no GNN can differentiate between the two instances. This illustrates a limitation of GNNs in representing MI-LCQP, which is ignored in the literature.

### 4.2 GNNs CAN REPRESENT PARTICULAR TYPES OF MI-LCQPs

We have shown a fundamental limitation of GNNs to represent properties of general MI-LCQP problems. Therefore, a natural question is: *Whether we can identify a subset of $\mathcal{G}_{\text{MI-LCQP}}$ on which it is possible to train reliable GNNs.* To address this, we need to gain a better understanding for the separation power of GNNs or equivalently the WL test, according to the discussion following Theorem 3.4. We state in Algorithm 1 the WL test for MI-LCQP-graphs associated to $\mathcal{F}_{\text{MI-LCQP}}$ or $\mathcal{F}^W_{\text{MI-LCQP}}$, where $C_i^{l,V}$ and $C_j^{l,W}$ are understood as the color of $i \in V$ and $j \in W$ at the $l$-th iteration.

---

**Algorithm 1** The WL test for MI-LCQP-graphs (Example provided in Appendix D)

---

**Require:** A LCQP-graph $G = (V, W, A, Q, H_V, H_W)$ and iteration limit $L > 0$.
1: Initialize with $C_i^{0,V} = \text{HASH}(v_i)$ and $C_j^{0,W} = \text{HASH}(w_j)$.
2: **for** $l = 1, 2, \cdots, L$ **do**
3:     $C_i^{l,V} = \text{HASH}\big(C_i^{l-1,V}, \{\{(C_j^{l-1,W}, A_{ij}) : j \in \mathcal{N}_i^W\}\}\big)$.
4:     $C_j^{l,W} = \text{HASH}\big(C_j^{l-1,W}, \{\{(C_i^{l-1,V}, A_{ij}) : i \in \mathcal{N}_j^V\}\}, \{\{(C_{j'}^{l-1,W}, Q_{jj'}) : j' \in \mathcal{N}_j^W\}\}\big)$.
5: **end for**
6: **return** The multisets containing all colors $\big\{\{C_i^{L,V}\}\big\}_{i=0}^m, \big\{\{C_j^{L,W}\}\big\}_{j=0}^n$.

---

Initially, each vertex is labeled a color according to its attributes ($v_i$ or $w_j$). In the case that the hash functions introduce no collisions, two vertices are of the same color at the $l$-th iteration if and only if at the $(l-1)$-th iteration, they have the same color and the same information aggregation from neighbors in terms of multiset of colors and edge weights. This is a *color refinement* procedure. One can have a ***partition*** of the vertex set $V \cup W$ at each iteration based on vertices' colors: two vertices are classified in the same class if and only if they are of the same color. Such a partition is strictly refined in the first $\mathcal{O}(m + n)$ iterations and will ***remain stable or unchanged*** afterward if no collision, see e.g. Berkholz et al. (2017).

Intuitively, vertices in the same class of the final stable partition generated by the WL test will always have identical attributes in message-passing layers for all GNNs in $\mathcal{F}_{\text{MI-LCQP}}$ or $\mathcal{F}^W_{\text{MI-LCQP}}$, and vice versa, since the color refinement procedure in Algorithm 1 follows the same mechanism as the message-passing process. Thus, to identify a subset of $\mathcal{F}_{\text{MI-LCQP}}$ on which GNNs have sufficiently strong separation power, we propose the following definition generalized from Chen et al. (2024)

for mixed-integer linear programs (MILPs), which basically states that vertices in the same class generated by the WL test can indeed be treated same in some sense.

**Definition 4.4** (MP-tractable MI-LCQP)**.** *Let $G_{MI\text{-}LCQP} \in \mathcal{G}_{\text{MI-LCQP}}^{m,n}$ be a MI-LCQP problem and let $(\mathcal{I}, \mathcal{J})$ be the final stable partition of $V \cup W$ generated by WL test without collision, where $\mathcal{I} = \{I_1, I_2, \dots, I_s\}$ is a partition of $V = \{1, 2, \dots, m\}$ and $\mathcal{J} = \{J_1, J_2, \dots, J_t\}$ is a partition of $W = \{1, 2, \dots, n\}$. We say that $G_{MI\text{-}LCQP}$ is message-passing-tractable (MP-tractable) if:*

*(a) For any $p \in \{1, 2, \dots, s\}$ and $q \in \{1, 2, \dots, t\}$, $A_{ij}$ is constant in $i \in I_p, j \in J_q$.*

*(b) For any $q, q' \in \{1, 2, \dots, t\}$, $Q_{jj'}$ is constant in $j \in J_q, j' \in J_{q'}$.*

*We use $\mathcal{G}_{\text{MP}}^{m,n} \subset \mathcal{G}_{\text{MI-LCQP}}^{m,n}$ to denote the collection of all MP-tractable MI-LCQP-graphs.*

Under the assumption of MP-tractability, we can establish universal approximation results for GNNs on MI-LCQPs regarding feasibility and optimal objective value. *While GNNs cannot universally represent all MI-LCQPs, they can represent MP-tractable ones.*

**Assumption 4.5.** $\mathbb{P}$ *is a Borel regular probability measure on $\mathcal{G}_{\text{MI-LCQP}}^{m,n}$* [3].

**Theorem 4.6.** *Let $\mathbb{P}$ be a probability measure satisfying Assumption 4.5 and $\mathbb{P}[G_{MI\text{-}LCQP} \in \mathcal{G}_{\text{MP}}^{m,n}] = 1$, i.e., the MP-tractability holds almost surely. For any $\epsilon > 0$, there exists $F \in \mathcal{F}_{\text{MI-LCQP}}$ such that*

$$\mathbb{P}\left[\mathbb{I}_{F(G_{MI\text{-}LCQP}) > \frac{1}{2}} \neq \Phi_{\text{feas}}(G_{MI\text{-}LCQP})\right] < \epsilon.$$

**Theorem 4.7.** *Let $\mathbb{P}$ be a probability measure satisfying Assumption 4.5 and $\mathbb{P}[G_{MI\text{-}LCQP} \in \mathcal{G}_{\text{MP}}^{m,n}] = 1$, i.e., the MP-tractability holds almost surely. For any $\epsilon > 0$, there exists $F_1 \in \mathcal{F}_{\text{MI-LCQP}}$ such that*

$$\mathbb{P}\left[\mathbb{I}_{F_1(G_{MI\text{-}LCQP}) > \frac{1}{2}} \neq \mathbb{I}_{\Phi_{\text{obj}}(G_{MI\text{-}LCQP}) \in \mathbb{R}}\right] < \epsilon.$$

*Additionally, if $\mathbb{P}[\Phi_{\text{obj}}(G_{MI\text{-}LCQP}) \in \mathbb{R}] = 1$, for any $\epsilon, \delta > 0$, there exists $F_2 \in \mathcal{F}_{\text{MI-LCQP}}$ such that*

$$\mathbb{P}\left[|F_2(G_{MI\text{-}LCQP}) - \Phi_{\text{obj}}(G_{MI\text{-}LCQP})| > \delta\right] < \epsilon.$$

To extend these results to predicting optimal solutions with GNNs, we introduce two additional assumptions. First, we assume the MI-LCQPs have an optimal solution. We define $\mathcal{G}_{\text{sol}}^{m,n}$ as the set of MI-LCQPs for which an optimal solution exists. The assumption is expressed as $G_{MI\text{-}LCQP} \in \mathcal{G}_{\text{sol}}^{m,n}$. The second assumption is that MI-LCQPs are unfoldable, defined below in Definition 4.8, extending the concept from Chen et al. (2023b) for MILPs.

**Definition 4.8** (Unfoldable MI-LCQP)**.** *In the same setting as in Definition 4.4, we say that $G_{MI\text{-}LCQP}$ is unfoldable if $t = n$ and $|J_1| = |J_2| = \cdots = |J_n| = 1$, i.e., all vertices in $W$ have different colors. We use $\mathcal{G}_{\text{unfold}}^{m,n} \subset \mathcal{G}_{\text{MI-LCQP}}^{m,n}$ to denote the collection of all unfoldable MI-LCQP-graphs.*

With the two assumptions—that the MI-LCQPs have an optimal solution and are unfoldable—we can establish a universal approximation result for optimal solution prediction: GNNs can universally approximate the optimal solutions for this specific class of MI-LCQPs.

**Theorem 4.9.** *Let $\mathbb{P}$ be a probability measure on $\mathcal{G}_{\text{MI-LCQP}}^{m,n}$ satisfying Assumption 4.5 and $\mathbb{P}[G_{MI\text{-}LCQP} \in \mathcal{G}_{\text{sol}}^{m,n} \cap \mathcal{G}_{\text{unfold}}^{m,n}] = 1$. For any $\epsilon, \delta > 0$, there exists $F_W \in \mathcal{F}_{\text{MI-LCQP}}^W$ such that*

$$\mathbb{P}\left[\|F_W(G_{MI\text{-}LCQP}) - \Phi_{\text{sol}}(G_{MI\text{-}LCQP})\| > \delta\right] < \epsilon.$$

Theorems 4.6, 4.7, and 4.9 precisely characterize the subsets of MI-LCQPs where GNNs can succeed and their proofs can be found in Appendix C.

## 4.3 Practical characterization of "solvable" MI-LCQPs

To better illustrate the practical implications of Theorems 4.6, 4.7, and 4.9, we make more discussion of MP-tractability and unfoldability in this subsection.

**MP-tractability vs unfoldability.** While all unfoldable MI-LCQPs must be MP-tractable (strictly proved in the appendix), not all MP-tractable problems are necessarily unfoldable. This difference can be clearly illustrated with an example that is MP-tractable but not unfoldable:

$$\min \frac{1}{2}x_2^2 + x_1 + x_2 + x_3, \text{ s.t. } x_1 + x_3 \leq 1, \ x_1 - x_2 + x_3 \leq 1, \ 0 \leq x_1, x_2, x_3 \leq 1, \ x_1, x_2, x_3 \in \mathbb{Z}.$$

The related discussions, proofs, and this example are further detailed in Appendix D.

---

[3]The topology of $\mathcal{G}_{\text{MI-LCQP}}^{m,n}$ is defined in the same way as $\mathcal{G}_{\text{LCQP}}^{m,n}$.

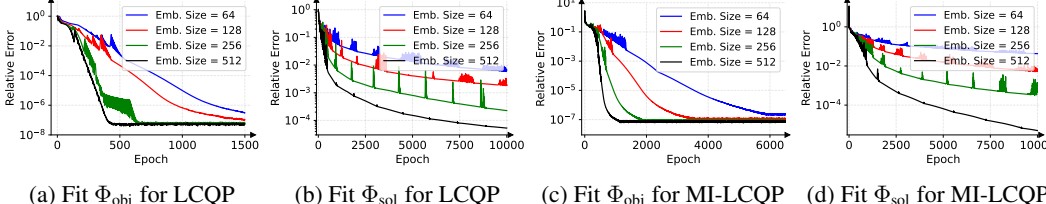

| (a) Fit $\Phi_{\text{obj}}$ for LCQP | (b) Fit $\Phi_{\text{sol}}$ for LCQP | (c) Fit $\Phi_{\text{obj}}$ for MI-LCQP | (d) Fit $\Phi_{\text{sol}}$ for MI-LCQP |
|---|---|---|---|

Figure 2: Relative errors when training GNNs to fit $\Phi_{\text{obj}}$ and $\Phi_{\text{sol}}$ for LCQP (2a-2b) and MI-LCQP (2c-2d). GNNs are trained on 100 randomly generated problem instances.

**Numerical verification of MP-tractability and unfoldability.** In practice, both MP-tractability and unfoldability can be efficiently verified. In particular, one can apply the WL test, which requires at most $\mathcal{O}(m+n)$ iterations. The complexity of each iteration is bounded by the number of edges in the graph Shervashidze et al. (2011), which, in our context, is the number of nonzeros in matrices $A$ and $Q$: $\text{nnz}(A)+\text{nnz}(Q)$. Therefore, the overall complexity of Algorithm 1 is $\mathcal{O}((m+n)\cdot(\text{nnz}(A)+\text{nnz}(Q)))$. After running Algorithm 1, MP-tractability can be directly verified using Definition 4.4, and unfoldability can be directly verified using Definition 4.8.

**Frequency of MP-tractability and unfoldability.** In practice, the frequency of MP-tractable and unfoldable instances largely depends on the dataset. In the earlier example, two of three variables, $x_1$ and $x_3$, display symmetry — they are labeled with the same color by WL test and swapping them does not alter the problem. Generally, unfoldable problems lack symmetry and MP-tractability allows for some degree of symmetry. Another example in Section 4.1 admits strong symmetry across all variables, making it neither MP-tractable nor unfoldable. Thus, the frequency of MP-tractability and unfoldability relates to ***the level of symmetry*** in the data. When there is symmetry in MI-LCQP, it becomes foldable; and higher symmetry increases the risk of being MP-intractable. Fortunately, *unfoldable and MP-tractable instances make up the majority of the MI-LCQP set* (shown in Appendix D). The dataset used in our experiments, which includes synthetic MI-LCQPs, portfolio problems, and SVMs, consists entirely of unfoldable and MP-tractable instances. However, it's important to note that in some challenging, artificially created datasets like MIPLIB 2017 Gleixner et al. (2021), about 1/4 of the examples exhibit significant symmetry in half of the variables.

**How to handle bad instances?** Two potential approaches to deal with symmetry. (I) Adding features: Introducing additional features can differentiate nodes in symmetric graphs. For example, adding a random feature to nodes with identical attributes ensures they are no longer symmetric Sato et al. (2021). (II) Using higher-order GNNs: These models can distinguish nodes that standard message-passing GNNs cannot, enhancing their expressive power Morris et al. (2019).

## 5 NUMERICAL EXPERIMENTS

**Numerical validation of GNNs' expressive power.** We train GNNs to fit $\Phi_{\text{obj}}$ or $\Phi_{\text{sol}}$ for LCQP or MI-LCQP instances.[4] For both LCQP and MI-LCQP, we randomly generate 100 instances, each of which contains 10 constraints and 50 variables. The generated MI-LCQPs are all unfoldable and MP-tractable with probability one. The optimal solutions and corresponding objective function values are collected using existing solvers. Details on the data generation and training schemes can be found in Appendix F. We train four GNNs with four different embedding sizes and record their relative errors averaged on all instances during training.[5] The results are reported in Figure 2. We can see that GNNs can fit $\Phi_{\text{obj}}$ and $\Phi_{\text{sol}}$ well for both LCQP and MI-LCQP. These results validate Theorems 3.3, 3.4, 4.7 and 4.9 on a small set of instances. We also observe that a larger embedding size increases the capacity of a GNN, resulting in not only lower final errors but also faster convergence.

**Numerical validation on a larger scale.** To further validate the theorems, we expand the number of problem instances to 500 and 2,500, and conduct training on the four GNNs along with a larger variant with an embedding size of 1,024. The results are reported in Figure 3. We can observe that GNN can achieve near-zero fitting errors as long as it has a large enough embedding size and thus enough capacity for approximation, which directly validate Theorems 3.3, 3.4, 4.7 and 4.9.

---

[4]Since LCQP and MI-LCQP are linearly constrained, predicting feasibility falls to the case of LP and MILP, which has been numerically investigated in Chen et al. (2023a;b). Hence we omit the feasibility experiments.

[5]The relative error of a GNN $F_W$ on a single problem instance $G$ is defined as $\|F_W(G) - \Phi(G)\|_2 / \max(\|\Phi(G)\|_2, 1)$, where $\Phi$ could be either $\Phi_{\text{obj}}$ or $\Phi_{\text{sol}}$.

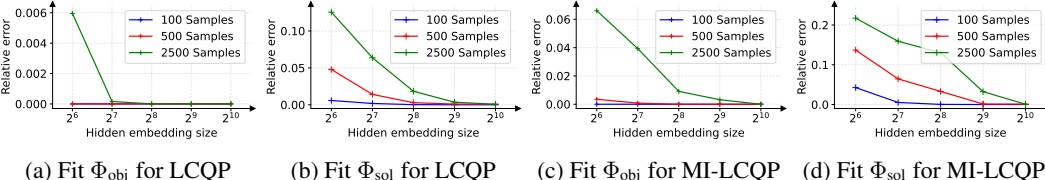

(a) Fit $\Phi_{obj}$ for LCQP     (b) Fit $\Phi_{sol}$ for LCQP     (c) Fit $\Phi_{obj}$ for MI-LCQP     (d) Fit $\Phi_{sol}$ for MI-LCQP

Figure 3: Empirical results on randomly generated generic LCQP and MI-LCQP problems as formulated in (2.1) and (C.1). The figures illustrate the relative errors achieved during training for various combinations of embedding sizes and numbers of training samples. We can achieve near zero errors when the GNN is large enough.

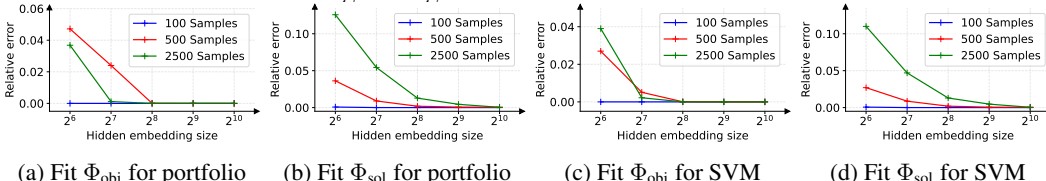

(a) Fit $\Phi_{obj}$ for portfolio     (b) Fit $\Phi_{sol}$ for portfolio     (c) Fit $\Phi_{obj}$ for SVM     (d) Fit $\Phi_{sol}$ for SVM

Figure 4: Empirical results on randomly generated portfolio optimization and SVM optimization problems (see Appendix F for formulation). The figures illustrate the best relative errors achieved during training for various combinations of embedding sizes and numbers of training samples. We can achieve near-zero errors when the GNN is large enough.

**Various types of LCQP.** Besides the generic LCQP formulation (2.1), we also extend the numerical experiments to other types of optimization problems, namely portfolio optimization and support vector machine (SVM) following Jung et al. (2022). The results of fitting solutions or objective values on 100/500/2,500 randomly generated problem instances are illustrated in Figure 4. We can observe similar fitting behaviors as those in the generic LCQP experiments where the expressive power of GNNs increase as they become larger, evidenced by the fitting errors decreasing to near zero when the embedding size increases. The formulation of the portfolio and SVM optimization and how the problem instances are generated are explained in Appendix F.

**Generalization.** Besides investigating GNNs' expressive capacity, we also explore their generalization ability and observed positive results. However, since the generalization ability is out of the main topic of this work, we refer the interested readers to Appendix F for details.

**Analysis of GNN computation complexity.** GNNs are superior over QP solvers in terms of running time, especially when we fully exploit parallel computing with GPU acceleration. To show this, we measure the average running time using OSQP (Stellato et al., 2020) and a trained GNN with different batch sizes over the 1,000 synthetic LCQP problems generated in the experiment above. We applied OSQP to solve all instances to a relative error of $10^{-3}$, which is slightly less accurate than the trained GNN (with an average relative error of $6.31 \times 10^{-4}$). All running times were measured in milliseconds. The results are shown in the Table 1. The sufficiently acceler-

Table 1: Average solving times of GNN and OSQP on 1,000 LCQP instances.

| Method | Batch Size | Solving Time (ms) |
|--------|------------|-------------------|
| OSQP   | -          | 2.44              |
| GNN    | 1          | 47.56             |
|        | 10         | 6.13              |
|        | 100        | 0.79              |
|        | 1,000      | 0.41              |

ated computation validates GNNs' capacity as a real-time QP solver or fast warm-start, numerically supporting the rationality of our theoretical study of GNNs for QPs.

## 6 CONCLUSION

This paper establishes theoretical foundations for using GNNs to represent the feasibility, optimal objective value, and optimal solution, of LCQPs and MI-LCQPs. In particular, we prove the existence of GNNs that can predict those properties of LCQPs universally well and show with explicit examples that such results are generally not true for MI-LCQPs when integer constraints are introduced. Moreover, we precisely identify subclasses of MI-LCQP problems on which such universal approximation results are still valid. All our findings are also verified numerically. However, our universal approximation theorems only show the existence of the GNNs, without discussing the training, generalization, and the size of GNNs, which are important future directions.

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

## A    Proofs for Section 3

In this appendix, we present the proofs for theorems in Section 3. The proofs will based on Weisfeiler-Lehman (WL) test and its separation power to distinguish LCQP problems with different properties.

The Weisfeiler-Lehman (WL) test (Weisfeiler & Leman, 1968) is a classical algorithm for the graph isomorphism problem. In particular, it implements color refinement on vertices by applying a hash function on the previous vertex color and aggregation of colors from neighbors, and identifies two graphs as isomorphic if their final color multisets are the same. It is worth noting that WL test may incorrectly identify two non-isomorphic graphs as isomorphic. We slightly modify the standard WL test to fit the structure of LCQP-graphs, see Algorithm 2.

We define two equivalence relations as follows. Intuitively, LCQP-graphs in the same equivalence class will be identified as isomorphic by WL test, though they may be actually non-isomorphic.

**Definition A.1.** *For two LCQP-graphs* $G_{\text{LCQP}}, \hat{G}_{\text{LCQP}} \in \mathcal{G}_{\text{LCQP}}^{m,n}$, *let* $\{\{C_i^{L,V}\}\}_{i=0}^m, \{\{C_j^{L,W}\}\}_{j=0}^n$ *and* $\{\{\hat{C}_i^{L,V}\}\}_{i=0}^m, \{\{\hat{C}_j^{L,W}\}\}_{j=0}^n$ *be color multisets output by Algorithm 2 on* $G_{\text{LCQP}}$ *and* $\hat{G}_{\text{LCQP}}$.

> *1. We say* $G_{\text{LCQP}} \sim \hat{G}_{\text{LCQP}}$ *if* $\{\{C_i^{L,V}\}\}_{i=0}^m = \{\{\hat{C}_i^{L,V}\}\}_{i=0}^m$ *and* $\{\{C_j^{L,W}\}\}_{j=0}^n = \{\{\hat{C}_j^{L,W}\}\}_{j=0}^n$ *hold for all* $L \in \mathbb{N}$ *and all hash functions.*

---

**Algorithm 2** The WL test for LCQP-graphs

---

**Require:** A LCQP-graph $G = (V, W, A, Q, H_V, H_W)$ and iteration limit $L > 0$.

1: Initialize with $C_i^{0,V} = \text{HASH}(v_i)$ and $C_j^{0,W} = \text{HASH}(w_j)$.

2: **for** $l = 1, 2, \cdots, L$ **do**

3:    Refine the colors

$$C_i^{l,V} = \text{HASH}\left(C_i^{l-1,V}, \sum_{j=1}^{n} A_{ij}\text{HASH}\left(C_j^{l-1,W}\right)\right),$$

$$C_j^{l,W} = \text{HASH}\left(C_j^{l-1,W}, \sum_{i=1}^{m} A_{ij}\text{HASH}\left(C_i^{l-1,V}\right), \sum_{j'=1}^{n} Q_{jj'}\text{HASH}\left(C_{j'}^{l-1,W}\right)\right).$$

4: **end for**

5: **return** The multisets containing all colors $\left\{\left\{C_i^{L,V}\right\}\right\}_{i=0}^{m}, \left\{\left\{C_j^{L,W}\right\}\right\}_{j=0}^{n}$.

---

   2. We say $G_{\text{LCQP}} \overset{W}{\sim} \hat{G}_{\text{LCQP}}$ if $\{\{C_i^{L,V}\}\}_{i=0}^{m} = \{\{\hat{C}_i^{L,V}\}\}_{i=0}^{m}$ and $C_j^{L,W} = \hat{C}_j^{L,W}$, $\forall\, j \in \{1, 2, \ldots, n\}$, for all $L \in \mathbb{N}$ and all hash functions.

Our main finding leading to the results in Section 3 is that, for LCQP-graphs in the same equivalence class, even if they are non-isomorphic, their optimal objective values and optimal solutions must be the same (up to a permutation perhaps).

**Theorem A.2.** *For any $G_{\text{LCQP}}, \hat{G}_{\text{LCQP}} \in \mathcal{G}_{\text{LCQP}}^{m,n}$ with $Q, \hat{Q} \succeq 0$, if $G_{\text{LCQP}} \sim \hat{G}_{\text{LCQP}}$, then $\Phi_{\text{obj}}(G_{\text{LCQP}}) = \Phi_{\text{obj}}(\hat{G}_{\text{LCQP}})$.*

**Theorem A.3.** *For any $G_{\text{LCQP}}, \hat{G}_{\text{LCQP}} \in \mathcal{G}_{\text{LCQP}}^{m,n}$ with $Q, \hat{Q} \succeq 0$ that are feasible and bounded, if $G_{\text{LCQP}} \sim \hat{G}_{\text{LCQP}}$, then there exists some permutation $\sigma_W \in S_n$ such that $\Phi_{\text{sol}}(G_{\text{LCQP}}) = \sigma_W(\Phi_{\text{sol}}(\hat{G}_{\text{LCQP}}))$. Furthermore, if $G_{\text{LCQP}} \overset{W}{\sim} \hat{G}_{\text{LCQP}}$, then $\Phi_{\text{sol}}(G_{\text{LCQP}}) = \Phi_{\text{sol}}(\hat{G}_{\text{LCQP}})$.*

We need the following lemma to prove Theorem A.2 and Theorem A.3.

**Lemma A.4.** *Suppose that $M \in \mathbb{R}^{n \times n}$ is a symmetric and positive semidefinite matrix and that $\mathcal{J} = \{J_1, J_2, \ldots, J_t\}$ is a partition of $\{1, 2, \ldots, n\}$ satisfying that for any $q, q' \in \{1, 2, \ldots, t\}$, $\sum_{j' \in J_{q'}} M_{jj'}$ is a constant over $j \in J_q$. For any $x \in \mathbb{R}^n$, it holds that*

$$\frac{1}{2}x^\top M x \geq \frac{1}{2}\hat{x}^\top M \hat{x}, \tag{A.1}$$

*where $\hat{x} \in \mathbb{R}^n$ is defined via $\hat{x}_j = y_q = \frac{1}{|J_q|}\sum_{j' \in J_q} x_{j'}$ for $j \in J_q$.*

*Proof.* Fixe $x \in \mathbb{R}^n$ and consider the problem

$$\min_{z \in \mathbb{R}^n}\ \frac{1}{2}z^\top M z, \quad \text{s.t.}\ \sum_{j \in J_q} z_j = \sum_{j \in J_q} x_j,\ q = 1, 2, \ldots, t, \tag{A.2}$$

which is a convex program. The Lagrangian is given by

$$\mathcal{L}(z, \lambda) = \frac{1}{2}z^\top M z - \sum_{q=1}^{t} \lambda_q\left(\sum_{j \in J_q} z_j - \sum_{j \in J_q} x_j\right).$$

It can be computed that

$$\frac{\partial}{\partial z_j}\mathcal{L}(z, \lambda) = \sum_{j'=1}^{n} M_{jj'}z_{j'} - \lambda_q, \quad j \in J_q,$$

and

$$\frac{\partial}{\partial \lambda_q} \mathcal{L}(z, \lambda) = \sum_{j \in J_q} x_j - \sum_{j \in J_q} z_j,$$

It is clear that

$$\frac{\partial}{\partial \lambda_q} \mathcal{L}(\hat{x}, \lambda) = \sum_{j \in J_q} x_j - \sum_{j \in J_q} \hat{x}_j = 0,$$

by the definition of $\hat{x}$. Furthermore, consider any fixed $q \in \{1, 2, \ldots, t\}$ and we have for any $j \in J_q$ that

$$\frac{\partial}{\partial z_j} \mathcal{L}(\hat{x}, \lambda) = \sum_{q'=1}^{t} y_{q'} \sum_{j' \in J_{q'}} M_{jj'} - \lambda_q = 0,$$

if $\lambda_q = \sum_{q'=1}^{t} y_{q'} \sum_{j' \in J_{q'}} M_{jj'}$ that is independent in $j \in q$ since $\sum_{j' \in J_{q'}} M_{jj'}$ is constant over $j \in J_q$ for any $q' \in \{1, 2, \ldots, t\}$. Since the problem (A.2) is convex and the first-order optimality condition is satisfied at $\hat{x}$, we can conclude that $\hat{x}$ is a minimizer of (A.2), which implies (A.1). $\square$

*Proof of Theorem A.2.* Let $G_{\text{LCQP}}$ and $\hat{G}_{\text{LCQP}}$ be the LCQP-graphs associated to (2.1) and

$$\min_{x \in \mathbb{R}^n} \frac{1}{2} x^\top \hat{Q} x + \hat{c}^\top x, \quad \text{s.t. } \hat{A} x \hat{\circ} \hat{b}, \ \hat{l} \le x \le \hat{u}, \tag{A.3}$$

Suppose that there are no collisions of hash functions or their linear combinations when applying the WL test to $G_{\text{LCQP}}$ and $\hat{G}_{\text{LCQP}}$ and there are no strict color refinements in the $L$-th iteration. Since $G_{\text{LCQP}} \sim \hat{G}_{\text{LCQP}}$, after performing some permutation, there exist $\mathcal{I} = \{I_1, I_2, \ldots, I_s\}$ and $\mathcal{J} = \{J_1, J_2, \ldots, J_t\}$ that are partitions of $\{1, 2, \ldots, m\}$ and $\{1, 2, \ldots, n\}$, respectively, such that the followings hold:

- $C_i^{L,V} = C_{i'}^{L,V}$ if and only if $i, i' \in I_p$ for some $p \in \{1, 2, \ldots, s\}$.

- $C_i^{L,V} = \hat{C}_{i'}^{L,V}$ if and only if $i, i' \in I_p$ for some $p \in \{1, 2, \ldots, s\}$.

- $\hat{C}_i^{L,V} = \hat{C}_{i'}^{L,V}$ if and only if $i, i' \in I_p$ for some $p \in \{1, 2, \ldots, s\}$.

- $C_j^{L,W} = C_{j'}^{L,W}$ if and only if $j, j' \in J_q$ for some $q \in \{1, 2, \ldots, t\}$.

- $C_j^{L,W} = \hat{C}_{j'}^{L,W}$ if and only if $j, j' \in J_q$ for some $q \in \{1, 2, \ldots, t\}$.

- $\hat{C}_j^{L,W} = \hat{C}_{j'}^{L,W}$ if and only if $j, j' \in J_q$ for some $q \in \{1, 2, \ldots, t\}$.

Since there are no collisions, we have from the vertex color initialization that

- $v_i = (b_i, \circ_i) = \hat{v}_i = (\hat{b}_i, \hat{\circ}_i)$ and is constant over $i \in I_p$ for any $p \in \{1, 2, \ldots, s\}$.

- $w_j = (c_j, l_j, u_j) = \hat{w}_j = (\hat{c}_j, \hat{l}_j, \hat{u}_j)$ and is constant over $j \in J_q$ for any $q \in \{1, 2, \ldots, t\}$.

For any $p \in \{1, 2, \ldots, s\}$ and any $i, i' \in I_p$, one has

$$C_i^{L,V} = C_{i'}^{L,V} \implies \sum_{j \in W} A_{ij} \text{HASH}\left(C_j^{L-1,W}\right) = \sum_{j \in W} A_{i'j} \text{HASH}\left(C_j^{L-1,W}\right)$$

$$\implies \sum_{j \in W} A_{ij} \text{HASH}\left(C_j^{L,W}\right) = \sum_{j \in W} A_{i'j} \text{HASH}\left(C_j^{L,W}\right)$$

$$\implies \sum_{j \in J_q} A_{ij} = \sum_{j \in J_q} A_{i'j}, \quad \forall q \in \{1, 2, \ldots, t\}.$$

One can obtain similar conclusions from $C_i^{L,V} = \hat{C}_{i'}^{L,V}$ and $\hat{C}_i^{L,V} = \hat{C}_{i'}^{L,V}$, and hence conclude that

- For any $p \in \{1, 2, \ldots, s\}$ and $q \in \{1, 2, \ldots, t\}$, $\sum_{j \in J_q} A_{ij} = \sum_{j \in J_q} \hat{A}_{ij}$ and is constant over $i \in I_p$.

Similarly, the followings also hold:

- For any $p \in \{1, 2, \ldots, s\}$ and $q \in \{1, 2, \ldots, t\}$, $\sum_{i \in I_p} A_{ij} = \sum_{i \in I_p} \hat{A}_{ij}$ and is constant over $j \in J_q$.

- For any $q, q' \in \{1, 2, \ldots, t\}$, $\sum_{j' \in J_{q'}} Q_{jj'} = \sum_{j' \in J_{q'}} \hat{Q}_{jj'}$ and is constant over $j \in J_q$.

If $G_{\text{LCQP}}$ or (2.1) is infeasible, then $\Phi_{\text{obj}}(G_{\text{LCQP}}) = +\infty$ and clearly $\Phi_{\text{obj}}(G_{\text{LCQP}}) \geq \Phi_{\text{obj}}(\hat{G}_{\text{LCQP}})$. If (2.1) is feasible, let $x \in \mathbb{R}^n$ be any feasible solution to (2.1) and define $\hat{x} \in \mathbb{R}^n$ via $\hat{x}_j = y_q = \frac{1}{|J_q|} \sum_{j' \in J_q} x_{j'}$ for $j \in J_q$. By the proofs of Lemma B.2 and Lemma B.3 in Chen et al. (2023a), we know that $\hat{x}$ is a feasible solution to (A.3) and $c^\top x = \hat{c}^\top \hat{x}$. In addition, we have

$$\frac{1}{2} x^\top Q x \overset{(A.1)}{\geq} \frac{1}{2} \hat{x}^\top Q \hat{x} = \frac{1}{2} \sum_{q,q'=1}^{t} \sum_{j \in J_q} \sum_{j' \in J_{q'}} \hat{x}_j Q_{jj'} \hat{x}_{j'} = \frac{1}{2} \sum_{q,q'=1}^{t} y_q y_{q'} \sum_{j' \in J_{q'}} Q_{jj'}$$

$$= \frac{1}{2} \sum_{q,q'=1}^{t} y_q y_{q'} \sum_{j' \in J_{q'}} \hat{Q}_{jj'} = \frac{1}{2} \sum_{q,q'=1}^{t} \sum_{j \in J_q} \sum_{j' \in J_{q'}} \hat{x}_j \hat{Q}_{jj'} \hat{x}_{j'} = \frac{1}{2} \hat{x}^\top \hat{Q} \hat{x},$$

which then implies that

$$\frac{1}{2} x^\top Q x + c^\top x \geq \frac{1}{2} \hat{x}^\top \hat{Q} \hat{x} + \hat{c}^\top \hat{x},$$

and hence that $\Phi_{\text{obj}}(G_{\text{LCQP}}) \geq \Phi_{\text{obj}}(\hat{G}_{\text{LCQP}})$. Till now we have proved $\Phi_{\text{obj}}(G_{\text{LCQP}}) \geq \Phi_{\text{obj}}(\hat{G}_{\text{LCQP}})$ regardless of the feasibility of $G_{\text{LCQP}}$. The reverse direction $\Phi_{\text{obj}}(G_{\text{LCQP}}) \leq \Phi_{\text{obj}}(\hat{G}_{\text{LCQP}})$ is also true and we can conclude that $\Phi_{\text{obj}}(G_{\text{LCQP}}) = \Phi_{\text{obj}}(\hat{G}_{\text{LCQP}})$. $\qquad \square$

*Proof of Theorem A.3.* Under the same setting as in the proof of Theorem A.2, the results can be proved using the same arguments as in the proof of Lemma B.4 and Corollary B.7 in Chen et al. (2023a). We present the proof here for completeness.

Let $x \in \mathbb{R}^n$ be the optimal solution to (2.1) with the smallest $\ell_2$-norm, and let $\hat{x} \in \mathbb{R}^n$ be defined as in the proof of Theorem A.2. By the arguments in the proof of Theorem A.2, $\hat{x}$ is an optimal solution to (A.3). In particular, $\hat{x}$ is also an optimal solution to (2.1) since one can set $(\hat{A}, \hat{b}, \hat{c}, \hat{Q}, \hat{l}, \hat{u}, \hat{\circ}) = (A, b, c, Q, l, u, \circ)$. Therefore, by the minimality of $\|x\|^2$, we have that

$$\|x\|^2 \leq \|\hat{x}\|^2 = \sum_{q=1}^{t} \sum_{j \in J_q} \hat{x}_j^2 = \sum_{q=1}^{t} |J_q| \left( \frac{1}{|J_q|} \sum_{j \in J_q} x_j \right)^2 \leq \sum_{q=1}^{t} \sum_{j \in J_q} x_j^2 = \|x\|^2,$$

which implies that $x_j$ is a constant in $j \in J_q$ and $x = \hat{x}$. Thus, $x$ is also an optimal solution to (A.3).

Let $x' \in \mathbb{R}^n$ be the optimal solution to (A.3) with the smallest $\ell_2$-norm. Then $\|x'\| \leq \|\hat{x}\| = \|x\|$ and the reverse direction $\|x\| \leq \|x'\|$ is also true, which implies that $\|x\| = \|x'\|$. Therefore, we have $x = x'$ by the uniqueness of the optimal solution with the smallest $\ell_2$-norm.

Noticing that the above arguments are made after permuting vertices in $V$ and $W$, we can conclude that $\Phi_{\text{sol}}(G_{\text{LCQP}}) = \sigma_W(\Phi_{\text{sol}}(\hat{G}_{\text{LCQP}}))$ for some $\sigma_W \in S_n$. Additionally, if $G_{\text{LCQP}} \overset{W}{\sim} \hat{G}_{\text{LCQP}}$, then there is no need to perform the permutation on $W$ and we have $\Phi_{\text{sol}}(G_{\text{LCQP}}) = \Phi_{\text{sol}}(\hat{G}_{\text{LCQP}})$. $\qquad \square$

**Corollary A.5.** *For any $G_{\text{LCQP}} \in \mathcal{G}_{\text{LCQP}}^{m,n}$ that is feasible and bounded and any $j, j' \in \{1, 2, \ldots, n\}$, if $C_j^{L,W} = C_{j'}^{L,W}$ holds for all $L \in \mathbb{N}_+$ and all hash functions, then $\Phi_{\text{sol}}(G_{\text{LCQP}})_j = \Phi_{\text{sol}}(G_{\text{LCQP}})_{j'}$.*

*Proof.* Let $\hat{G}_{\text{LCQP}}$ be the LCQP-graph obtained from $G_{\text{LCQP}}$ by relabeling $j$ as $j'$ and relabeling $j'$ as $j$. By Theorem A.3, we have $\Phi_{\text{sol}}(G_{\text{LCQP}}) = \Phi_{\text{sol}}(\hat{G}_{\text{LCQP}})$, which implies $\Phi_{\text{sol}}(G_{\text{LCQP}})_j = \Phi_{\text{sol}}(\hat{G}_{\text{LCQP}})_j = \Phi_{\text{sol}}(G_{\text{LCQP}})_{j'}$. $\qquad \square$

It is well-known from previous literature that the separation power of GNNs is equivalent to that of WL test and that GNNs can universally approximate any continuous function whose separation is not stronger than that of WL test; see e.g. Chen et al. (2023a); Xu et al. (2019); Azizian & Lelarge (2021); Geerts & Reutter (2022). We have established in Theorem A.2, Theorem A.3, and Corollary A.5 that the separation power of $\Phi_{\text{obj}}$ and $\Phi_{\text{sol}}$ is upper bounded by the WL test (Algorithm 2) that shares the same information aggregation mechanism as the GNNs in $\mathcal{F}_{\text{LCQP}}$ and $\mathcal{F}_{\text{LCQP}}^W$. Therefore, Theorem 3.3 and Theorem 3.4 can be proved using standard arguments in the previous literature.

*Proof of Theorem 3.3.* Based on Theorem A.2, Theorem 3.3 can be proved following the same lines as in the proof of Theorem 3.4 in Chen et al. (2023a), with straightforward modifications to generalize results for LP-graphs to the LCQP setting. We sketch the proof here for the sake of self-containedness.

The separation power of GNNs is equivalent to that of the WL test, i.e., for any $G_{\text{LCQP}}, \hat{G}_{\text{LCQP}} \in \mathcal{G}_{\text{LCQP}}^{m,n}$ with $Q, \hat{Q} \succeq 0$,

$$G_{\text{LCQP}} \sim \hat{G}_{\text{LCQP}} \iff F(G_{\text{LCQP}}) = F(\hat{G}_{\text{LCQP}}), \ \forall \, F \in \mathcal{F}_{\text{LCQP}}, \tag{A.4}$$

which combined with Theorem A.2 leads to that

$$F(G_{\text{LCQP}}) = F(\hat{G}_{\text{LCQP}}), \ \forall \, F \in \mathcal{F}_{\text{LCQP}} \implies \Phi_{\text{obj}}(G_{\text{LCQP}}) = \Phi_{\text{obj}}(\hat{G}_{\text{LCQP}}), \tag{A.5}$$

indicating that the separation power of $\mathcal{F}_{\text{LCQP}}$ is upper bounded by that of $\Phi_{\text{obj}}$.

The indicator function $\mathbb{I}_{\Phi_{\text{obj}}(\cdot) \in \mathbb{R}} : \mathcal{G}_{\text{LCQP}}^{m,n} \to \{0,1\} \subset \mathbb{R}$ is measurable, and hence by Lusin's theorem, there exists a compact and permutation-invariant subspace $X \subset \mathcal{G}_{\text{LCQP}}^{m,n}$ such that $\mathbb{P}[\mathcal{G}_{\text{LCQP}}^{m,n} \backslash X] < \epsilon$ and that $\mathbb{I}_{\Phi_{\text{obj}}(\cdot) \in \mathbb{R}}$ restricted on $X$ is continuous. Therefore, by the Stone-Weierstrass theorem and (A.5), we have that there exists $F_1 \in \mathcal{F}_{\text{LCQP}}$ satisfying

$$\sup_{G_{\text{LCQP}} \in X} \left| F_1(G_{\text{LCQP}}) - \mathbb{I}_{\Phi_{\text{obj}}(G_{\text{LCQP}}) \in \mathbb{R}} \right| < \frac{1}{2}$$

Therefore, it holds that

$$\mathbb{P} \left[ \mathbb{I}_{F_1(G_{\text{LCQP}}) > \frac{1}{2}} \neq \mathbb{I}_{\Phi_{\text{obj}}(G_{\text{LCQP}}) \in \mathbb{R}} \right] \leq \mathbb{P} \left[ \mathcal{G}_{\text{LCQP}}^{m,n} \backslash X \right] < \epsilon,$$

which proves (3.1). Additionally, (3.2) can be proved by applying similar arguments to $\Phi_{\text{obj}} : \Phi_{\text{obj}}^{-1}(\mathbb{R}) \to \mathbb{R}$, where $\Phi_{\text{obj}}^{-1}(\mathbb{R}) \subset \mathcal{G}_{\text{LCQP}}^{m,n}$ is the collection of feasible and bounded $G_{\text{LCQP}} \in \mathcal{G}_{\text{LCQP}}^{m,n}$. $\quad\square$

*Proof of Theorem 3.4.* Based on Theorem A.3 and Corollary A.5, Theorem 3.4 can be proved following the same lines as in the proof of Theorem 3.6 in Chen et al. (2023a), with trivial modifications to generalize results for LP-graphs to the LCQP setting. We sketch the proof here for the sake of self-containedness.

In addition to (A.4), it can be proved that the separation powers of GNNs and the WL test are equivalent in the following sense:

- For any $G_{\text{LCQP}}, \hat{G}_{\text{LCQP}} \in \mathcal{G}_{\text{LCQP}}^{m,n}$, $G_{\text{LCQP}} \overset{W}{\sim} \hat{G}_{\text{LCQP}}$ if and only if $F_W(G_{\text{LCQP}}) = F_W(\hat{G}_{\text{LCQP}})$ for all $F_W \in \mathcal{F}_{\text{LCQP}}^W$.

- For any $G_{\text{LCQP}} \in \mathcal{G}_{\text{LCQP}}^{m,n}$ and any $j, j' \in W$, $C_j^{L,W} = C_{j'}^{L,W}$ for any $L \in \mathbb{N}$ and any hash function if and only if $F_W(G_{\text{LCQP}})_j = F_W(G_{\text{LCQP}})_{j'}$ for all $F_W \in \mathcal{F}_{\text{LCQP}}^W$.

Therefore, with Theorem A.3 and Corollary A.5, the separation power of GNNs is upper bounded by that of $\Phi_{\text{sol}}$ in the following sense that for any $G_{\text{LCQP}}, \hat{G}_{\text{LCQP}} \in \mathcal{G}_{\text{LCQP}}^{m,n}$ with $Q, \hat{Q} \succeq 0$ and any $j, j' \in W$,

- $F(G_{\text{LCQP}}) = F(\hat{G}_{\text{LCQP}})$, $\forall\, F \in \mathcal{F}_{\text{LCQP}}$ implies $\Phi_{\text{sol}}(G_{\text{LCQP}}) = \sigma_W(\Phi_{\text{sol}}(\hat{G}_{\text{LCQP}}))$ for some $\sigma_W \in S_n$.

- $F_W(G_{\text{LCQP}}) = F_W(\hat{G}_{\text{LCQP}})$, $\forall\, F_W \in \mathcal{F}_{\text{LCQP}}^W$ implies $\Phi_{\text{sol}}(G_{\text{LCQP}}) = \Phi_{\text{sol}}(\hat{G}_{\text{LCQP}})$.

- $F_W(G_{\text{LCQP}})_j = F_W(G_{\text{LCQP}})_{j'}$, $\forall\, F_W \in \mathcal{F}_{\text{LCQP}}^W$ implies $\Phi_{\text{sol}}(G_{\text{LCQP}})_j = \Phi_{\text{sol}}(G_{\text{LCQP}})_{j'}$.

The optimal solution mapping $\Phi_{\text{sol}} : \Phi_{\text{obj}}^{-1}(\mathbb{R}) \to \mathbb{R}$ is measurable, and hence by Lusin's theorem, there exists a compact and permutation-invariant subspace $X \subset \Phi_{\text{obj}}^{-1}(\mathbb{R})$ such that $\mathbb{P}[\Phi_{\text{obj}}^{-1}(\mathbb{R}) \backslash X] < \epsilon$ and that $\Phi_{\text{sol}}$ restricted on $X$ is continuous. Therefore, applying the generalized Stone-Weierstrass theorem for equivariant functions (Azizian & Lelarge, 2021, Theorem 22), we know that there exists $F_W \in \mathcal{F}_{\text{LCQP}}^W$ satisfying

$$\sup_{G_{\text{LCQP}} \in X} \|F_W(G_{\text{LCQP}}) - \Phi_{\text{sol}}(G_{\text{LCQP}})\| < \delta.$$

Therefore, it holds that

$$\mathbb{P}\left[\|F_W(G_{\text{LCQP}}) - \Phi_{\text{sol}}(G_{\text{LCQP}})\| > \delta\right] \leq \mathbb{P}\left[\Phi_{\text{obj}}^{-1}(\mathbb{R}) \backslash X\right] < \epsilon,$$

which completes the proof. $\qquad\square$

## B  PROOFS FOR SECTION 4.1

The proof of Proposisition 4.1 is directly from Chen et al. (2023b) since adding a quadratic term in the objective function of an MILP problem does not change the feasible region. However, Proposisitions 4.2 and 4.3 are not covered in Chen et al. (2023b) and we present their proofs here.

*Proof of Proposisition 4.2.* As discussed in Section 4.1, we consider the following two examples whose optimal objective values are $\frac{9}{2}$ and 6, respectively.

$$\min_{x \in \mathbb{R}^6} \frac{1}{2}\sum_{i=1}^{6} x_i^2 + \sum_{i=1}^{6} x_i,$$
$$\text{s.t. } x_1 + x_2 \geq 1,\ x_2 + x_3 \geq 1,\ x_3 + x_4 \geq 1,$$
$$x_4 + x_5 \geq 1,\ x_5 + x_6 \geq 1,\ x_6 + x_1 \geq 1,$$
$$x_j \in \{0,1\},\ \forall\, j \in \{1,2,\ldots,6\}.$$

$$\min_{x \in \mathbb{R}^6} \frac{1}{2}\sum_{i=1}^{6} x_i^2 + \sum_{i=1}^{6} x_i,$$
$$\text{s.t. } x_1 + x_2 \geq 1,\ x_2 + x_3 \geq 1,\ x_3 + x_1 \geq 1,$$
$$x_4 + x_5 \geq 1,\ x_5 + x_6 \geq 1,\ x_6 + x_4 \geq 1,$$
$$x_j \in \{0,1\},\ \forall\, j \in \{1,2,\ldots,6\}.$$

Denote $G_{\text{MI-LCQP}}$ and $\hat{G}_{\text{MI-LCQP}}$ as the graph representations of the above two MI-LCQP problems. Let $s_i^l, t_j^l$ and $\hat{s}_i^l, \hat{t}_j^l$ be the attributes at the $l$-th layer when apply a GNN $F \in \mathcal{F}_{\text{MI-LCQP}}$ to $G_{\text{MI-LCQP}}$ and $\hat{G}_{\text{MI-LCQP}}$. We will prove by induction that for any $0 \leq l \leq L$, the followings hold:

(a)  $s_i^l = \hat{s}_i^l$ and is constant over $i \in \{1,2,\ldots,6\}$.

(b)  $t_j^l = \hat{t}_j^l$ and is constant over $j \in \{1,2,\ldots,6\}$.

It is clear that the conditions (a) and (b) are true for $l = 0$, since $v_i = \hat{v}_i = (1, \geq)$ is constant in $i \in \{1,2,\ldots,6\}$, and $w_j = \hat{w}_j = (1,0,1,1)$ is constant in $j \in \{1,2,\ldots,6\}$. Now suppose that the conditions (a) and (b) are true for $l-1$ where $1 \leq l \leq L$. We denote that $s^{l-1} = s_i^{l-1} = $

$\bar{s}_i^{l-1}, \forall i \in \{1, 2, \ldots, 6\}$ and $t^{l-1} = t_j^{l-1} = \hat{t}_j^{l-1}, \forall j \in \{1, 2, \ldots, 6\}$. It can be computed for any $i \in \{1, 2, \ldots, 6\}$ and $j \in \{1, 2, \ldots, 6\}$ that

$$s_i^l = f_l^V \left( s^{l-1}, \sum_{j \in \mathcal{N}_i^W} g_l^W(t_j^{l-1}, A_{ij}) \right) = f_l^V \left( s^{l-1}, 2g_l^W(t^{l-1}, 1) \right) = \hat{s}_i^l,$$

$$t_j^l = f_l^W \left( t_j^{l-1}, \sum_{i \in \mathcal{N}_j^V} g_l^V(s_i^{l-1}, A_{ij}), \sum_{j' \in \mathcal{N}_j^W} g_l^Q(t_{j'}^{l-1}, Q_{jj'}) \right)$$

$$= f_l^W \left( t^{l-1}, 2g_l^V(s^{l-1}, 1), g_l^Q(t^{l-1}, 1) \right) = \hat{t}_j^l,$$

which proves (a) and (b) for $l$. Thus, we can conclude that $F(G_{\text{MI-LCQP}}) = F(\hat{G}_{\text{MI-LCQP}}), \forall F \in \mathcal{F}_{\text{MI-LCQP}}$. $\qquad\square$

*Proof of Proposition 4.3.* Consider the following two MI-LCQPs:

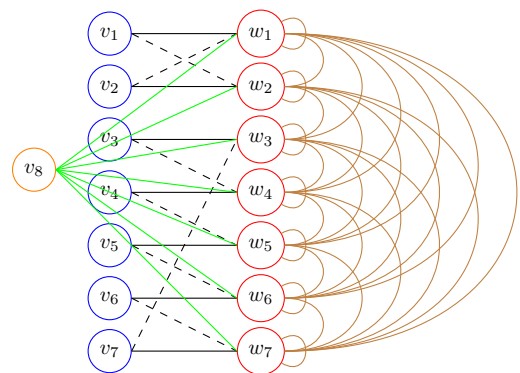

$$\min_{x \in \mathbb{R}^7} \frac{1}{2} x^\top \mathbf{1} \mathbf{1}^\top x + \mathbf{1}^\top x,$$

$$\text{s.t. } x_1 - x_2 = 0, \ x_2 - x_1 = 0,$$
$$x_3 - x_4 = 0, \ x_4 - x_5 = 0,$$
$$x_5 - x_6 = 0, \ x_6 - x_7 = 0, \ x_7 - x_3 = 0,$$
$$x_1 + x_2 + x_3 + x_4 + x_5 + x_6 + x_7 = 6$$
$$0 \le x_j \le 3, \ x_j \in \mathbb{Z}, \ \forall j \in \{1, 2, \ldots, 7\}.$$

and

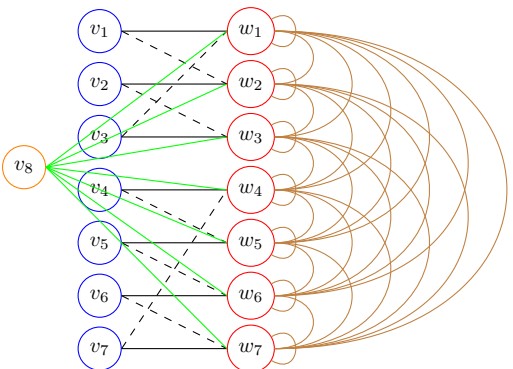

$$\min_{x \in \mathbb{R}^7} \frac{1}{2} x^\top \mathbf{1} \mathbf{1}^\top x + \mathbf{1}^\top x,$$

$$\text{s.t. } x_1 - x_2 = 0, \ x_2 - x_3 = 0, \ x_3 - x_1 = 0,$$
$$x_4 - x_5 = 0, \ x_5 - x_6 = 0,$$
$$x_6 - x_7 = 0, \ x_7 - x_4 = 0,$$
$$x_1 + x_2 + x_3 + x_4 + x_5 + x_6 + x_7 = 6$$
$$0 \le x_j \le 3, \ x_j \in \mathbb{Z}, \ \forall j \in \{1, 2, \ldots, 7\}.$$

As we mentioned in Section 4.1, both problems are feasible with the same optimal objective value, but have disjoint optimal solution sets.

On the other hand, it can be analyzed using the same argument as in the proof of Proposition 4.2 that for any $0 \le l \le L$ that

(a) $s_i^l = \hat{s}_i^l$ is constant over $i \in \{1, 2, \ldots, 7\}$, and $s_8^l = \hat{s}_8^l$.

(b) $t_j^l = \hat{t}_j^l$ is constant over $j \in \{1, 2, \ldots, 7\}$.

These two conditions guarantee that $F(G_{\text{MI-LCQP}}) = F(\hat{G}_{\text{MI-LCQP}}), \forall F \in \mathcal{F}_{\text{MI-LCQP}}$ and $F_W(G_{\text{MI-LCQP}}) = F_W(\hat{G}_{\text{MI-LCQP}}), \forall F_W \in \mathcal{F}_{\text{MI-LCQP}}$. $\qquad\square$

## C  PROOFS FOR SECTION 4.2

This section collects the proofs of Theorems 4.6, 4.7, and 4.9. Similar to the LCQP case, the proofs are also based on the WL test (Algorithm 1) and its separation power to distinguish MI-LCQP problems with different properties. We define the separation power of Algorithm 1 as follows.

**Definition C.1.** *Let* $G_{\text{MI-LCQP}}, \hat{G}_{\text{MI-LCQP}} \in \mathcal{G}_{\text{MI-LCQP}}^{m,n}$ *be two MI-LCQP-graphs and let* $\{\{C_i^{L,V}\}\}_{i=0}^m, \{\{C_j^{L,W}\}\}_{j=0}^n$ *and* $\{\{\hat{C}_i^{L,V}\}\}_{i=0}^m, \{\{\hat{C}_j^{L,W}\}\}_{j=0}^n$ *be color multisets output by Algorithm 1 on* $G_{\text{MI-LCQP}}$ *and* $\hat{G}_{\text{MI-LCQP}}$.

1. *We say* $G_{\text{MI-LCQP}} \sim \hat{G}_{\text{MI-LCQP}}$ *if* $\{\{C_i^{L,V}\}\}_{i=0}^m = \{\{\hat{C}_i^{L,V}\}\}_{i=0}^m$ *and* $\{\{C_j^{L,W}\}\}_{j=0}^n = \{\{\hat{C}_j^{L,W}\}\}_{j=0}^n$ *hold for all* $L \in \mathbb{N}$ *and all hash functions.*

2. *We say* $G_{\text{MI-LCQP}} \overset{W}{\sim} \hat{G}_{\text{MI-LCQP}}$ *if* $\{\{C_i^{L,V}\}\}_{i=0}^m = \{\{\hat{C}_i^{L,V}\}\}_{i=0}^m$ *and* $C_j^{L,W} = \hat{C}_j^{L,W}, \ \forall\, j \in \{1, 2, \ldots, n\}$, *for all* $L \in \mathbb{N}$ *and all hash functions.*

The key component in the proof is to show that for unfoldable/MP-tractable MI-LCQP problems, if they are indistinguishable by WL test, then they must share some common properties.

**Theorem C.2.** *For two MP-tractable MI-LCQP-graphs* $G_{\text{MI-LCQP}}, \hat{G}_{\text{MI-LCQP}} \in \mathcal{G}_{\text{MP}}^{m,n}$, *if* $G_{\text{MI-LCQP}} \sim \hat{G}_{\text{MI-LCQP}}$, *then* $\Phi_{\text{feas}}(G_{\text{MI-LCQP}}) = \Phi_{\text{feas}}(\hat{G}_{\text{MI-LCQP}})$ *and* $\Phi_{\text{obj}}(G_{\text{MI-LCQP}}) = \Phi_{\text{obj}}(\hat{G}_{\text{MI-LCQP}})$.

*Proof.* Let $G_{\text{MI-LCQP}}$ and $\hat{G}_{\text{MI-LCQP}}$ be the MI-LCQP-graphs associated to

$$\min_{x \in \mathbb{R}^n} \ \frac{1}{2} x^\top Q x + c^\top x, \quad \text{s.t. } Ax \circ b, \ l \leq x \leq u, \ x_j \in \mathbb{Z}, \ \forall\, j \in I. \tag{C.1}$$

and

$$\min_{x \in \mathbb{R}^n} \ \frac{1}{2} x^\top \hat{Q} x + \hat{c}^\top x, \quad \text{s.t. } \hat{A}x \hat{\circ} \hat{b}, \ \hat{l} \leq x \leq \hat{u}, \ x_j \in \mathbb{Z}, \ \forall\, j \in \hat{I}. \tag{C.2}$$

Suppose that there are no collisions of hash functions or their linear combinations when applying the WL test to $G_{\text{MI-LCQP}}$ and $\hat{G}_{\text{MI-LCQP}}$ and there are no strict color refinements in the $L$-th iteration. Since $G_{\text{MI-LCQP}} \sim \hat{G}_{\text{MI-LCQP}}$ and both of them are MP-tractable, after performing some permutation, there exist $\mathcal{I} = \{I_1, I_2, \ldots, I_s\}$ and $\mathcal{J} = \{J_1, J_2, \ldots, J_t\}$ that are partitions of $\{1, 2, \ldots, m\}$ and $\{1, 2, \ldots, n\}$, respectively, such that the followings hold:

- $C_i^{L,V} = C_{i'}^{L,V}$ if and only if $i, i' \in I_p$ for some $p \in \{1, 2, \ldots, s\}$.

- $C_i^{L,V} = \hat{C}_{i'}^{L,V}$ if and only if $i, i' \in I_p$ for some $p \in \{1, 2, \ldots, s\}$.

- $\hat{C}_i^{L,V} = \hat{C}_{i'}^{L,V}$ if and only if $i, i' \in I_p$ for some $p \in \{1, 2, \ldots, s\}$.

- $C_j^{L,W} = C_{j'}^{L,W}$ if and only if $j, j' \in J_q$ for some $q \in \{1, 2, \ldots, t\}$.

- $C_j^{L,W} = \hat{C}_{j'}^{L,W}$ if and only if $j, j' \in J_q$ for some $q \in \{1, 2, \ldots, t\}$.

- $\hat{C}_j^{L,W} = \hat{C}_{j'}^{L,W}$ if and only if $j, j' \in J_q$ for some $q \in \{1, 2, \ldots, t\}$.

By similar analysis as in the proof of Theorem A.2, we have

(a) $v_i = \hat{v}_i$ and is constant over $i \in I_p$ for any $p \in \{1, 2, \ldots, s\}$.

(b) $w_j = \hat{w}_j$ and is constant over $j \in J_q$ for any $q \in \{1, 2, \ldots, t\}$.

(c) For any $p \in \{1, 2, \ldots, s\}$ and any $q \in \{1, 2, \ldots, t\}$, $\{\{A_{ij} : j \in J_q\}\} = \{\{\hat{A}_{ij} : j \in J_q\}\}$ and is constant over $i \in I_p$.

(d) For any $p \in \{1, 2, \ldots, s\}$ and any $q \in \{1, 2, \ldots, t\}$, $\{\{A_{ij} : i \in I_p\}\} = \{\{\hat{A}_{ij} : i \in I_p\}\}$ and is constant over $j \in J_q$.

(e) For any $q, q' \in \{1, 2, \ldots, t\}$, $\{\{Q_{jj'} : j' \in J_{q'}\}\} = \{\{\hat{Q}_{jj'} : j' \in J_{q'}\}\}$ and is constant over $j \in J_q$.

Note that $G_{\text{MI-LCQP}}$ and $\hat{G}_{\text{MI-LCQP}}$ are both MP-tractable, i.e., all submatrices $(A_{ij})_{i \in I_p, j \in J_q}$, $(\hat{A}_{ij})_{i \in I_p, j \in J_q}$, $(Q_{jj'})_{j \in J_q, j' \in J_{q'}}$, and $(\hat{Q}_{jj'})_{j \in J_q, j' \in J_{q'}}$ have identical entries. The above conditions (c)-(e) suggest that

(f) For any $p \in \{1, 2, \ldots, s\}$ and any $q \in \{1, 2, \ldots, t\}$, $A_{ij} = \hat{A}_{ij}$ and is constant over $i \in I_p, j \in J_q$.

(g) For any $q, q' \in \{1, 2, \ldots, t\}$, $Q_{jj'} = \hat{Q}_{jj'}$ and is constant over $j \in J_q, j' \in J_{q'}$.

Combining conditions (a), (b), (f), and (g), we can conclude that $G_{\text{MI-LCQP}}$ and $\hat{G}_{\text{MI-LCQP}}$ are actually identical after applying some permutation, i.e., they are isomorphic, which implies $\Phi_{\text{feas}}(G_{\text{MI-LCQP}}) = \Phi_{\text{feas}}(\hat{G}_{\text{MI-LCQP}})$ and $\Phi_{\text{obj}}(G_{\text{MI-LCQP}}) = \Phi_{\text{obj}}(\hat{G}_{\text{MI-LCQP}})$. □

Before stating the next result, we comment on the construction/definition of the MI-LCQP optimal solution mapping $\Phi_{\text{sol}}$. Different from the LCQP setting, the optimal solution to an MI-LCQP problem may not exist even if it is feasible and bounded, i.e., $\Phi_{\text{obj}}(G_{\text{MI-LCQP}}) \in \mathbb{R}$. Thus, we have to work with $\mathcal{G}_{\text{sol}}^{m,n} \subset \Phi_{\text{obj}}^{-1}(\mathbb{R}) \subset \mathcal{G}_{\text{MI-LCQP}}^{m,n}$ where $\mathcal{G}_{\text{sol}}^{m,n}$ is the collection of all MI-LCQP-graphs for which an optimal solution exists. For $G_{\text{MI-LCQP}} \in \mathcal{G}_{\text{sol}}^{m,n}$, it is possible that it admits multiple optimal solution. Moreover, there may even exist multiple optimal solutions with the smallest $\ell_2$-norm due to its non-convexity, which means that we cannot define the optimal solution mapping $\Phi_{\text{sol}}$ using the same approach as in the LCQP case. If we further assume that $G_{\text{MI-LCQP}} \in \mathcal{G}_{\text{sol}}^{m,n}$ is unfoldable, then using the same approach as in Chen et al. (2023b, Appendix C), one can define a total ordering on the optimal solution set and hence define $\Phi_{\text{sol}}(G_{\text{MI-LCQP}})$ as the minimal element in the optimal solution set, which is unique and permutation-equivariant, meaning that if one relabels vertices of $G_{\text{MI-LCQP}}$, then entries of $\Phi_{\text{sol}}(G_{\text{MI-LCQP}})$ are relabelled accordingly.

**Theorem C.3.** *For any two MI-LCQP-graphs $G_{\text{MI-LCQP}}, \hat{G}_{\text{MI-LCQP}} \in \mathcal{G}_{\text{sol}}^{m,n} \cap \mathcal{G}_{\text{unfold}}^{m,n}$ that are unfoldable with nonempty optimal solution sets, if $G_{\text{MI-LCQP}} \sim \hat{G}_{\text{MI-LCQP}}$, then there exists some permutation $\sigma_W \in S_n$ such that $\Phi_{\text{sol}}(G_{\text{MI-LCQP}}) = \sigma_W(\Phi_{\text{sol}}(\hat{G}_{\text{MI-LCQP}}))$. Furthermore, if $G_{\text{MI-LCQP}} \overset{W}{\sim} \hat{G}_{\text{MI-LCQP}}$, then $\Phi_{\text{sol}}(G_{\text{MI-LCQP}}) = \Phi_{\text{sol}}(\hat{G}_{\text{MI-LCQP}})$.*

*Proof.* By Proposition D.1, $G_{\text{MI-LCQP}}$ and $\hat{G}_{\text{MI-LCQP}}$ are also MP-tractable, and hence, all analysis in the proof of Theorem C.2 applies. If $G_{\text{MI-LCQP}} \sim \hat{G}_{\text{MI-LCQP}}$, then they are isomorphic and $\Phi_{\text{sol}}(G_{\text{MI-LCQP}}) = \sigma_W(\Phi_{\text{sol}}(\hat{G}_{\text{MI-LCQP}}))$ for some permutation $\sigma_W \in S_n$. If $G_{\text{MI-LCQP}} \overset{W}{\sim} \hat{G}_{\text{MI-LCQP}}$, then these two graphs will become identical after applying some permutation on $V$ with the labeling in $W$ unchanged, which guarantees $\Phi_{\text{sol}}(G_{\text{MI-LCQP}}) = \Phi_{\text{sol}}(\hat{G}_{\text{MI-LCQP}})$. □

With Theorem C.2 and Theorem C.3, one can adopt standard argument in the previous literature to prove Theorems 4.6, 4.7, and 4.9.

*Proof of Theorem 4.6.* Based on Theorem C.2, Theorem 4.6 can be proved following the same lines as in the proof of Theorem 3.2 in Chen et al. (2023a), with straightforward modifications to generalize results for LP-graphs to the MI-LCQP setting. In particular, the proof outline is the same as the proof of Theorem 3.3. □

*Proof of Theorem 4.7.* Based on Theorem C.2, Theorem 4.7 can be proved following the same lines as in the proof of Theorem 3.4 in Chen et al. (2023a), with straightforward modifications to generalize results for LP-graphs to the MI-LCQP setting. In particular, the proof outline is the same as the proof of Theorem 3.3. □

*Proof of Theorem 4.9.* Based on Theorem C.3 and the unfoldability assumption that different vertices in $W$ will eventually have different colors in the WL test without collision, which automatically provides a result of the same spirit as Corollary A.5, Theorem 4.9 can be proved following the same lines as in the proof of Theorem 3.6 in Chen et al. (2023a), with straightforward modifications to generalize results for LP-graphs to the MI-LCQP setting. In particular, the proof outline is the same as the proof of Theorem 3.4. □

**Discussions on various GNN architectures:** In our work we use the sum aggregation, and all results are still valid for the weighted average aggregation. In particular, all our proofs (such as the proof of Theorem A.2) hold almost verbatimly for the average aggregation. The attention aggregation Veličković et al. (2017) has stronger separation power, which implies that all universal approximation results still hold. Moreover, all the counter examples for MI-LCQPs work for every aggregation approach, since the color refinement in Algorithm 1 is implemented on multisets, with separation power stronger than or equal to all aggregations of neighboring information. We have included the above discussion in our updated draft.

# D CHARACTERIZATION OF MP-TRACTABILITY AND UNFOLDABILITY

In this section, we discuss some further characterizations of the MP-tractability and the unfoldability for MI-LCQP-graphs defined in Section 4.3.

## D.1 RELATIONSHIP BETWEEN MP-TRACTABILITY AND UNFOLDABILITY

We first prove that unfoldability implies MP-tractability but they are not equivalent.

**Proposition D.1.** *If $G_{\text{MI-LCQP}} \in \mathcal{G}_{\text{MI-LCQP}}^{m,n}$ is unfoldable, then it is also MP-tractable.*

*Proof.* Let $(\mathcal{I}, \mathcal{J})$ be the final stable partition of $V \cup W$ generated by WL test on $G_{\text{MI-LCQP}}$ without collision, where $\mathcal{I} = \{I_1, I_2, \ldots, I_s\}$ is a partition of $V = \{1, 2, \ldots, m\}$ and $\mathcal{J} = \{J_1, J_2, \ldots, J_t\}$ is a partition of $W = \{1, 2, \ldots, n\}$. Since we assume that $G_{\text{MI-LCQP}}$ is foldable, we have $t = n$ and $|J_1| = |J_2| = \cdots = |J_n| = 1$. Then for any $q, q' \in \{1, 2, \ldots, t\}$, the submatrix $(Q_{jj'})_{j \in J_q, j' \in J_{q'}}$ is a $1 \times 1$ matrix and hence has identical entries.

Consider any $p \in \{1, 2, \ldots, s\}$ and $q \in \{1, 2, \ldots, t\}$. Suppose that the color positioning is stabilized at the $L$-th iteration of WL test. Then for any $i, i' \in I_p$, we have

$$C_i^{L,V} = C_{i'}^{L,V}$$
$$\implies \left\{\left\{\text{HASH}\left(C_j^{L-1,W}, A_{ij}\right) : j \in \mathcal{N}_i^W\right\}\right\} = \left\{\left\{\text{HASH}\left(C_j^{L-1,W}, A_{i'j}\right) : j \in \mathcal{N}_i^W\right\}\right\}$$
$$\implies \{\{A_{ij} : j \in J_q\}\} = \{\{A_{i'j} : j \in J_q\}\},$$

which implies that the submatrix $(A_{ij})_{i \in I_p, j \in J_q}$ has identical entries since $|J_q| = 1$. Therefore, $G_{\text{MI-LCQP}}$ is MP-tractable. □

**Proposition D.2.** *There exist MP-tractable instances in $\mathcal{G}_{\text{MI-LCQP}}^{m,n}$ that are not unfoldable.*

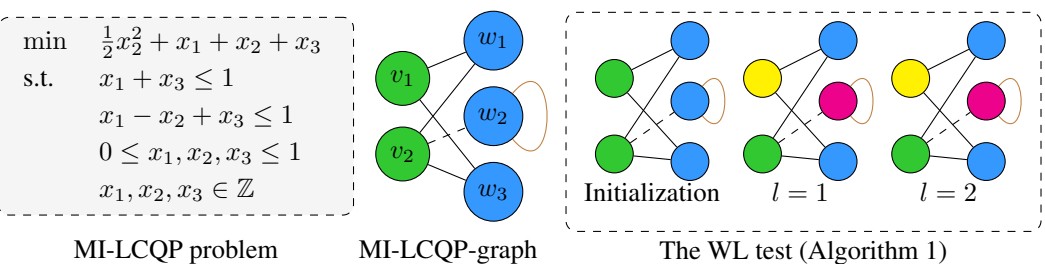

MI-LCQP problem      MI-LCQP-graph      The WL test (Algorithm 1)

Figure 5: Example for proving Proposition D.2

*Proof.* Consider the example in Figure 5, for which the final stable partition is $\mathcal{I} = \{\{1\}, \{2\}\}$ and $\mathcal{J} = \{\{1, 3\}, \{2\}\}$. It is not unfoldable since the class $\{1, 3\}$ in $\mathcal{J}$ has two elements. However, it is MP-tractable since $A_{11} = A_{13} = 1$ and $A_{21} = A_{23} = 1$. □

## D.2 FREQUENCY OF MP-TRACTABILITY AND UNFOLDABILITY

It can be proved that a generic MI-LCQP-graph in $\mathcal{G}_{\text{MI-LCQP}}^{m,n}$ is unfoldable almost surely under some mild conditions. Intuitively, if $c \in \mathbb{R}^n$ is randomly sampled from a continuous distribution with density, then almost surely it holds that $x_j \neq x_{j'}$ for any $j \neq j'$, which implies that the vertices in $W$ have different colors initially and always, if there are no collisions of hash functions.

**Proposition D.3.** *Let $\mathbb{P}$ be a probability measure over $\mathcal{G}_{\text{MI-LCQP}}$ such that the marginal distribution $\mathbb{P}_c$ of $c \in \mathbb{R}^n$ has density. Then $\mathbb{P}[\mathcal{G}_{\text{MI-LCQP}} \in \mathcal{G}_{\text{unfold}}^{m,n}] = 1$.*

*Proof.* Since the marginal distribution $\mathbb{P}_c$ has density, almost surely we have for any $j \neq j'$ that

$$c_j \neq c_{j'} \implies C_j^{0,W} \neq C_{j'}^{0,W} \implies C_j^{l,W} \neq C_{j'}^{l,W}, \quad \forall\, l \geq 0,$$

where we assumed that no collisions happen in hash functions. Therefore, any $j, j' \in W$ with $j \neq j'$ are not the in same class of the final stable partition $(\mathcal{I}, \mathcal{J})$, which proves the unfoldability. □

As a direct corollary of Proposition D.1 and Proposition D.3, a generic MI-LCQP-graph in $\mathcal{G}_{\text{MI-LCQP}}^{m,n}$ must also be MP-tractable.

**Corollary D.4.** *Let $\mathbb{P}$ be a probability measure over $\mathcal{G}_{\text{MI-LCQP}}$ such that the marginal distribution $\mathbb{P}_c$ of $c \in \mathbb{R}^n$ has density. Then $\mathbb{P}[\mathcal{G}_{\text{MI-LCQP}} \in \mathcal{G}_{\text{MP}}^{m,n}] = 1$.*

# E EXTENSION TO QUADRATICALLY CONSTRAINED QUADRATIC PROGRAMS

A general quadratically constrained quadratic programming (QCQP) is given by

$$\min_{x \in \mathbb{R}^n} \quad \frac{1}{2} x^\top Q x + c^\top x, \quad \text{s.t.} \quad \frac{1}{2} x^\top P_i x + a_i^\top x \leq b_i, \ 1 \leq i \leq m, \ l \leq x \leq u, \quad \text{(E.1)}$$

where $Q, P_i \in \mathbb{R}^{n \times n}$ are symmetric, $c, a_i \in \mathbb{R}^n$, $b_i \in \mathbb{R}$, $l \in (\mathbb{R} \cup \{-\infty\})^n$, and $u \in (\mathbb{R} \cup \{+\infty\})^n$. We denote $A = \begin{bmatrix} a_1 & a_2 & \cdots & a_m \end{bmatrix}^\top \in \mathbb{R}^{m \times n}$ for consistent notation with (2.1).

## E.1 GRAPH REPRESENTATION AND GNNS FOR QCQPS

**Graph representation for QCQPs** The QCQP-graph for representing (E.1) is based on the LCQP-graph introduced in Section 2. More specifically, The QCQP graph can be constructed by incorporating the information from $P = (P_1, P_2, \ldots, P_m)$ into the LCQP graph:

- The multiset $\{\{i, j, j'\}\}$ is viewed as a hyperedge with weight $(H_i)_{jj'}$ for each $i \in V$ and $j, j' \in W$, where $j = j'$ is allowed.

We use $\mathcal{G}_{\text{QCQP}}^{m,n}$ to denote the set of all QCQP-graphs with $m$ constraints and $n$ variables.

**GNNs for solving QCQP** Note GNNs on LCQP-graphs that iterate vertex features with message-passing mechanism, which does not naturally adapt to the hyperedges in QCQP graphs. Thus, one idea is to add edge features for each pair $(i, j)$, $i \in V, j \in W$. We describe the GNN architecture for QCQP tasks in detail as follows.

The initial layer computes node features $s_i^0, t_j^0$ and edge features $e_{ij}^0$ via embedding:

- $s_i^0 = f_0^V(v_i)$ for $i \in V$,
- $t_j^0 = f_0^W(w_j)$ for $j \in W$, and
- $e_{ij}^0 = f_0^E(A_{ij})$ for $i \in V, j \in W$.

The $l$-th message-passing layers ($l = 1, 2, \ldots, L$) update the node features using neighbors' information:

- $s_i^l = f_l^V \left( s_i^{l-1}, \sum_{j \in W} g_l^V(t_j^{l-1}, e_{ij}^{l-1}) \right)$ for $i \in V$,

- $t_j^l = f_l^W \left( t_j^{l-1}, \sum_{i \in V} g_l^W(s_i^{l-1}, e_{ij}^{l-1}), \sum_{j' \in W} Q_{jj'} g_l^Q(t_{j'}^{l-1}) \right)$ for $j \in W$, and

- $e_{ij}^l = f_l^E \left( e_{ij}^{l-1}, \sum_{j' \in W} (P_i)_{jj'} g_l^E(t_{j'}^{l-1}) \right)$ for $i \in V, j \in W$.

Finally, there are two types of output layers. The graph-level output computes a single real number for the whole graph

- $y = r_1 \left( \sum_{i \in V} s_i^L, \sum_{j \in W} t_j^L \right) \in \mathbb{R}$,

and the node-level output computes a vector $y \in \mathbb{R}^n$ with the $j$-th entry being

- $y_j = r_2 \left( \sum_{i \in V} s_i^L, \sum_{j \in W} t_j^L, t_j^L \right)$.

We use $\mathcal{F}_{\text{QCQP}}$ (or $\mathcal{F}_{\text{QCQP}}^W$) to denote the collection of all message-passing GNNs with graph-level (or node-level) outputs that are constructed by continuous $f_0^V, f_0^W, f_0^E,$ $f_l^V, f_l^W, f_l^E, g_l^V, g_l^W, g_l^E, g_l^Q$ ($1 \le l \le L$), and $r_1$ (or $r_2$).

### E.2 Universal Approximation of GNNs for QCQPs

For QCQPs, we still consider the three target mappings, i.e., the feasible mapping $\Phi_{\text{feas}} : \mathcal{G}_{\text{QCQP}}^{m,n} \to \{0, 1\}$, the optimal objective value mapping $\Phi_{\text{obj}} : \mathcal{G}_{\text{QCQP}}^{m,n} \to \mathbb{R} \cup \{\pm\infty\}$, and the optimal solution mapping $\Phi_{\text{obj}}$ that computes the unique optimal solution with the smallest $\ell_2$-norm of feasible and bounded QCQPs with $Q, P_i \succeq 0$, $i = 1, 2, \ldots, m$. The main results that GNNs can universally approximate these three target mappings are stated as follows.

**Assumption E.1.** $\mathbb{P}$ *is a Borel regular probability measure on $\mathcal{G}_{\text{QCQP}}^{m,n}$[6].*

**Theorem E.2.** *Let $\mathbb{P}$ be a probability measure satisfying Assumption E.1 and $\mathbb{P}[Q \succeq 0] = \mathbb{P}[P_i \succeq 0] = 1$, $i = 1, 2, \ldots, m$. For any $\epsilon > 0$, there exists $F \in \mathcal{F}_{\text{MI-LCQP}}$ such that*

$$\mathbb{P}\left[ \mathbb{I}_{F(G_{\text{QCQP}}) > \frac{1}{2}} \ne \Phi_{\text{feas}}(G_{\text{QCQP}}) \right] < \epsilon.$$

**Theorem E.3.** *Let $\mathbb{P}$ be a probability measure satisfying Assumption E.1 and $\mathbb{P}[Q \succeq 0] = \mathbb{P}[P_i \succeq 0] = 1$, $i = 1, 2, \ldots, m$. For any $\epsilon > 0$, there exists $F_1 \in \mathcal{F}_{\text{QCQP}}$ such that*

$$\mathbb{P}\left[ \mathbb{I}_{F_1(G_{\text{QCQP}}) > \frac{1}{2}} \ne \mathbb{I}_{\Phi_{\text{obj}}(G_{\text{QCQP}}) \in \mathbb{R}} \right] < \epsilon.$$

*Additionally, if $\mathbb{P}[\Phi_{\text{obj}}(G_{\text{QCQP}}) \in \mathbb{R}] = 1$, for any $\epsilon, \delta > 0$, there exists $F_2 \in \mathcal{F}_{\text{QCQP}}$ such that*

$$\mathbb{P}\left[ |F_2(G_{\text{QCQP}}) - \Phi_{\text{obj}}(G_{\text{QCQP}})| > \delta \right] < \epsilon.$$

**Theorem E.4.** *Let $\mathbb{P}$ be a probability measure satisfying Assumption E.1 and $\mathbb{P}[Q \succeq 0] = \mathbb{P}[P_i \succeq 0] = 1$, $i = 1, 2, \ldots, m$. For any $\epsilon, \delta > 0$, there exists $F_W \in \mathcal{F}_{\text{QCQP}}^W$ such that*

$$\mathbb{P}\left[ \|F_W(G_{\text{QCQP}}) - \Phi_{\text{sol}}(G_{\text{QCQP}})\| > \delta \right] < \epsilon.$$

Similarly, the proofs of Theorem E.2, E.3, and E.4 are based on showing that the WL test associated with the GNN classes $\mathcal{F}_{\text{QCQP}}$ and $\mathcal{F}_{\text{QCQP}}^W$ have sufficiently strong separation power to distinguish QCQP problems with different properties. We will present and prove such separation results (Theorem E.5, Theorem E.6, and Corollary E.7) in the rest of this subsection, and do not repeat the same arguments as described in the Proof of Theorem 3.3 and Theorem 3.4.

We state in Algorithm 3 the WL test for QCQPs. For QCQP-graphs $G_{\text{QCQP}}, \hat{G}_{\text{QCQP}} \in \mathcal{G}_{\text{QCQP}}^{m,n}$,

---

[6]The space $\mathcal{G}_{\text{QCQP}}^{m,n}$ is equipped with the subspace topology induced from the product space $\{(A, b, c, Q, P, l, u, \circ) : A \in \mathbb{R}^{m \times n}, b \in \mathbb{R}^m, c \in \mathbb{R}^n, Q \in \mathbb{R}^{n \times n}, P \in (\mathbb{R}^{n \times n})^m, l \in (\mathbb{R} \cup \{-\infty\})^n, u \in (\mathbb{R} \cup \{+\infty\})^n\}$, where all Euclidean spaces have standard Eudlidean topologies, discrete spaces $\{-\infty\}$ and $\{+\infty\}$ have the discrete topologies, and all unions are disjoint unions.

1. We say $G_{\text{QCQP}} \sim \hat{G}_{\text{QCQP}}$ if $\{\{C_i^{L,V}\}\}_{i=0}^m = \{\{\hat{C}_i^{L,V}\}\}_{i=0}^m$ and $\{\{C_j^{L,W}\}\}_{j=0}^n = \{\{\hat{C}_j^{L,W}\}\}_{j=0}^n$ hold for all $L \in \mathbb{N}$ and all hash functions.

2. We say $G_{\text{QCQP}} \overset{W}{\sim} \hat{G}_{\text{QCQP}}$ if $\{\{C_i^{L,V}\}\}_{i=0}^m = \{\{\hat{C}_i^{L,V}\}\}_{i=0}^m$ and $C_j^{L,W} = \hat{C}_j^{L,W}$, $\forall\, j \in \{1, 2, \ldots, n\}$, for all $L \in \mathbb{N}$ and all hash functions.

---

**Algorithm 3** The WL test for QCQP-Graphs

---

**Require:** A QCQP-graph $G = (V, W, A, Q, P, H_V, H_W)$ and iteration limit $L > 0$.

1: Initialize with

$$C_i^{0,V} = \text{HASH}(v_i),\ C_j^{0,W} = \text{HASH}(w_j),\ C_{ij}^{0,E} = \text{HASH}(A_{ij}).$$

2: **for** $l = 1, 2, \cdots, L$ **do**

3:     Refine the color

$$C_i^{l,V} = \text{HASH}\left(C_i^{l-1,V}, \sum_{j \in W} \text{HASH}\left(C_j^{l-1,W}, C_{ij}^{l-1,E}\right)\right),$$

$$C_j^{l,W} = \text{HASH}\left(C_j^{l-1,W}, \sum_{i \in V} \text{HASH}\left(C_i^{l-1,V}, C_{ij}^{l-1,E}\right), \sum_{j' \in W} Q_{jj'} \text{HASH}(C_{j'}^{l-1,W})\right),$$

$$C_{ij}^{l,E} = \text{HASH}\left(C_{ij}^{l-1,E}, \sum_{j' \in W} (P_i)_{jj'} \text{HASH}(C_{j'}^{l-1,W})\right).$$

4: **end for**

5: **return** The multisets containing all vertex colors $\left\{\left\{C_i^{L,V}\right\}\right\}_{i=0}^m, \left\{\left\{C_j^{L,W}\right\}\right\}_{j=0}^n$.

---

**Theorem E.5.** *Given* $G_{\text{QCQP}}, \hat{G}_{\text{QCQP}} \in \mathcal{G}_{\text{QCQP}}^{m,n}$ *with* $Q, \hat{Q}, P_i, \hat{P}_i \succeq 0$ *for all* $i \in \{1, 2, \ldots, m\}$, *if* $G_{\text{QCQP}} \sim \hat{G}_{\text{QCQP}}$, *then* $\Phi_{\text{feas}}(G_{\text{QCQP}}) = \Phi_{\text{feas}}(\hat{G}_{\text{QCQP}})$ *and* $\Phi_{\text{obj}}(G_{\text{QCQP}}) = \Phi_{\text{obj}}(\hat{G}_{\text{QCQP}})$.

*Proof.* We only show the proof of $\Phi_{\text{obj}}(G_{\text{QCQP}}) = \Phi_{\text{obj}}(\hat{G}_{\text{QCQP}})$ and $\Phi_{\text{feas}}(G_{\text{QCQP}}) = \Phi_{\text{feas}}(\hat{G}_{\text{QCQP}})$ will be a direct corollary.

Let $G_{\text{QCQP}}$ and $\hat{G}_{\text{QCQP}}$ be the QCQP-graph associated to (E.1) and

$$\min_{x \in \mathbb{R}^n}\ \frac{1}{2} x^\top \hat{Q} x + \hat{c}^\top x, \quad \text{s.t.}\ \frac{1}{2} x^\top \hat{P}_i x + \hat{a}_i^\top x \le \hat{b}_i,\ 1 \le i \le m,\ \hat{l} \le x \le \hat{u}, \qquad \text{(E.2)}$$

Suppose that there are no collisions of hash functions or their linear combinations when applying the WL test to $G$ and $\hat{G}$ and there are no strict color refinements in the $L$-th iteration. Since $G$ and $\hat{G}$ are indistinguishable by the WL test, after performing some permutation, there exist $\mathcal{I} = \{I_1, I_2, \ldots, I_s\}$ and $\mathcal{J} = \{J_1, J_2, \ldots, J_t\}$ that are partitions of $\{1, 2, \ldots, m\}$ and $\{1, 2, \ldots, n\}$, respectively, such that the followings hold:

- $C_i^{L,V} = C_{i'}^{L,V}$ if and only if $i, i' \in I_p$ for some $p \in \{1, 2, \ldots, s\}$.

- $C_i^{L,V} = \hat{C}_{i'}^{L,V}$ if and only if $i, i' \in I_p$ for some $p \in \{1, 2, \ldots, s\}$.

- $\hat{C}_i^{L,V} = \hat{C}_{i'}^{L,V}$ if and only if $i, i' \in I_p$ for some $p \in \{1, 2, \ldots, s\}$.

- $C_j^{L,W} = C_{j'}^{L,W}$ if and only if $j, j' \in J_q$ for some $q \in \{1, 2, \ldots, t\}$.

- $C_j^{L,W} = \hat{C}_{j'}^{L,W}$ if and only if $j, j' \in J_q$ for some $q \in \{1, 2, \ldots, t\}$.

- $\hat{C}_j^{L,W} = \hat{C}_{j'}^{L,W}$ if and only if $j, j' \in J_q$ for some $q \in \{1, 2, \ldots, t\}$.

The followings hold by the same arguments as in the proof of Theorem A.2:

- $b_i = \hat{b}_i$ and is constant over $i \in I_p$, for any $p \in \{1, 2, \ldots, s\}$.

- $(c_j, l_j, u_j) = (\hat{c}_j, \hat{l}_j, \hat{u}_j)$ and is constant over $j \in J_q$ for any $q \in \{1, 2, \ldots, t\}$.

- For any $p \in \{1, 2, \ldots, s\}$ and $q \in \{1, 2, \ldots, t\}$, $\sum_{j \in J_q} A_{ij} = \sum_{j \in J_q} \hat{A}_{ij}$ and is constant over $i \in I_p$.

- For any $p \in \{1, 2, \ldots, s\}$ and $q \in \{1, 2, \ldots, t\}$, $\sum_{i \in I_p} A_{ij} = \sum_{i \in I_p} \hat{A}_{ij}$ and is constant over $j \in J_q$.

- For any $q, q' \in \{1, 2, \ldots, t\}$, $\sum_{j' \in J_{q'}} Q_{jj'} = \sum_{j' \in J_{q'}} \hat{Q}_{jj'}$ and is constant over $j \in J_q$.

Fix $p \in \{1, 2, \ldots, s\}$ and $q, q' \in \{1, 2, \ldots, t\}$. For any $j, j' \in J_q$, we have

$$C_j^{L,W} = C_{j'}^{L,W}$$
$$\implies \sum_{i \in V} \text{HASH}\left(C_i^{L-1,V}, C_{ij}^{L-1,E}\right) = \sum_{i \in V} \text{HASH}\left(C_i^{L-1,V}, C_{ij'}^{L-1,E}\right)$$
$$\implies \left\{\left\{ C_{ij}^{L,E} : i \in I_p \right\}\right\} = \left\{\left\{ C_{ij'}^{L,E} : i \in I_p \right\}\right\}$$
$$\implies \left\{\left\{ \sum_{j'' \in W} (P_i)_{jj''} \text{HASH}(C_{j''}^{L-1,W}) : i \in I_p \right\}\right\}$$
$$= \left\{\left\{ \sum_{j'' \in W} (P_i)_{j'j''} \text{HASH}(C_{j''}^{L-1,W}) : i \in I_p \right\}\right\}$$
$$\implies \left\{\left\{ \sum_{j'' \in J_{q'}} (P_i)_{jj''} : i \in I_p \right\}\right\} = \left\{\left\{ \sum_{j'' \in J_{q'}} (P_i)_{j'j''} : i \in I_p \right\}\right\}$$
$$\implies \sum_{j'' \in J_{q'}} \sum_{i \in I_p} (P_i)_{jj''} = \sum_{j'' \in J_{q'}} \sum_{i \in I_p} (P_i)_{j'j''}.$$

One can do a similar analysis for $C_j^{L,W} = \hat{C}_{j'}^{L,W}$ and $\hat{C}_j^{L,W} = \hat{C}_{j'}^{L,W}$ where $j, j' \in J_q$. This concludes that

$$\sum_{j' \in J_{q'}} \sum_{i \in I_p} (P_i)_{jj'} = \sum_{j' \in J_{q'}} \sum_{i \in I_p} (\hat{P}_i)_{jj'}$$

is constant over $j \in J_q$.

Let $x \in \mathbb{R}^n$ be any feasible solution to (E.1) and define $\hat{x} \in \mathbb{R}^n$ via $\hat{x}_j = y_q = \frac{1}{|J_q|} \sum_{j' \in J_q} x_{j'}$ for $j \in J_q$. For any $p \in \{1, 2, \ldots, s\}$, it follows from

$$\frac{1}{2} x^\top P_i x + a_i^\top x \leq b_i, \quad i \in I_p,$$

and Lemma A.4 that

$$\frac{1}{I_p} \sum_{i \in I_p} \hat{b}_i = \frac{1}{I_p} \sum_{i \in I_p} b_i \geq \frac{1}{2} x^\top \left(\frac{1}{|I_p|} \sum_{i \in I_p} P_i\right) x + \left(\frac{1}{I_p} \sum_{i \in I_p} a_i\right)^\top x$$
$$\geq \frac{1}{2} \hat{x}^\top \left(\frac{1}{|I_p|} \sum_{i \in I_p} P_i\right) \hat{x} + \left(\frac{1}{I_p} \sum_{i \in I_p} a_i\right)^\top \hat{x} = \frac{1}{2} \hat{x}^\top \left(\frac{1}{|I_p|} \sum_{i \in I_p} \hat{P}_i\right) \hat{x} + \left(\frac{1}{I_p} \sum_{i \in I_p} \hat{a}_i\right)^\top \hat{x}.$$

Note that for any $i, i' \in I_p$ and any $q, q' \in \{1, 2, \ldots, t\}$, we have

$$\hat{C}_i^{L,V} = \hat{C}_{i'}^{L,V}$$

$$\implies \sum_{j \in W} \text{HASH}\left(\hat{C}_j^{L-1,W}, \hat{C}_{ij}^{L-1,E}\right) = \sum_{j \in W} \text{HASH}\left(\hat{C}_j^{L-1,W}, \hat{C}_{i'j}^{L-1,E}\right)$$

$$\implies \left\{\left\{\hat{C}_{ij}^{L,E} : j \in J_q\right\}\right\} = \left\{\left\{\hat{C}_{i'j}^{L,E} : j \in J_q\right\}\right\}$$

$$\implies \left\{\left\{\sum_{j' \in W}(\hat{P}_i)_{jj'}\text{HASH}(\hat{C}_{j'}^{L-1,W}) : j \in J_q\right\}\right\}$$

$$= \left\{\left\{\sum_{j' \in W}(\hat{P}_{i'})_{jj'}\text{HASH}(\hat{C}_{j'}^{L-1,W}) : j \in J_q\right\}\right\}$$

$$\implies \left\{\left\{\sum_{j' \in J_{q'}}(\hat{P}_i)_{jj'} : j \in J_q\right\}\right\} = \left\{\left\{\sum_{j' \in J_{q'}}(\hat{P}_{i'})_{jj'} : j \in J_q\right\}\right\}$$

$$\implies \sum_{j \in J_q}\sum_{j' \in J_{q'}}(\hat{P}_i)_{jj'} = \sum_{j \in J_q}\sum_{j' \in J_{q'}}(\hat{P}_{i'})_{jj'}.$$

Therefore, it holds that

$$\frac{1}{2}\hat{x}^\top \left(\frac{1}{|I_p|}\sum_{i' \in I_p}\hat{P}_{i'}\right)\hat{x} = \frac{1}{2}\hat{x}^\top \hat{P}_i \hat{x}, \quad \forall\, i \in I_p,$$

and hence that

$$\frac{1}{2}\hat{x}^\top P_i \hat{x} + \hat{a}_i^\top x \le \hat{b}_i, \quad \forall\, i \in I_p.$$

We thus know that $\hat{x}$ is a feasible solution to (A.3). In addition, we have

$$\frac{1}{2}x^\top Q x + c^\top x \ge \frac{1}{2}\hat{x}^\top Q \hat{x} + c^\top \hat{x} = \frac{1}{2}\hat{x}^\top \hat{Q}\hat{x} + \hat{c}^\top \hat{x},$$

which implies that $\Phi_{\text{obj}}(G_{\text{QCQP}}) \ge \Phi_{\text{obj}}(\hat{G}_{\text{QCQP}})$. The reverse direction $\Phi_{\text{obj}}(G_{\text{QCQP}}) \le \Phi_{\text{obj}}(\hat{G}_{\text{QCQP}})$ is also true and we can conclude that $\Phi_{\text{obj}}(G_{\text{QCQP}}) = \Phi_{\text{obj}}(\hat{G}_{\text{QCQP}})$. $\qquad\square$

**Theorem E.6.** *For any $G_{\text{QCQP}}, \hat{G}_{\text{QCQP}} \in \mathcal{G}_{\text{QCQP}}^{m,n}$ with $Q, \hat{Q}, P_i, \hat{P}_i \succeq 0$, $i \in \{1, 2, \ldots, m\}$ that are feasible and bounded, if $G_{\text{QCQP}} \sim \hat{G}_{\text{QCQP}}$, then there exists some permutation $\sigma_W \in S_n$ such that $\Phi_{\text{sol}}(G_{\text{QCQP}}) = \sigma_W(\Phi_{\text{sol}}(\hat{G}_{\text{QCQP}}))$. Furthermore, if $G_{\text{QCQP}} \overset{W}{\sim} \hat{G}_{\text{QCQP}}$, then $\Phi_{\text{sol}}(G_{\text{QCQP}}) = \Phi_{\text{sol}}(\hat{G}_{\text{QCQP}})$.*

*Proof.* Based on Theorem E.5, Theorem E.6 can be proved by the same arguments as in the proof of Lemma B.4 and Corollary B.7 in Chen et al. (2023a), which is included in the proof of Theorem A.2. $\qquad\square$

**Corollary E.7.** *For any $G_{\text{QCQP}} \in \mathcal{G}_{\text{QCQP}}^{m,n}$ that is feasible and bounded and any $j, j' \in \{1, 2, \ldots, n\}$, if $C_j^{L,W} = C_{j'}^{L,W}$ holds for all $L \in \mathbb{N}_+$ and all hash functions, then $\Phi_{\text{sol}}(G_{\text{QCQP}})_j = \Phi_{\text{sol}}(G_{\text{QCQP}})_{j'}$.*

*Proof.* Let $\hat{G}_{\text{QCQP}}$ be the QCQP-graph obtained from $G_{\text{QCQP}}$ by relabeling $j$ as $j'$ and relabeling $j'$ as $j$. By Theorem E.6, we have $\Phi_{\text{sol}}(G_{\text{QCQP}}) = \Phi_{\text{sol}}(\hat{G}_{\text{QCQP}})$, which implies $\Phi_{\text{sol}}(G_{\text{QCQP}})_j = \Phi_{\text{sol}}(\hat{G}_{\text{QCQP}})_j = \Phi_{\text{sol}}(G_{\text{QCQP}})_{j'}$. $\qquad\square$

## F    IMPLEMENTATION DETAILS AND ADDITIONAL NUMERICAL RESULTS

In this section, we explain how we formulate the optimization problems used in the numerical experiments and how to randomly generate problem instances. We mainly follow the settings of OSQP (Stellato et al., 2020) with slight modifications.

### F.1    RANDOM LCQP AND MI-LCQP INSTANCE GENERATION

**Generic LCQP and MI-LCQP generation.** For all instances generated and used in our numerical experiments, we set $m = 10$ and $n = 50$, which means each instance contains 10 constraints and 50 variables. The sampling schemes of problem components are described below.

- Matrix $Q$ in the objective function. We sample sparse, symmetric and positive semidefinite $Q$ using the `make_sparse_spd_matrix` function provided by the `scikit-learn` Python package, which imposes sparsity on the Cholesky factor. We set the `alpha` value to 0.95 so that there will be around $10\%$ non-zero elements in the resulting $Q$ matrix.

- The coefficients $c$ in the objective function: $c_j \sim \mathcal{N}(0, 0.1^2)$.

- The non-zero elements in the coefficient matrix: $A_{ij} \sim \mathcal{N}(0, 1)$. The coefficient matrix $A$ contains 100 non-zero elements. The positions are sampled randomly.

- The right hand side $b$ of the linear constraints: $b_i \sim \mathcal{N}(0, 1)$.

- The constraint types $\circ$. We first sample equality constraints following the Bernoulli distribution $Bernoulli(0.3)$. Then other constraints takes the type $\leq$. Note that this is equivalent to sampling $\leq$ and $\geq$ constraints separately with equal probability, because the elements in $A$ and $b$ are sampled from symmetric distributions.

- The lower and upper bounds of variables: $l_j, u_j \sim \mathcal{N}(0, 10^2)$. We swap their values if $l_j > u_j$ after sampling.

- (MI-LCQP only) The variable types are randomly sampled. Each type (*continuous* or *integer*) occurs with equal probability.

After instance generation is done, we collect labels, i.e., the optimal objective function values and optimal solutions, using one of the commercial solvers.

**LCQP instance generation for generalization experiments.** In this setting, we only sample different coefficients $c$ for different LCQP instances. We sample other components only once, i.e., $Q$, $A$, $b$, $l$, $u$ and $\circ$ in (2.1), and keep them constant and shared by all instances. We also slightly adjust the distributions from which these components are sampled as described below.

- Matrix $Q$. We follow the same sampling scheme as above.

- The coefficients $c$ in the objective function: $c_j \sim \mathcal{N}(0, 1/n)$.

- The non-zero elements in the coefficient matrix: $A_{ij} \sim \mathcal{N}(0, 1/n)$. The coefficient matrix $A$ contains 100 non-zero elements. The positions are sampled randomly.

- The right hand side $b$ of the linear constraints: $b_i \sim \mathcal{N}(0, 1/n)$.

- The constraint types $\circ$. We follow the same sampling scheme as above.

- The lower and upper bounds of variables: $l_j, u_j \sim \mathcal{N}(0, 1)$. We swap their values if $l_j > u_j$ after sampling.

For the generalization experiments, we first generate 25,000 LCQP instances for training, and then take the first 100/500/25,00/5,000/10,000 instances to form the smaller training sets. This ensures that the smaller training sets are subsets of the larger sets. The validation set contains 1,000 instances that are generated separately.

**Portfolio optimization formulation and instance generation.** The portfolio optimization problems are formulated as below.

$$\min_{x,y} \quad \frac{1}{2} x^\top D x + \frac{1}{2} y^\top y - \mu^\top x \tag{F.1}$$

$$\text{s.t.} \quad y = Fx, \quad \mathbf{1}^\top x = 1, \quad x \geq 0$$

Here $x \in \mathbb{R}^s$ and $y \in \mathbb{R}^t$ are the optimization variables, $D \in \mathbb{R}^{s \times s}$ is a diagonal matrix with non-negative diagonal elements, $F \in \mathbb{R}^{t \times s}$ is the factor modeling matrix. We generate portfolio optimization instances following the scheme below.

- We set $s = 50$ and $t = 5$, resulting in LCQP instances with $m = 6$ constraints and $n = 55$ variables.

- The diagonal elements of $D$ are independently sampled from uniform distribution: $D_{ii} \sim U(0, \sqrt{t})$. $D$ is then used to form the matrix $Q = \begin{pmatrix} D & \\ & I_t \end{pmatrix}$.

- The coefficients $\mu$ in the objective function: $\mu_j \sim \mathcal{N}(0, 1)$.

- The non-zero elements in the factor modeling matrix $F$: $F_{ij} \sim \mathcal{N}(0, 1)$. The coefficient matrix $F$ contains 25 non-zero elements. The positions are sampled randomly.

**SVM optimization formulation and instance generation.** The support vector machine optimization problems are formulated as below.

$$\min_{x,t} \quad \frac{1}{2} x^\top x + \lambda \mathbf{1}^\top t \tag{F.2}$$
$$\text{s.t.} \quad t \geq \text{diag}(y) D x + \mathbf{1}, \quad t \geq 0$$

Here $x \in \mathbb{R}^s$ and $t \in \mathbb{R}^t$ are the optimization variables, $D \in \mathbb{R}^{t \times s}$ is the data matrix, $y \in \mathbb{R}^t$ is the binary label vector, and $\lambda$ is a hyperparameter which we set to $1/2$. We generate SVM optimization instances following the scheme below.

- We set $s = 5$ and $t = 50$.

- The non-zero elements in the data matrix $D$: $D_{ij} \sim \mathcal{N}(-0.1, 0.1)$ for $i \leq t/2$; $D_{ij} \sim \mathcal{N}(0.1, 0.1)$ otherwise. The coefficient matrix $D$ contains 100 non-zero elements. The positions are sampled randomly.

- The binary label vector $y$: $y_i = -1$ for $i \leq t/2$; $y_i = 1$ otherwise.

### F.2 DETAILS OF GNN IMPLEMENTATION

We implement GNN with Python 3.9 and TensorFlow 2.16.1 (Abadi et al., 2016). Our implementation is built by extending the GNN implementation in Gasse et al. (2019).[7] The embedding mappings $f_0^V, f_0^W$ are parameterized as linear layers followed by a non-linear activation function; $\{f_l^V, f_l^W, g_l^V, g_l^W, g_l^Q\}_{l=1}^L$ and the output mappings $r_1, r_2$ are parameterized as 2-layer multi-layer perceptrons (MLPs) with respective learnable parameters. The parameters of all linear layers are initialized as orthogonal matrices. We use ReLU as the activation function.

In our experiments, we train GNNs with embedding sizes of 64, 128, 256, 512 and 1,024. We show in Table 2 the number of learnable parameters in the resulting network with each embedding size.

Table 2: Number of learnable parameters in GNN with different embedding sizes.

| Embedding size | Number of parameters |
|---|---|
| 64 | 112,320 |
| 128 | 445,824 |
| 256 | 1,776,384 |
| 512 | 7,091,712 |
| 1,024 | 30,436,352 |

---

[7]See https://github.com/ds4dm/learn2branch.

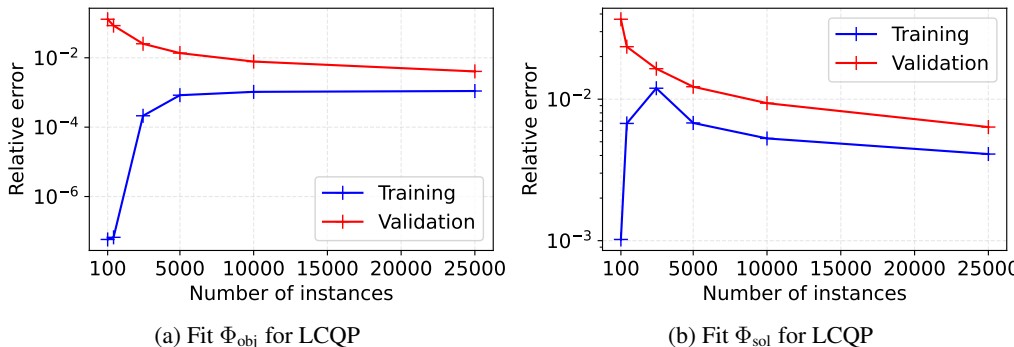

(a) Fit $\Phi_{\text{obj}}$ for LCQP  (b) Fit $\Phi_{\text{sol}}$ for LCQP

Figure 6: Training and validation errors when training GNNs with an embedding size of 512 on different numbers of LCQP problem instances to fit $\Phi_{\text{obj}}$ and $\Phi_{\text{sol}}$.

### F.3 DETAILS OF GNN TRAINING

We adopt Adam (Kingma & Ba, 2014) to optimize the learnable parameters during training. We use an initial learning rate of $5 \times 10^{-4}$ for all networks. We set the batch size to 2,500 or the size of the training set, whichever is the smaller. In each mini-batch, we combine the graphs into one large graph to accelerate training. All experiments are conducted on a single NVIDIA Tesla V100 GPU.

We use mean squared relative error as the loss function, which is defined as

$$L_{\mathcal{G}}(F_W) = \mathbb{E}_{G \sim \mathcal{G}} \left[ \frac{\|F_W(G) - \Phi(G)\|_2^2}{\max(\|\Phi(G)\|, 1)^2} \right], \tag{F.3}$$

where $F_W$ is the GNN, $\mathcal{G}$ is a mini-batch sampled from the whole training set, $G$ is a problem instance in the mini-batch $\mathcal{G}$, and $\Phi(G)$ is the label of instance $G$. During training, we monitor the average training error in each epoch. If the training loss does not improve for 50 epochs, we will half the learning rate and reset the parameters of the GNN to those that yield the lowest training error so far. We observe that this helps to stabilize the training process significantly and can also improve the final loss achieved.

### F.4 GENERALIZATION RESULTS ON LCQP

Figure 6 shows the variations of training and validation errors when training GNNs of an embedding size of 512 on different numbers of LCQP problem instances. We observe similar trends for both prediction tasks, that the generalization gap decreases and the generalization ability improves as more instances are used for training. This result implies the potential of applying trained GNNs to solve QP problems that are unseen during training but are sampled from the same distribution, as long as enough training instances are accessible and the instance distribution is specific enough (in contrast to the generic instances used in experiments of Figure 2 and 3).

### F.5 NUMERICAL RESULTS ON MAROS-MESZAROS TEST SET

To show the fitting ability of GNNs on more realistic QP problems, we train GNNs on the Maros and Meszaros Convex Quadratic Programming Test Problem Set (Maros & Mészáros, 1999), which contains 138 quadratic programs that are designed to be challenging. We apply equilibrium scaling to each problem and also scale the objective function so that the $Q$ matrix will not contain too large elements. We collect the optimal solutions and objective values of the test instances using an open-sourced QP solver called PIQP Schwan et al. (2023), which is benchmarked to achieve best performances on the Maros Meszaros test set among many other solvers (Caron et al., 2024). PIQP solves 136 problem instances successfully, which are then used to train four GNNs with with embedding size of $64, 128, 256, 512$. The training protocol follows the experiments using synthesized QP instances in Section 5.

The results are shown in Figure 7. We observe that while the broad range of numbers of instances in the Maros Meszaros test set caused numerical difficulties for training, GNNs can still be trained

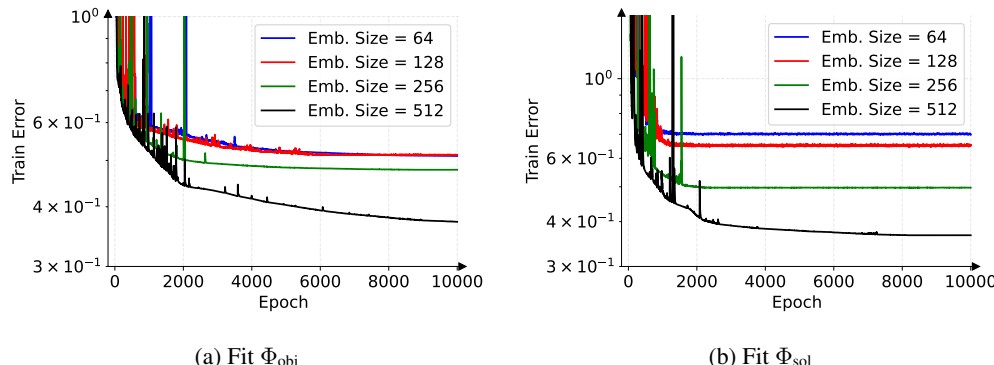

(a) Fit $\Phi_{\text{obj}}$             (b) Fit $\Phi_{\text{sol}}$

Figure 7: Training errors of fitting $\Phi_{\text{obj}}$ and $\Phi_{\text{sol}}$ on the Maros Meszaros test set. We trained four GNNs with embedding size of 64, 128, 256 and 512, respectively.

to fit the objectives and solutions to some extent. And we can observe similar tendency as in the synthesized experiments that the expressive power increases as the model capacity enlarges when we increase the embedding size.

