# OpenReview forum: "Expressive Power of Graph Neural Networks for (Mixed-Integer) Quadratic Programs"
_ICLR.cc/2025/Conference — Submitted to ICLR 2025_

### Official Review · Reviewer_CX5B · 2024-10-28

**Soundness:** 3
**Presentation:** 3
**Contribution:** 3
**Rating:** 8
**Confidence:** 4

**Summary:**

The paper investigates the expressive power of Graph Neural Networks (GNNs) for solving linearly constrained quadratic programming (LCQP) and its mixed-integer variant (MI-LCQP). The authors focus on a key question: Do GNNs possess sufficient expressive power to predict feasibility, optimal objective value, and an optimal solution? The paper provides an affirmative answer for general LCQPs but a negative one for general MI-LCQPs. However, the authors also identify specific subclasses of MI-LCQP for which the answer is affirmative.

**Strengths:**

This paper establishes a solid theoretical foundation for the application of GNNs to LCQPs and MI-LCQPs. The writing is clear.

**Weaknesses:**

1. In order to strengthen the motivation, I would suggest including a survey of previous empirical successes of GNNs in quadratic programming or providing 2-3 specific examples of recent empirical successes of GNNs in quadratic programming, along with brief descriptions of the key results and implications.
2. The paper lacks an analysis of GNN size and depth. These factors are crucial in practical applications.

**Questions:**

Refer to "Weakness"

---

> ### Author Response · Authors · 2024-11-21
> **Our responses to Reviewer CX5B**
>
> We appreciate the reviewer’s encouraging feedback and provide a detailed clarification to address the reviewer’s concerns:
>
> **(Previous empirical success of GNNs for QPs).** Thank you very much for your suggestion. We kindly refer the reviewer to Sections 1 and 2, which provide a review of previous empirical successes. For your convenience, we have highlighted them both in the revised paper and in this response:
>
> > By conceptualizing QP problems as graphs, where all information including coefficients and boundaries are encoded into the graph’s attributes, GNNs can efficiently handle these QP tasks (Nowak et al., 2017; Wang et al., 2020b; 2021; Qu et al., 2021; Gao et al., 2021; Tan et al., 2024; Jung et al., 2022). Such a graph representation is illustrated in Figure 1. This approach leverages key advantages of GNNs: they naturally adapt to varying graph sizes, allowing the same model to be applied to various QP problems, and they are permutation invariant, ensuring consistent outputs regardless of node order.
>
> > To the best of our knowledge, this particular representation is only detailed in Jung et al. (2022), yet it forms the foundation or core module for numerous related studies. For instance, removing nodes in V and their associated edges reduces the graph into the assignment graph used in graph matching problems (Nowak et al., 2017; Wang et al., 2020b; 2021; Qu et al., 2021; Gao et al., 2021; Tan et al., 2024). In these cases, the linear constraints $Ax\circ b$ are typically bypassed by applying the Sinkhorn algorithm to ensure that $x$ meets these constraints. Another scenario involves LP and MILP: removing edges associated with $Q$ simplifies the graph to a bipartite structure, which reduces the LCQP to an LP (Chen et al., 2023a; Fan et al., 2023; Liu et al., 2024; Qian et al., 2024). Further, by incorporating an additional node feature, an approach detailed in Section 4, this bipartite graph is also capable of representing MILP (Gasse et al., 2019; Chen et al., 2023b; Nair et al., 2020; Gupta et al., 2020; Shen et al., 2021; Gupta et al., 2022; Khalil et al., 2022; Paulus et al., 2022; Scavuzzo et al., 2022; Liu et al., 2022b; Huang et al., 2023).
>
> If you believe any critical references or examples have been overlooked, we would be happy to include them.
>
> **(Size and depth of GNNs).** We certainly agree with you that the size and depth of GNNs are crucial in practice and we would like to kindly highlight a result implied by our analysis regarding the depth of GNNs: **All our universal approximation results are true for GNNs with $\mathcal{O}(m+n)$ layers**, since the WL test has at most $\mathcal{O}(m+n)$ iterations with strict color refinement (as mentioned after Algorithm 1). Regarding the size of GNNs, we believe this is an important topic for future research, and we honestly pointed it out in the updated conclusion section.
>
> We hope these clarifications address your concerns. If you have further questions or require additional clarifications, we would be more than happy to provide them.

---

> > ### Comment · Reviewer_CX5B · 2024-11-22
> >
> > Thank you for your detailed response.
> >
> > **(Previous empirical success of GNNs for QPs)** While your explanation highlights the conceptualization of QPs as graphs, it does not directly address my question. I was specifically asking whether there are previous numerical experiments demonstrating that GNNs can effectively solve QP tasks. If any, could you provide such an experiment and briefly summarize their findings in 1-2 sentence? For instance, it is reported by [xxx] that a xxx-layer xxx-size GNN is capable of approximately solving QPs of xxx sizes
> >
> > **(Size and depth of GNNs).**  Could you provide an upper bound on the size of your GNNs? For example, would it be correct to say that your GNNs uses at most $\exp(O(m+n))$ neuros?

---

> > > ### Author Response · Authors · 2024-11-23
> > >
> > > Thank you very much for your reply! Below, we provide a point-by-point response to address your concerns:
> > >
> > > __(Previous empirical success of GNNs for QPs)__ There have indeed been some empirical sucess of GNNs for solving QPs and we would like to mention some for the graph matching probem/quadratic assignment problem (QAP), that is a important family of QPs.
> > >
> > > * [R1] applies GNNs to solve Lawler's QAP [R4] with number of variables up to $150^2$.
> > > * [R2] and [R3] apply GNNs to solve Koopman-Beckmann's QAP [R5] with $256^2$ variables.
> > >
> > > All [R1,R2,R3] use $3$-layer GNNs with hidden dimension being 512/1024/2048, domenstrating strong ability and potential of GNNs for solving QP problems.
> > >
> > > These examples are discussed in the latest revision of our paper (see highlights in Section 1).
> > >
> > > __(Size and depth of GNNs)__ Characterizing the size of GNNs quantitatively is one of our ongoing and future directions, as mentioned in the conclusion of our paper. We would like to mention a preliminary result for LCQPs (though the detailed proof is too long to be included in this reply):
> > > * According to [R6], GNNs can be connected to classic iterative algorithms: with specific parameter settings, a GNN reduces to an iterative algorithm, while parameterizing an iterative algorithm transforms it into a GNN. By this perspective, each iteration of a classic algorithm corresponds to a layer in the GNN, and the size of the GNN can be interpreted as the computational complexity of the iterative algorithm.
> > > * Now let's consider a classic iterative algorithm, PDHG, for convex LCQP [R7], and conceptualize it as a GNN with specific parameters. The single-iteration complexity (or, in the context of GNNs, the number of neurons per layer) is bounded by $\mathcal{O}(\mathrm{nnz}(A)+\mathrm{nnz}(Q)+m+n)$. Additionally, according to [R7], the number of iterations (or layers in the GNN) is bounded by $C(A,Q,b,c) \cdot \mathcal{O}(\log(1/\delta))$, where $\delta$ is the accuracy tolerance (used in our theorems), and $C(A,Q,b,c)$ is a positive number depending on properties of the QP.
> > > * Therefore, if we define a sub-class of convex LCQP: $\\{(A,Q,b,c): C(A,Q,b,c) \leq M\\}$ where $M$ is a constant, the complexity (or the number of neurons in the GNN) for this type of LCQP can be expressed as $\mathcal{O}((\mathrm{nnz}(A)+\mathrm{nnz}(Q)+m+n)\log(1/\delta))$.
> > >
> > > Note that the above results are parallel to the theorems in our paper: there is no one can exactly cover the other. Our theorems provide a precise upper bound on the number of layers, $\mathcal{O}(m+n)$, but do not address size per layer. In contrast, the above results offer an exact bound for the size per layer, though the bound on the number of layers is loose.
> > >
> > > **The above discussions apply only to convex LCQPs** (the same settings with our Theorems 3.3 and 3.4) To the best of our knowledge, there is no fully quantitative bound of GNNs' size for solving general QPs (or even general linear programs). For mixed-integer QPs, unfortunately, we do not have a bound of GNNs' size at this moment, but we expect that it is **at least exponential** in $m$ and $n$ due to the NP-hardness.
> > >
> > > We hope that these answer your question and please let us know if you have further questions.
> > >
> > > __References:__
> > >
> > > [R1] Wang, R., Yan, J., & Yang, X. (2021). Neural graph matching network: Learning lawler’s quadratic assignment problem with extension to hypergraph and multiple-graph matching. IEEE Transactions on Pattern Analysis and Machine Intelligence, 44(9), 5261-5279.
> > >
> > > [R2] Wang, R., Yan, J., & Yang, X. (2019). Learning combinatorial embedding networks for deep graph matching. In Proceedings of the IEEE/CVF international conference on computer vision (pp. 3056-3065).
> > >
> > > [R3] Yu, T., Wang, R., Yan, J., & Li, B. (2020). Learning deep graph matching with channel-independent embedding and hungarian attention. In International conference on learning representations.
> > >
> > > [R4] Lawler, E. L. (1963). The quadratic assignment problem. Management science, 9(4), 586-599.
> > >
> > > [R5] Loiola, E. M., De Abreu, N. M. M., Boaventura-Netto, P. O., Hahn, P., & Querido, T. (2007). A survey for the quadratic assignment problem. European journal of operational research, 176(2), 657-690.
> > >
> > > [R6] Yang, Y., Liu, T., Wang, Y., Zhou, J., Gan, Q., Wei, Z., ... & Wipf, D. (2021). Graph neural networks inspired by classical iterative algorithms. In International Conference on Machine Learning (pp. 11773-11783). PMLR.
> > >
> > > [R7] Lu, H., & Yang, J. (2023). A practical and optimal first-order method for large-scale convex quadratic programming. arXiv preprint arXiv:2311.07710.

---

> > > > ### Comment · Reviewer_CX5B · 2024-11-25
> > > >
> > > > Thank you for your detailed response. My questions are addressed.

---

> > > > > ### Author Response · Authors · 2024-11-29
> > > > >
> > > > > Thank you very much for your encouraging response! We appreciate all your valuable comments.

---

### Official Review · Reviewer_fJ5r · 2024-11-01

**Soundness:** 4
**Presentation:** 4
**Contribution:** 4
**Rating:** 6
**Confidence:** 4

**Summary:**

This work investigates the expressive power of graph neural networks (GNNs) in representing both continuous and discrete quadratic programs (QPs). Specifically, it validates three key capabilities related to the empirical performance of GNNs for QPs: accurately predicting feasibility, optimality, and optimal solutions. By answering these questions, the study bridges the gap between theoretical and empirical aspects, shedding light on the strong performance observed in prior research in this field. The authors also provide sufficient numerical results to support their claims.

**Strengths:**

1. Theoretical foundation: The paper successfully bridges the gap between theoretical predictions and empirical results by demonstrating the capacity of GNNs to represent QPs.
2. Solid theoratical proofs: The authors provide well-founded proofs to support their theoretical claims.
3. Impactful: The discoveries in this work pave the way for further exploration of GNN applications in accelerating solutions to QP problems, holding the potential for significant empirical advancements.
4. The paper is generally well-written and easy to follow.

**Weaknesses:**

The primary concern regarding this work is the specific type of MI-LCQPs that can be represented by GNNs.
Providing examples of application scenarios for this type of problem would help strengthen the authors' claims, as mixed-integer problems with unique optimal solutions do not appear to be very common in practice.

**Questions:**

Please refer to "Weakness".

---

> ### Author Response · Authors · 2024-11-21
> **Our responses to Reviewer fJ5r**
>
> We appreciate the reviewer’s thoughtful feedback and provide a detailed clarification to address the reviewer’s concerns:
>
> **(Application scenarios for MP-tractable and unfoldable MI-LCQPs).** An application scenario for these types of problems is: all the coefficients in the objective function are continuously distributed. With this assumption, we show in Proposition D.3 that MI-LCQPs are unfoldable and MP-tractable with probability 1. This ensures that all theorems presented in our paper can be directly applied. Notably, the dataset used in our experiments, which includes portfolio problems and SVMs, consists entirely of unfoldable and MP-tractable instances. These discussions have been included in Section 4.3 and Appendix D.
>
> **(Our contributions beyond the above scenarios).** Although in some manually crafted datasets (like the counterexamples presented in our paper), "bad" instances (MP-intractable or foldable) may cause GNNs to fail, our theoretical results still offer practical insights. The two proposed criteria, MP-tractability and unfoldability, are both **_computationally efficient to verify_**, with a polynomial complexity detailed in Section 4.3. Practitioners can use these criteria to evaluate their datasets before applying GNNs. Alternatively, if difficulties arise during GNN training on MI-LCQPs, checking for MP-tractability and unfoldability may help identify the issue. Although GNNs have been widely applied to QP tasks (e.g., Nowak et al., 2017; Wang et al., 2020b, 2021; Qu et al., 2021; Gao et al., 2021; Tan et al., 2024; Jung et al., 2022), there remains a significant lack of theoretical studies. Our findings explicitly clarify **_what GNNs can and cannot do for QP tasks_**, provide insights and theoretical guidance in this domain.
>
> **(Assumptions regarding unique optimal solutions).** We would like to kindly argue that we do **NOT** need uniqueness of optimal solutions. For continuous LCQP, we allow multiple optimal solutions, and we show that GNNs can approximate the optimal solution with the smallest $\ell_2$ norm (Definition 2.3). For MI-LCQP, we also allow multiple optimal solutions, and define a total ordering on the optimal solution set, and show that GNNs can approximate the minimal element in the optimal solution set. Discussions have been included in Appendix C.
>
> We hope these clarifications address your concerns. If you have further questions or require additional clarifications, we would be more than happy to provide them. We greatly appreciate it if you could update your evaluation should you find these responses satisfactory.

---

> ### Author Response · Authors · 2024-11-24
> **Need more clarifications?**
>
> Dear Reviewer fJ5r,
>
> Thank you for your insightful comments and constructive feedback. We have carefully reviewed and addressed each of your concerns in our detailed responses.
>
> Given the impending rebuttal deadline, we kindly request your review of our responses to ensure that all issues have been addressed satisfactorily. Should you require further clarification or additional details, we would be happy to provide any necessary explanations.
>
> We truly appreciate your time and effort in reviewing our work.
>
> Best regards.

---

### Official Review · Reviewer_LrCY · 2024-11-02

**Soundness:** 3
**Presentation:** 3
**Contribution:** 3
**Rating:** 6
**Confidence:** 3

**Summary:**

This work explores the expressive power of GNNs for LCQP and MI-LCQP, representing a significant contribution to the field. The authors provide a valuable framework and a theoretical foundation to demonstrate GNNs' capabilities in representing feasibility, optimal objective values, and solutions for LCQPs. Detailed proofs and well-structured analysis are commendable, and the organization of the paper is logical and flows well.

**Strengths:**

The paper offers a strong contribution to understanding the expressive power of Graph Neural Networks (GNNs) for Quadratic Programming (QP), particularly for linearly constrained (LCQP) and mixed-integer (MI-LCQP) problems. The strengths of the paper include:
1. The authors have innovatively connected GNNs with LCQPs and MI-LCQPs, an area that has received limited attention, thereby expanding the potential applications of GNNs in optimization.
2. The proofs are well-structured, and the reliance on the Weisfeiler-Lehman (WL) test to support the expressive power of GNNs is a thoughtful and sound approach.
3. The paper is well-organized, with a logical flow from background concepts to theoretical contributions and then to experimental results.
4. The significance of this work lies in its potential impact on optimization problems where approximate solutions are acceptable.

**Weaknesses:**

Several aspects could be improved for greater applicability and clarity:
1. The assumptions such as MP-tractability and unfoldability in Section 4, as well as the requirements for Q to be symmetric and the mappings to be continuous, might limit the broader applicability of the results in practical scenarios.
2. Certain assumptions, such as the existence of an optimal solution for MI-LCQPs and continuity in mappings, are necessary for the proofs but lack a detailed discussion of their purpose and impact on the proof process. Without understanding why these assumptions are critical, readers might find it difficult to gauge the generalizability and limitations of the theoretical results.
3. The experimental validation relies on randomly generated LCQP and MI-LCQP instances, which may not fully capture the diversity and complexity of real-world problems.
4. While embedding size is explored, there are few ablation studies to test other aspects of the model, such as alternative message-passing mechanisms or different GNN architectures.

**Questions:**

Here are several questions and suggestions for the authors:
1. Could the authors elaborate on the situations where MP-tractability and unfoldability assumptions might not hold? Are there practical examples where these conditions might fail, and how would that impact the use of GNNs in those cases?
2. The paper includes several assumptions, such as Q being symmetric and the mappings being continuous. Could the authors explain why these assumptions are necessary and how they influence the proof process?
3. Beyond the testing on synthetic data, can the authors discuss the limitations or difficulties of applying for the GNN in practical LCQP and MI-LCQP problems, e.g., applications or optimizations in power system operations or financial portfolio?
4. Could the authors provide more experimental insights into how different message-passing mechanisms or GNN architectures might affect performance?
5. The WL test is central to the theoretical results but may be challenging for some readers. Can the authors provide more intuitive insights or examples to illustrate how the WL test is applied in this context?

---

> ### Author Response · Authors · 2024-11-21
> **Our response to Reviewer LrCY (Part I)**
>
> We appreciate the reviewer’s thoughtful feedback and provide a detailed clarification to address the reviewer’s concerns:
>
> **(MP-tractability and Unfoldability: Addressing Weakness 1 and Question 1).** We kindly refer the reviewer to Section 4.3, where we provide practical insights regarding MP-tractability and unfoldability. Below is a summary of the key points, along with some addtional discussions during this rebuttal:
> - **(Occurance of MP-tractability and unfoldability).** In practice, the frequency of MP-tractable and unfoldable instances largely depends on the dataset, particularly **_the level of symmetry_** in the data. When there is symmetry in MI-LCQP, it becomes foldable; and higher symmetry increases the risk of being MP-intractable.
> Fortunately, **_unfoldable and MP-tractable instances make up the majority of the MI-LCQP set_** (shown in Appendix D). The dataset used in our experiments, which includes synthetic MI-LCQPs, portfolio problems, and SVMs, consists entirely of unfoldable and MP-tractable instances. However, it's important to note that in some challenging, artificially created datasets like MIPLIB 2017 (Gleixner et al., 2021), about 1/4 of the examples exhibit significant symmetry in half of the variables.
> - **(Impact of symmetry on the use of GNNs).** When a dataset contains a significant proportion of “bad” instances (MP-intractable or foldable), achieving high training accuracy becomes impossible, as discussed in Section 4.1. Regardless of their size, GNNs cannot distinguish two distinct MI-LCQPs as symmetry makes them indistinguishable. As a result, a GNN that performs well on one such instance will inevitably fail on the other.
> - **(Our contributions and practical insights).** Our theoretical results provide practical insights, as the two proposed criteria, MP-tractability and unfoldability, are both efficient to verify with a polynomial complexity detailed in Section 4.3. Practitioners can use these criteria to evaluate their datasets before applying GNNs. Alternatively, if difficulties arise during GNN training on MI-LCQPs, checking for MP-tractability and unfoldability may help identify the issue. Although GNNs have been widely applied to QP tasks (e.g., Nowak et al., 2017; Wang et al., 2020b, 2021; Qu et al., 2021; Gao et al., 2021; Tan et al., 2024; Jung et al., 2022), there remains a significant lack of theoretical studies. Our findings explicitly clarify **_what GNNs can and cannot do for QP tasks_**, and provide theoretical insights in this domain.
> - **(How to handle bad instances?).** As we discussed, bad instances typically exhibit strong symmetry. To fix this issue, there are two potential approaches: **(I) Adding Additional Features:** Incorporating additional features, such as random features, can help differentiate nodes in a symmetric graph. For instance, if two nodes have identical features, adding a random feature ensures they are no longer identical (and thus not symmetric). Such techniques have been explored in the GNN literature for various graph tasks [R1] **(II) Using Higher-order GNNs:** Higher-order GNNs can separate nodes that standard message-passing GNNs consider identical, thereby enhancing the expressive power of the model [R2]. Addressing symmetric graph data remains an active and important area in the GNN community, making this a promising direction for future research on MI-LCQPs. We have included the above discussion in our revision to highlight this point. (Section 4.3)

---

> ### Author Response · Authors · 2024-11-21
> **Our response to Reviewer LrCY (Part II)**
>
> **(Other Assumptions: Addressing Weakness 2 and Question 2).** We respectfully argue that all the assumptions mentioned by the reviewer are reasonable and align with common practices in the QP and GNN communities. Below are the details for each assumption:
> - **(Symmetry of Q).** This assumption is standard in QP studies, as any QP can be equivalently transformed into another QP with a symmetric Q. For example,
> $$\begin{bmatrix}x_1 & x_2\end{bmatrix}\begin{bmatrix}0 & 1 \\\\
>  2 & 0\end{bmatrix}
> \begin{bmatrix}x_1 \\\\
>  x_2\end{bmatrix}
> = 3x_1x_2
> = \begin{bmatrix}x_1 & x_2\end{bmatrix}
> \begin{bmatrix}0 & 2 \\\\
>  1 & 0\end{bmatrix}
> \begin{bmatrix}x_1 \\\\
>  x_2\end{bmatrix}
> = \begin{bmatrix}x_1 & x_2\end{bmatrix}
> \begin{bmatrix}0 & 1.5 \\\\ 1.5 & 0\end{bmatrix}
> \begin{bmatrix}x_1 \\\\ x_2\end{bmatrix}$$
> In general, $x^\top Q x = x^\top Q^\top x$ and hence $x^\top Q x = \frac{1}{2} x^\top (Q+Q^\top) x$ where $(Q+Q^\top)/2$ must be symmetric.
> - **(Continuity of mappings in GNNs).** The learnable mappings $f_l^V,f_l^W, g_l^V,f_l^W,g_l^Q,r_1,r_2$ are all assumed to be continuous, as this is a common practice in the analysis of GNNs (Azizian & Lelarge, 2021; Geerts & Reutter, 2022; Zhang et al., 2023). In practice, we use MLPs with ReLU (which are continuous) to parameterize these mappings. Many other parameterization choices, such as RBF networks, polynomial functions or sin/cos functions, are also continuous. Thus, the continuity of these mappings aligns well with most practical implementations.
> - **(Target mappings).** The target mappings $\Phi_{\mathrm{feas}},\Phi_{\mathrm{obj}},\Phi_{\mathrm{sol}}$ given in Definition 2.3 are **NOT** assumed to be continuous. In particular, they are measurable and by Lusin's theorem, there exists a closed/compact domain such that the target mappings constrained on the domain is continuous.
> - **(Existence of optimal solutions).** In general, we do **NOT** assume that LCQPs or MI-LCQPs have optimal solutions. In Theorems 3.2 and 4.6, the QPs are even possibly infeasible (in which case optimal solutions do not exist) and we prove that GNNs can classify feasibility under certain assumptions. In Theorems 3.3 and 4.7, the QPs are possibly unbounded (in which case optimal solutions do not exist), and we prove that GNNs can classify boundedness under certain assumptions. Only in Theorems 3.4 and 4.9, we assume that the QPs have an optimal solution, which is a natural assumption. Since the goal is to show that GNNs can approximate the optimal solution of QPs, it is reasonable to require the existence of a solution. If no solution exists, there is nothing to approximate, correct?
>
> **(Additional Experiments: Addressing Weakness 3 and Question 3).** To address this concern, we train GNNs on the Maros and Meszaros Convex Quadratic Programming Test Problem Set, which contains 138 quadratic programs that are designed to be challenging. We apply equilibrium scaling to each problem and also scale the objective function so that the $Q$ matrix will not contain too large elements. We collecte the optimal solutions and objective values of the test instances using an open-sourced QP sovler called PIQP [R4], which is benchmarked to achieve best performances on the Maros Meszaros test set among many other solvers [R5]. PIQP solves 136 problem instances successfully, which are then used to train four GNNs with with embedding size of $64,128,256,512$.
>
> **_The results are included in the updated draft in Appendix F.5._** We observe that while the broad range of numbers of instances in the Maros Meszaros test set caused numerical difficulties for training, GNNs can still be trained to fit the objectives and solutions to some extent. And we can observe similar tendency as in the synthesized experiments that the expressive power increases as the model capacity enlarges when we increase the embedding size.
>
> **(Various GNN architectures: Addressing Weakness 4 and Question 4).** As the first pave towards the theretical analysis for GNN on QPs, we choose the most fundamental architecture of GNN. To address the reviewer's concern, we provide the following discussion on alternative GNN architectures: In our work we use the sum aggregation, and all results are still valid for the weighted average aggregation. In particular, all our proofs (such as the proof of Theorem A.2) hold almost verbatimly for the average aggregation. The attention aggregation (Veličković et al, 2017) has stronger separation power, which implies that all universal approximation results still hold. Moreover, all the counter examples for MI-LCQPs work for every aggregation approach, since the color refinement in Algorithm 1 is implemented on multisets, with separation power stronger than or equal to all aggregations of neighboring information. We have included the above discussion in our updated draft (at the end of Appendix C).

---

> ### Author Response · Authors · 2024-11-21
> **Our response to Reviewer LrCY (Part III)**
>
> **(Illustration of the WL test: Addressing Question 5).** We kindly refer the reviewer to Appendix D.1 (Figure 5), where we provide an illustration of the WL test. In this section, we include a small example to demonstrate the process of the WL test and how the node colors are updated. Due to the limitation of pages, it may be challenging to incorporate this content into the main text, we have added **_a pointer to this example_** in title of Algorithm 1 for the readers' convenience. Please see our revised paper for details.
>
>
> We hope these clarifications address your concerns. If you have further questions or require additional clarifications, we would be more than happy to provide them. We greatly appreciate it if you could update your evaluation should you find these responses satisfactory.
>
> ------------------------
> ## References added in rebuttal
>
> [R1] Sato et al. "Random features strengthen graph neural networks." Proceedings of the 2021 SIAM international conference on data mining (SDM). 2021.
>
> [R2] Morris, et al. "Weisfeiler and leman go neural: Higher-order graph neural networks." Proceedings of the AAAI conference on artificial intelligence. 2019.
>
> [R3] Veličković, et al. "Graph attention networks." arXiv preprint arXiv:1710.10903. 2017.
>
> [R4] Schwan R, Jiang Y, Kuhn D, Jones CN. PIQP: A Proximal Interior-Point Quadratic Programming Solver. In2023 62nd IEEE Conference on Decision and Control (CDC) 2023 Dec 13 (pp. 1088-1093). IEEE.
>
> [R5] https://github.com/qpsolvers/maros_meszaros_qpbenchmark/

---

> ### Author Response · Authors · 2024-11-24
> **Need more clarifications?**
>
> Dear Reviewer LrCY,
>
> Thank you for your insightful comments and constructive feedback. We have carefully reviewed and addressed each of your concerns in our detailed responses.
>
> Given the impending rebuttal deadline, we kindly request your review of our responses to ensure that all issues have been addressed satisfactorily. Should you require further clarification or additional details, we would be happy to provide any necessary explanations.
>
> We truly appreciate your time and effort in reviewing our work.
>
> Best regards.

---

> > ### Comment · Reviewer_LrCY · 2024-11-25
> >
> > Thank you for your detailed response. My questions are addressed.

---

> > > ### Author Response · Authors · 2024-11-29
> > >
> > > Thank you very much for your encouraging response! We appreciate all your valuable comments.

---

### Official Review · Reviewer_aVvJ · 2024-11-04

**Soundness:** 3
**Presentation:** 2
**Contribution:** 3
**Rating:** 6
**Confidence:** 4

**Summary:**

In this article the authors present new regarding the expressivity of GNNs for Quadratic Programming, and more specifically for Quadratic Programs (QPs) with linear constraints (continuous and mixed integer). In particular, they explicit a class of QPs (called MP-tractable) such that for each QP in the class, there exists a GNN that can provide accurate information about the feasability and the optimal value of the QP. The main results are stated in Theorem 3.3 and Theorem 3.4, and hold in probability, given a distribution on the considered class of QPs.

**Strengths:**

- The paper has complete introduction and clearly states the contribution.

- The authors combine results in Approximation Theory and properties of GNNs to answer fundamental questions about their expressivity in the framework of QPs.

- The authors present positive and negative results regarding this expressive power.

- The authors have added numerical experiments to illustrate and support their claims.

Overall, I find the motivations of the study and the results interesting. In particular the positive results in the positive semi-definite case interesting. However, I have some reservations as detailed next.

**Weaknesses:**

- The MP-tractable class is not very insightful or practical to work with. Can the authors elaborate if the criterion they propose is efficient to be tested? (as one of the initial motivations in the introduction was to improve efficiency in solving QPs?)

- The presentation of the technical parts could be shortened for clarity.  Cf. also my two questions below.

- The notations and statements in the proofs of the appendix are not very well chosen, making them more difficult to follow for ``superficial'' reasons. The authors do not include a lot of their proofs, very often coming from other papers.

**Questions:**

- Can the authors elaborate on the difference of second part of Theorem 3.3 and Theorem 3.4? Isn't it true that Theorem 3.4 implies the second part of Theorem 3.3 simply by continuity of the evaluation of the objective function?

- Same question applies to Th. 4.7 vs Th. 4.9.

- (Due to weakness 1) In which part exactly did the authors prove the separation property claimed page 5?

---

> ### Author Response · Authors · 2024-11-21
> **Our response to Reviewer aVvJ**
>
> We appreciate the reviewer’s thoughtful feedback and provide a detailed clarification to address the reviewer’s concerns:
>
> **(MP-tractable Class).** We kindly refer the reviewer to Section 4.3, where we provide practical insights and discuss the complexity of verifying MP-tractability. Below is a summary of the key points:
> - **(Verifying MP-tractability).** In practice, both MP-tractability can be efficiently verified by applying the WL test (Algorithm 1), which requires at most $\mathcal{O}(m+n)$ iterations. The complexity of each iteration is bounded by the number of edges in the graph, which, in our context, is the number of nonzeros in matrices $A$ and $Q$: $\text{nnz}(A)+\text{nnz}(Q)$. Therefore, the overall complexity of Algorithm 1 is $\mathcal{O}((m+n)\cdot (\text{nnz}(A)+\text{nnz}(Q)))$. After running Algorithm 1, MP-tractability can be directly verified using Definition 4.4. It is worth noting that although MI-LCQP is generally NP-hard, verifying MP-tractability is significantly simpler than solving the MI-LCQP itself.
> - **(Frequency of MP-tractable instances).** In practice, the occurrence  of MP-tractable instances largely depends on the dataset, particularly the level of symmetry (regarding symmetry, we provide definition and examples in Section 4.3) within the MI-LCQPs: higher symmetry increases the risk of being MP-intractable. Fortunately, MP-tractable instances make up the majority of the MI-LCQP set (shown in Appendix D). The dataset used in our experiments, which includes synthetic MI-LCQPs, portfolio problems, and SVMs, consists entirely of MP-tractable instances, so that the theorems in our paper can be directly validated. However, it's important to note that in some challenging, artificially created datasets like MIPLIB 2017 (Gleixner et al., 2021), about 1/4 of the examples exhibit significant symmetry.
>
> **(Potentially Shortening Presentation).** Yes, there is a clear difference between the second part of Theorem 3.3 and Theorem 3.4, which lies in the output layer of the GNNs. In Theorem 3.3, The GNN produces a graph-level output (i.e., a single real number for the entire graph). Accordingly, the GNN is selected from ${\mathcal{F}}\_\mathrm{LCQP}$. In contrast, in Theorem 3.4, The GNN produces a node-level output (i.e., a real number for each node). In this case, the GNN is selected from the space $\mathcal{F}^W_{\mathrm{LCQP}}$. Relevant definitions can be found at the beginning of Page 4 and in Definition 2.2. This distinction similarly applies to Theorems 4.7 and 4.9.
>
> **(Handwaving Claims in Proofs).** We have updated our appendix for the sake of completeness and highlighted the modifications, please kindly refer to our revision.
>
> **(Proof of Separation Powers).** Theorems A.2 and A.3 in Appendix A establish the separation properties referenced on Page 5. In these theorems, we use the notation $G_{\mathrm{LCQP}} \sim \hat{G}\_{\mathrm{LCQP}}$ to indicate that $G_{\mathrm{LCQP}}$ and $\hat{G}_{\mathrm{LCQP}}$ are indistinguishable by the WL test (and, equivalently, by any GNNs). We have updated our draft to further clarify the connection between the discussions on Page 5 and the results in Appendix A.
>
> We hope these clarifications address your concerns. If you have further questions or require additional clarifications, we would be more than happy to provide them. We greatly appreciate it if you could update your evaluation should you find these responses satisfactory.

---

> ### Author Response · Authors · 2024-11-24
> **Need more clarifications?**
>
> Dear Reviewer aVvJ,
>
> Thank you for your insightful comments and constructive feedback. We have carefully reviewed and addressed each of your concerns in our detailed responses.
>
> Given the impending rebuttal deadline, we kindly request your review of our responses to ensure that all issues have been addressed satisfactorily. Should you require further clarification or additional details, we would be happy to provide any necessary explanations.
>
> We truly appreciate your time and effort in reviewing our work.
>
> Best regards.

---

> > ### Comment · Reviewer_aVvJ · 2024-11-24
> >
> > Thanks for your detailed answer. I am satisfied with how the authors addressed my questions, and raised my score accordingly.
> > However, I am not raising my score further for the two following reasons: i) overall, the results are not so insightful and their applicability are limited, ii) the writing of the article and presentation of the results could be improved.

---

> > > ### Author Response · Authors · 2024-11-29
> > >
> > > Thank you very much for your encouraging response and for raising your assessment!
> > >
> > > Below, we clarify and expand on our contributions, insights, and presentations to ensure the reviewer’s concerns are thoroughly addressed:
> > >
> > > **(Insights).** We believe that our paper provide enough insights to practationers, particularly in understanding the capabilities and limitations of GNNs for QP tasks. **_For continuous LCQPs, our theorems show that practitioners need not worry about the expressive power of GNNs_**, as they act as universal approximators. **_However, for mixed-integer QPs, our theorems show that practitioners must carefully evaluate the expressive power of GNNs before application._**
> > >
> > > To facilitate this evaluation, we introduce **_two useful criteria_**: MP-tractability and unfoldability. Both are computationally efficient to verify, with polynomial complexity detailed in Section 4.3. Practitioners can use these criteria to evaluate their datasets before applying GNNs. Alternatively, if difficulties arise during GNN training on MI-LCQPs, checking for MP-tractability and unfoldability may help identify the issue.
> > >
> > > Although GNNs have been widely applied to QP tasks (e.g., Nowak et al., 2017; Wang et al., 2020b, 2021; Qu et al., 2021; Gao et al., 2021; Tan et al., 2024; Jung et al., 2022), there remains a significant lack of theoretical studies. Our findings explicitly clarify **_what GNNs can and cannot do for QP tasks_**, and provide theoretical insights in this domain.
> > >
> > > **(Applications).** As we discussed in Section 1, QP has extensive applications across domains such as graph matching, portfolio optimization, and dynamic control (Vogelstein et al., 2015; Markowitz, 1952; Rockafellar, 1987), and recent literature empirically show GNNs' great potential in QP tasks (e.g., Nowak et al., 2017; Wang et al., 2020b, 2021; Qu et al., 2021; Gao et al., 2021; Tan et al., 2024; Jung et al., 2022). Building on this empirical evidence, our theorems offer useful criteria and practical insights, supporting a broad range of applications for GNNs in solving QPs.
> > >
> > > **(Presentations).** To further address the reviewer's concern, we **_provide additional discussion on the connection between Theorems 3.3 and 3.4_** (as well as Theorems 4.7 and 4.9). Specifically, we would like to include the following clarification in our revision:
> > > > While Theorem 3.4 does not directly imply the second part of Theorem 3.3, the second part of Theorem 3.3 can still be derived from Theorem 3.4. By appending an additional aggregation layer to the GNN in Theorem 3.4, the modified GNN can fulfill the requirements of Theorem 3.3, leading to the result stated in 3.3.

---

### Official Review · Reviewer_y6ua · 2024-11-04

**Soundness:** 3
**Presentation:** 3
**Contribution:** 2
**Rating:** 3
**Confidence:** 4

**Summary:**

The paper investigates the expressive power of GNNs in representing key properties of quadratic programs (QP) and mixed-integer QP (MI-QP). It provides a theoretical foundation for using GNNs to approximate feasibility, optimal objective values, and solutions for QPs. The study includes both theoretical proofs and experimental results.

**Strengths:**

1. **Importance of GNN for QP**: The paper tackles an important problem by providing a theoretical foundation on the representational power of GNNs for quadratic programming, which is relevant to various practical applications.
2. **Clarity of Writing**: The paper is well-structured, making it easy to follow. The clear explanations and organized framework enhance readability and accessibility.

**Weaknesses:**

1. **Novelty and Limited Contributions**: While the paper aims to establish theoretical insights into GNNs' representation capabilities for QPs, much of its theoretical approach relies on Chen et al. (2023a, b), which studies the representation power of GNNs for LP and MILP. The paper’s theoretical contributions to the QP domain are thus incremental and may lack non-trivial novelty.

2. **Empirical Validation**: Although the paper includes some experimental validation, the scope is limited. A stronger empirical component using QP benchmark datasets is needed to substantiate the claims further. Additionally, the current experiments focus mainly on synthetic data, which may not sufficiently represent real-world QP applications.

3. **Handwaving Claims in Proofs**: Several proof sections lack rigor and rely on handwaving arguments. For example, statements like "xxx can be proved following the same lines as in the proof of Theorem xxx in Chen et al. (2023a), with trivial modifications to generalize results for LP-graphs to the LCQP setting," fail to provide a rigorous and self-contained argument, making it challenging for readers to verify the results.

**Questions:**

Compared to Chen et al. (2023a, b), what are the non-trivial contributions of the theoretical analysis?

---

> ### Author Response · Authors · 2024-11-21
> **Our response to Reviewer y6ua (Part I)**
>
> We appreciate the reviewer’s thoughtful feedback and provide a detailed clarification to address the reviewer’s concerns:
>
> **(Novelty regarding continuous LCQP).** While our work draws some inspiration from (Chen et al., 2023a), there are substantial differences in conclusions regarding LPs and QPs. Consider the following two LPs:
>
> $$\min \sum_{j=1}^6 x_j ~~ \textup{s.t.}~ x_1 + x_2 = 2,x_2 + x_3 = 2,x_3 + x_4 = 2,x_4 + x_5 = 2,x_5 + x_6 = 2,x_6 + x_1 = 2$$
>
> and
>
> $$\min \sum_{j=1}^6 x_j ~~ \textup{s.t.}~ x_1 + x_2 = 2,x_2 + x_3 = 2,x_3 + x_1 = 2,x_4 + x_5 = 2,x_5 + x_6 = 2,x_6 + x_4 = 2$$
>
> Both of them share a common optimal solution $(1,1,1,1,1,1)$, and hence the two LPs share a common optimal objective value $6$. However, by adding the **same** quadratic term, their optimal solution and optimal objective varies. Specifically, let's consider the two QPs:
>
> $$\min \sum_{j=1}^6 x_j + {\color{blue}{x_1^2}} ~~ \textup{s.t.}~ x_1 + x_2 = 2,x_2 + x_3 = 2,x_3 + x_4 = 2,x_4 + x_5 = 2,x_5 + x_6 = 2,x_6 + x_1 = 2$$
>
> and
>
> $$\min \sum_{j=1}^6 x_j + {\color{blue}{x_1^2}} ~~ \textup{s.t.}~ x_1 + x_2 = 2,x_2 + x_3 = 2,x_3 + x_1 = 2,x_4 + x_5 = 2,x_5 + x_6 = 2,x_6 + x_4 = 2$$
>
> For the first QP, the optimal solution changes to $(0, 2, 0, 2, 0, 2)$ with an optimal objective value of $6$. In contrast, the second QP retains $(1, 1, 1, 1, 1, 1)$ as the optimal solution, but its objective value increases to $7$. This demonstrates that conclusions regarding optimal objectives and solutions for LPs cannot be directly extended to LCQPs.
>
> Our findings show that GNNs can effectively capture such differences. Notably, while Theorem 3.2 (on feasibility) can be directly adapted from LP results, Theorems 3.3 and 3.4, which address the objective and solution, introduce significant novel contributions. The new techniques are mainly in the proofs of Lemma Lemma A.4 and Theorem A.2.
>
>
> **(Novelty regarding mixed-integer LCQP).** We acknowledge that the concept of unfoldable MI-LCQP is inspired by the definition of MILPs in (Chen et al., 2023b). However, we introduce several novel contributions in this paper:
>
> - **(MP-tractability).** In Theorems 4.6 and 4.7, we use a new concept named **MP-tractability** instead of unfoldability in (Chen et al., 2023b). MP-tractability is a weaker assumption than unfoldability: while all unfoldable MI-LCQPs are MP-tractable (as rigorously proved in Appendix D), not all MP-tractable problems are unfoldable. This difference can be clearly illustrated with an example that is MP-tractable but not unfoldable:
> $$\min ~ \frac{1}{2}x_2^2 + x_1 + x_2 + x_3, ~~
> \textup{s.t. } x_1 + x_3 \leq 1, ~
>   x_1 - x_2 + x_3\leq 1, ~
>   0 \leq x_1,x_2,x_3\leq 1, ~
>   x_1,x_2,x_3 \in \mathbb{Z}.$$
> In the above example, $x_1$ and $x_3$ exhibit symmetry --- swapping their positions does not change the problem. Generally, unfoldable problems lack such symmetry, whereas MP-tractability allows for some degree of symmetry. Therefore, MP-tractability is a strictly weaker assumption for MI-LCQPs. Related discussions can be found in Section 4.3 and Appendix D.
> Our results (Theorems 4.6 and 4.7) demonstrate that MP-tractability suffices to represent feasibility and the optimal objective of MI-LCQPs, while unfoldability remains necessary for representing the optimal solution (Theorem 4.9). As MILP is a special case of MI-LCQP, our findings can be applied on MILPs. Overall, our results are a clear advancement rather than a direct adaptation.
> - **(Counterexamples regarding objective and solution).** In (Chen et al., 2023b), the authors demonstrate that there exist two MILPs -- one feasible and the other infeasible -- that cannot be distinguished by any GNNs. However, their work does not address differences in objective values or optimal solutions. In contrast, our paper provides a more comprehensive analysis in Section 4.1. While Proposition 4.1 is a straightforward extension, as feasibility depends only on linear constraints, Propositions 4.2 and 4.3 introduce novel insights. Specifically, in Proposition 4.2, we show that even if two MI-LCQPs are both feasible and indistinguishable by GNNs, they can still have different optimal objective values. Furthermore, in Proposition 4.3, we demonstrate that even when two MI-LCQPs (indistinguishable by GNNs) share the same optimal objective value, they may still have different optimal solutions. Notably, these examples are not trivial extensions of those in (Chen et al., 2023b) but are constructed to highlight distinct differences.

---

> ### Author Response · Authors · 2024-11-21
> **Our response to Reviewer y6ua (Part II)**
>
> **(Novely regarding QCQPs: Added in rebuttal).** To add more nontrivial theoretical contributions, we extended our techniques and proved that GNNs can universally approximate the properties of quadratically constrained quadratic programs (QCQPs), which requires hyperedges and is significantly different from (Chen et al., 2023a; Chen et al., 2023b). More specifically, we add hyperedges in encode the quadratic terms in the constriants and the GNN architecture involving edge features is proved with sufficiently strong separation power for QCQPs. In comparison, GNNs in (Chen et al., 2023a; Chen et al., 2023b) only use vertex features. Please check Appendix E of our revision.
>
> **(Empirical Validation).** To address this concern, we train GNNs on the Maros and Meszaros Convex Quadratic Programming Test Problem Set, which contains 138 quadratic programs that are designed to be challenging. We apply equilibrium scaling to each problem and also scale the objective function so that the $Q$ matrix will not contain too large elements. We collecte the optimal solutions and objective values of the test instances using an open-sourced QP sovler called PIQP [R1], which is benchmarked to achieve best performances on the Maros Meszaros test set among many other solvers [R2]. PIQP solves 136 problem instances successfully, which are then used to train four GNNs with with embedding size of $64,128,256,512$.
>
> **_The results are included in the updated draft in Appendix F.5._** We observe that while the broad range of numbers of instances in the Maros Meszaros test set caused numerical difficulties for training, GNNs can still be trained to fit the objectives and solutions. And we can observe similar tendency as in the synthesized experiments that the expressive power increases as the model capacity enlarges when we increase the embedding size.
>
> **(Handwaving Claims in Proofs).** We have updated our appendix for the sake of completeness and highlighted the modifications, please kindly refer to our revision.
>
> We hope these clarifications address your concerns. If you have further questions or require additional clarifications, we would be more than happy to provide them. We greatly appreciate it if you could update your evaluation should you find these responses satisfactory.
>
> -------------------------
> ## References added in rebuttal
>
> [R1] Schwan R, Jiang Y, Kuhn D, Jones CN. PIQP: A Proximal Interior-Point Quadratic Programming Solver. In2023 62nd IEEE Conference on Decision and Control (CDC) 2023 Dec 13 (pp. 1088-1093). IEEE.
>
> [R2] https://github.com/qpsolvers/maros_meszaros_qpbenchmark/

---

> ### Author Response · Authors · 2024-11-24
> **Need more clarifications?**
>
> Dear Reviewer y6ua,
>
> Thank you for your insightful comments and constructive feedback. We have carefully reviewed and addressed each of your concerns in our detailed responses.
>
> Given the impending rebuttal deadline, we kindly request your review of our responses to ensure that all issues have been addressed satisfactorily. Should you require further clarification or additional details, we would be happy to provide any necessary explanations.
>
> We truly appreciate your time and effort in reviewing our work.
>
> Best regards.

---

> ### Comment · Reviewer_y6ua · 2024-12-03
> **concerns on limited novelty**
>
> Thanks for the response. I still have some concerns.
> 1. Novelty of continuous LCQP.
> In Section **(Novelty regarding continuous LCQP)**, the authors mention that '_The new techniques are mainly in the proofs of Lemma A.4 and Theorem A.2._'
> Based on my reading of the paper, I think:
> a) The proof framework for continuous LCQP is very much the same as before, with the only difference being Lemma A.4 (half page proof).
> b) Lemma A.4 and Thm A.2 are new, but relatively straightforward. Lemma A.4 and Thm A.2 primarily address the non-linearity introduced by LCQP. Previously, linearity ensures re-assgining values of coordinates of an optimal solution will keep the objective value. For the current problem, the convexity (ensured by positive semidefiniteness of the quadratic terms) ensures that the re-assigining will not increase the objective value. This modification seems to be relatively straightforward.
>
> 2. Novelty of mixed-integer LCQP.
> In Subsection **MP-tractability** in Section **Novelty regarding mixed-integer LCQP**, the aurthors wrote that "_In Theorems 4.6 and 4.7, we use a new concept named **MP-tractability** instead of unfoldability in (Chen et al., 2023b)._”
> However, the concept of MP-tractability was introduced in an earlier work [1] for MILP. The discussion on the relationship between unfoldable and MP-tractable has also appeared in [1].
> Thus, it seems that the contribution of the current paper is to apply MP-tractability to MI-LCQP.  The adaptation is relatively straightforward.
>
> Overall, I still think the significant overlap with Chen et al. (2023a), and the relation with [1] shows the lack of sufficient novelty.
>
> **Reference:**
> [1] Chen, Ziang, et al. "Rethinking the capacity of graph neural networks for branching strategy." _arXiv preprint arXiv:2402.07099_ (2024).

---

> ### Author Response · Authors · 2024-12-04
>
> Thank you very much for your reply and for your comments. The reviewer's concern primarily focuses on the contributions of our paper. Let us address this by considering the following questions:
> * **Is QP important?** Yes, QP is a fundamental problem with extensive applications in operations research, finance, scientific computing, and more. Advancing our understanding of QPs and their interaction with GNNs is both impactful and relevant to the research community.
> * **Do the previous results on LP (or MILP) directly imply results on QP?** No. LP is a special case of QP, so while results for QP can imply results for LP, the reverse is not true. This distinction, combined with the significance of QP, highlights the necessity of the results and conclusions presented in our paper.
> * **Does our work deliver useful insights to the community?** Yes, our paper offers clear guidance to practitioners, particularly in understanding what GNNs can and cannot achieve for QP tasks.
>
> Therefore, we believe our findings are meaningful and merit publication.
>
> Regarding connections to prior works, we have already addressed them transparently and will further clarify them in our revision:
> * Chen et al. (2023a): All conclusions and lemmas directly used from this paper have been clearly acknowledged in the main text and the appendix.
> * [1]: In our revision, we will explicitly clarify that the criteria in our paper extend the concept introduced in [1], with the key difference being the additional requirements on the matrix $Q$, which reflect the unique properties of QPs.

---

### Official Review · Reviewer_n1f6 · 2024-11-08

**Soundness:** 3
**Presentation:** 3
**Contribution:** 2
**Rating:** 5
**Confidence:** 5

**Summary:**

This paper studies whether GNN can represent key properties (such as feasibility, optimal value, and optimal solution) of LCQP (Linearly Constrained Quadratic Programming) and MI-LCQP (Mixed-Integer LCQP) problems. The study establishes the following theoretical results:

1. **LCQP**: GNNs can universally represent the feasibility of LCQPs. For **convex** LCQPs, GNN can also universally represent the optimal values and optimal solutions.

2. **MI-LCQP**: GNNs can NOT universally represent MI-LCQPs in terms of either feasibility, optimal values, or optimal solutions. The authors provide counter-examples to prove it. Then, the authors identify a subset of MI-LCQPs, on which the GNNs can universally represent MI-LCQPs in terms of all the three key properties.

The authors further provide small-scale experimental evidence to support the theoretical results.

**Strengths:**

1. The paper studies a crucial topic in the field of learning to optimize (L2O). GNNs have been extensively utilized for learning key properties—primarily optimal solutions—of optimization problems. By focusing on the representational power of GNNs in this context, the study offers theoretical insights and potential practical guidance.

2. In previous work, significant progress has been made in analyzing the representation power of GNNs in LP and MILP. Extending this analysis to nonlinear programming problems like LCQP and MI-LCQP is good. This paper made a commendable exploration of these extensions.

3. The paper is well-written and highly accessible, making it easy to follow.

**Weaknesses:**

My major concern lies in distinguishing the analysis in this paper from previous studies of GNN's representation power on LP/MILP (Chen et al. (2023a), Chen et al. (2023b)). Two significant similarities raise questions about the originality of the analysis technique:

1. To prove the universal representation power of LCQP,  the authors leverage the Stone-Weierstrass theorem to establish the GNN's approximation capabilities based on their separation capabilities. This approach appears identical to that used by Chen et al. (2023a) for LPs. In particular, the proof ideas of Theorems 3.3 and 3.4 (illustrated on Page 5) seem directly translatable to LPs, with minimal modification (i.e., replacing LCQPs with LPs). If this understanding is correct, the theoretical innovation for LCQPs might be limited.

2. The definition of unfoldable MI-LCQPs, which allows GNNs to achieve universal representation, closely mirrors the definition of unfoldable MILPs in Chen et al. (2023b). Furthermore, the proof techniques for establishing the representation power for unfoldable MI-LCQPs appear largely similar to those used for unfoldable MILPs in the prior work.

In summary, while the paper provides a comprehensive extension to LCQPs and MI-LCQPs, it is unclear whether its main proofs represent substantial innovations or are direct (albeit not completely trivial) extensions of the methodologies in Chen et al. (2023a, 2023b).

**Questions:**

I will consider changing my opinion if the authors can address my comments in "weakness".

---

> ### Author Response · Authors · 2024-11-21
> **Our response to Reviewer n1f6 (Part I)**
>
> We appreciate the reviewer’s thoughtful feedback, particularly regarding concerns about the novelty of our work. Below, we provide a detailed clarification of our contributions to address these concerns:
>
> **(Continuous LCQP).** While our work draws some inspiration from (Chen et al., 2023a), there are substantial differences in conclusions regarding LPs and QPs. Consider the following two LPs:
>
> $$\min \sum_{j=1}^6 x_j ~~ \textup{s.t.}~ x_1 + x_2 = 2,x_2 + x_3 = 2,x_3 + x_4 = 2,x_4 + x_5 = 2,x_5 + x_6 = 2,x_6 + x_1 = 2$$
>
> and
>
> $$\min \sum_{j=1}^6 x_j ~~ \textup{s.t.}~ x_1 + x_2 = 2,x_2 + x_3 = 2,x_3 + x_1 = 2,x_4 + x_5 = 2,x_5 + x_6 = 2,x_6 + x_4 = 2$$
>
> Both of them share a common optimal solution $(1,1,1,1,1,1)$, and hence the two LPs share a common optimal objective value $6$. However, by adding the **same** quadratic term, their optimal solution and optimal objective varies. Specifically, let's consider the two QPs:
>
> $$\min \sum_{j=1}^6 x_j + {\color{blue}{x_1^2}} ~~ \textup{s.t.}~ x_1 + x_2 = 2,x_2 + x_3 = 2,x_3 + x_4 = 2,x_4 + x_5 = 2,x_5 + x_6 = 2,x_6 + x_1 = 2$$
>
> and
>
> $$\min \sum_{j=1}^6 x_j + {\color{blue}{x_1^2}} ~~ \textup{s.t.}~ x_1 + x_2 = 2,x_2 + x_3 = 2,x_3 + x_1 = 2,x_4 + x_5 = 2,x_5 + x_6 = 2,x_6 + x_4 = 2$$
>
> For the first QP, the optimal solution changes to $(0, 2, 0, 2, 0, 2)$ with an optimal objective value of $6$. In contrast, the second QP retains $(1, 1, 1, 1, 1, 1)$ as the optimal solution, but its objective value increases to $7$. This demonstrates that conclusions regarding optimal objectives and solutions for LPs cannot be directly extended to LCQPs.
>
> Our findings show that GNNs can effectively capture such differences. Notably, while Theorem 3.2 (on feasibility) can be directly adapted from LP results, Theorems 3.3 and 3.4, which address the objective and solution, introduce significant novel contributions. The new techniques are mainly in the proofs of Lemma Lemma A.4 and Theorem A.2.
>
> **(Mixed-integer LCQP).** We acknowledge that the concept of unfoldable MI-LCQP is inspired by the definition of MILPs in (Chen et al., 2023b). However, we introduce several novel contributions in this paper:
>
> - **(MP-tractability).** In Theorems 4.6 and 4.7, we use a new concept named **MP-tractability** instead of unfoldability in (Chen et al., 2023b). MP-tractability is a weaker assumption than unfoldability: while all unfoldable MI-LCQPs are MP-tractable (as rigorously proved in Appendix D), not all MP-tractable problems are unfoldable. This difference can be clearly illustrated with an example that is MP-tractable but not unfoldable:
> $$\min ~ \frac{1}{2}x_2^2 + x_1 + x_2 + x_3, ~~
> \textup{s.t. } x_1 + x_3 \leq 1, ~
>   x_1 - x_2 + x_3\leq 1, ~
>   0 \leq x_1,x_2,x_3\leq 1, ~
>   x_1,x_2,x_3 \in \mathbb{Z}.$$
> In the above example, $x_1$ and $x_3$ exhibit symmetry --- swapping their positions does not change the problem. Generally, unfoldable problems lack such symmetry, whereas MP-tractability allows for some degree of symmetry. Therefore, MP-tractability is a strictly weaker assumption for MI-LCQPs. Related discussions can be found in Section 4.3 and Appendix D.
> Our results (Theorems 4.6 and 4.7) demonstrate that MP-tractability suffices to represent feasibility and the optimal objective of MI-LCQPs, while unfoldability remains necessary for representing the optimal solution (Theorem 4.9). As MILP is a special case of MI-LCQP, our findings can be applied on MILPs. Overall, our results are a clear advancement rather than a direct adaptation.
> - **(Counterexamples regarding objective and solution).** In (Chen et al., 2023b), the authors demonstrate that there exist two MILPs -- one feasible and the other infeasible -- that cannot be distinguished by any GNNs. However, their work does not address differences in objective values or optimal solutions. In contrast, our paper provides a more comprehensive analysis in Section 4.1. While Proposition 4.1 is a straightforward extension, as feasibility depends only on linear constraints, Propositions 4.2 and 4.3 introduce novel insights. Specifically, in Proposition 4.2, we show that even if two MI-LCQPs are both feasible and indistinguishable by GNNs, they can still have different optimal objective values. Furthermore, in Proposition 4.3, we demonstrate that even when two MI-LCQPs (indistinguishable by GNNs) share the same optimal objective value, they may still have different optimal solutions. Notably, these examples are not trivial extensions of those in (Chen et al., 2023b) but are constructed to highlight distinct differences.

---

> ### Author Response · Authors · 2024-11-21
> **Our response to Reviewer n1f6 (Part II)**
>
> **(QCQPs: Added in rebuttal).** To add more nontrivial theoretical contributions, we extended our techniques and proved that GNNs can universally approximate the properties of quadratically constrained quadratic programs (QCQPs), which is significantly different from (Chen et al., 2023a; Chen et al., 2023b). More specifically, we add **hyperedges** to encode the quadratic terms in the constriants and the GNN architecture involving edge features is proved with sufficiently strong separation power for QCQPs. In comparison, GNNs in (Chen et al., 2023a; Chen et al., 2023b) only use vertex features. Please check Appendix E of our revision.
>
>
> We hope these clarifications address your concerns. If you have further questions or require additional clarifications, we would be more than happy to provide them. We greatly appreciate it if you could update your evaluation should you find these responses satisfactory.

---

> ### Author Response · Authors · 2024-11-24
> **Need more clarifications?**
>
> Dear Reviewer n1f6,
>
> Thank you for your insightful comments and constructive feedback. We have carefully reviewed and addressed each of your concerns in our detailed responses.
>
> Given the impending rebuttal deadline, we kindly request your review of our responses to ensure that all issues have been addressed satisfactorily. Should you require further clarification or additional details, we would be happy to provide any necessary explanations.
>
> We truly appreciate your time and effort in reviewing our work.
>
> Best regards.

---

> ### Comment · Reviewer_n1f6 · 2024-12-02
>
> Thank you to the authors for providing a detailed reply. However, I find that some of my concerns remain unaddressed:
>
> **Theoretical Framework**: My concern is not about the difference between LP and QP but rather about the theoretical framework used in the paper. Specifically, the approach appears to be a direct extension of the framework used by Chen et al. (2023a) for LPs. I repeat my earlier comment here:
>
> _“… the authors leverage the Stone-Weierstrass theorem to establish the GNN's approximation capabilities based on their separation capabilities. **This approach appears identical to that used by Chen et al. (2023a) for LPs.** In particular, the proof ideas of Theorems 3.3 and 3.4 (illustrated on Page 5) seem **directly translatable to LPs, with minimal modification** (i.e., replacing LCQPs with LPs)…”_
>
> **Broader Class of MP-Tractable Problems**: I understand that all unfoldable MI-LCQPs are MP-tractable, but not vice versa. However, it has been proved that MI-LCQPs are unfoldable almost surely under some mild conditions. What are the significance and importance of identifying the broader class of MP-tractable?
>
> **Extension to QCQP**:
> I appreciate the authors’ effort to add an extension to QCQP in the appendix. However, I have concerns regarding the timing and relevance of this additional contribution for the following reasons:
> 1. The conclusion that "GNNs can universally represent convex QCQP" overlaps significantly with another ICLR submission ([link](https://openreview.net/attachment?id=68J0pJFCi3&name=pdf)). Given that all ICLR submissions are publicly accessible during the rebuttal period, I encourage the authors to explicitly elaborate on how their newly added content differs from the existing submission. (Please correct me if I am misunderstanding the principles regarding overlap and independence.)
> 2. While the original submission focuses on LCQP, the extension to QCQP feels like a late addition that is not well integrated into the main narrative of the paper. To incorporate this result, the authors have made significant modifications to the graph representation and corresponding GNN formulation. These changes do not seem like a natural extension of the LCQP framework but rather a distinct and separate development.

---

> ### Author Response · Authors · 2024-12-04
>
> Thank you very much for your reply and for your comments. Please find our point-to-point response below.
>
> __Extension to QCQP (1):__ Thanks for pointing out this concurrent paper. The main difference between their work and ours is the **_structure of the graph representation_**. In their paper, QCQPs are represented as a tripartite graph (with three groups of nodes representing variables, constraints, and quadratic terms), whereas our approach retains a bipartite structure with additional hyperedges representing coefficients of quadratic terms. These two approaches are significantly different and they are developed independently.
>
> Here **_we do not intend to evaluate which approach is better_**, especially given the limited time remaining before the rebuttal deadline. However, we believe that comparing these two methods would be an interesting direction for future research. In our revision, we will explicitly highlight this concurrent paper and clarify the key differences.
>
> __Extension to QCQP (2):__ Despite that our results/analysis are different for LCQPs and QCQPs, they are closely related. Specifically, **_the QCQP-graph is obtained by adding additional hyperedges in the LCQP-graph_**, rather than developing an entirely new architecture from scratch. We will clearly highlight this connection in our revision. Therefore, we believe that our QCQP results are suitable to be included in our current paper as an extension.
>
>
> __Broader Class of MP-Tractable Problems:__ In our paper (Appendix D), we prove that MI-LCQPs are almost surely unfoldable under the assumption that the linear objective coefficients $c \in \mathbb{R}^n$ are continuously distributed. However, in practice, particularly with manually designed instances (as opposed to randomly generated ones), **_foldability does occur_**, as discussed in Section 4.3. To further address the reviewer's concern, we performed additional experiments on **_QPLIB_** (https://qplib.zib.de/). In this dataset, 19 out of 96 binary-variable linear constraint instances are foldable. Furthermore, when coefficients are quantized (which increases the level of symmetry in the dataset), the ratio of foldable instances increases. (shown in the table below)
>
> Quantization refers to rounding continuous or high-precision values to discrete levels. For example, rounding coefficients of QP instances to the nearest integer reduces precision but can simplify analysis and potentially uncover additional properties. In our study, we examine foldable instances at different quantization step sizes (0.1, 0.5, and 1) to explore their tractability.
>
> While foldable instances are generally challenging for GNNs, **_a significant proportion of these hard instances are MP-tractable_**. For example, with a quantization step size of 1 (rounding coefficients to integers), 36 out of 63 foldable instances are MP-tractable. According to our theorems, GNNs can at least predict feasibility and boundedness (objective value) for such instances. Detailed results are as follows:
>
> |                           |  Quantization step size 0.1 | Quantization step size 0.5 | Quantization step size 1 |
> |---------------------------|---------------|---------------|-------------|
> | Total num. of instances                    |  96            | 96            | 96          |
> | Foldable (Hard) instances                  | 23            | 42            | 63          |
> | Foldable but MP-tractable | 1             | 19             | 36          |
>
> The results clearly show the significance of identifying the broader class of MP-tractable, we will add the results to our revision.
>
> __Theoretical Framework:__ We agree with you that our analysis for LCQPs is generalized from Chen et al. (2023a) for LPs. We believe that introducing such analysis and results to the QP community is a valid contribution and is meaningful, regardless of technical difficulty, as QP is an important family of nonlinear optimization with numerous real-world applications and GNNs are widely applied for solving QPs. Additionally, we have other technical contributions such as richer counter-examples for MI-LCQPs and the analysis for QCQPs.

---

### Author Response · Authors · 2024-11-21
**Revised paper has been uploaded**

We have uploaded the revised version of our paper, incorporating modifications to **both the main text and the appendix** to address the reviewers' concerns. All revisions are highlighted in blue. If you have further questions or require additional clarifications, we would be more than happy to provide them.

---

### Meta-Review · Area_Chair_b6Xe · 2024-12-21

**Metareview:**

This paper studies whether GNN can express solutions and feasibility properties of LCQP and MI-LCQP  problems. To this end, they extend existing proof techniques.

This paper received a mixed bag of reviews. The strengths include the topic's importance, examples of positive and negative results, and the paper's organization. The weaknesses include the writing of the proofs is not completely clear (one reviewer mentioned "handwaving" and another reviewer mentioned that they were difficult to follow); and the limited novelty given how similar the results and proof techniques are to Chen et al. (2023a, 2023b). To respond to this last weakness, the authors added additional theoretical results for QCQPs during the rebuttal period. These new results were not reviewed by most of the reviewers. The only reviewer that directly acknowledged them claimed that this addition wasn't well integrated into the main narrative of the paper.

**Additional Comments On Reviewer Discussion:**

The authors and reviewers engaged in extensive discussions. Most reviewers were satisfied by the authors' response except for reviewers y6ua and n1f6.

For reviewer y6ua, the main weakness is the lack of novelty and potential overlap with two recent papers. The author's response did not convince this reviewer; their final score was 3.

Reviewer n1f6 gave this paper a 5. They mantained their claims about the lack of novelty and overlap with existing works, and they didn't appreciate the addition of the new results for QCQPs, claiming that they were not well integrated into the main narrative of the paper. They also requested the authors to comment on a concurrent ICLR revision, but that request I think is unfair.

Reviewer aVvJ gave this paper a 6. However, they concluded their response by saying that overall, the results are not so insightful and their applicability is limited. And that the writing of the article and presentation of the results could be improved.

All other reviewers gave the paper a 6 or more and were satisfied with the authors' responses.

---

### Decision · Program_Chairs · 2025-01-22

Reject